# ORACLEKV: ORACLE GUIDANCE FOR QUESTION-INDEPENDENT KV CACHE EVICTION

## ABSTRACT

Key-Value (KV) caching is a widely adopted technique in large language models (LLMs) to accelerate long-context inference. While recent studies predominantly focus on question-dependent KV cache eviction where cache entries are evicted based on known queries. In this paper, however, we observe these approaches often fail in question-independent scenarios, such as multi-turn dialogues and chunk pre-caching in retrieval-augmented generation (RAG), where future queries remain unknown. Our empirical analysis reveals that most existing KV cache eviction methods underperform in this setting due to their heavy reliance on importance metrics derived from the attention score with question tokens. The core challenge here is to conduct well-founded estimation on token importance without access to future questions. To address this, we propose OracleKV for question-independent KV cache eviction. OracleKV operates by steering model's attention with an oracle guidance containing surface-level statistics of user preferences from large-scale real-world dialogues. Unlike existing methods, OracleKV operates at the data level, allowing seamless integration with other eviction algorithms in a plug-and-play manner. Experiments on several multi-turn and single-turn benchmarks demonstrate that OracleKV achieves higher accuracy-latency tradeoff than existing KV cache compression approaches. We hope our approach will expand the design space and serve as a solid baseline for future research in KV cache compression.

## 1 INTRODUCTION

Recently, long-context capabilities have become a standard feature in large language models (LLMs) (OpenAI, 2023; Anthropic, 2024; Meta, 2024; 2025; Yang et al., 2024b; Achiam et al., 2023)). For example, GPT-4.1 (OpenAI, 2023; Achiam et al., 2023) can process up to 1M tokens, Claude 3.7 (Anthropic, 2024) supports a 200K-token context window, and the instruction-tuned version of LLaMA-4 (Meta, 2025) extends this further to 10M tokens. These models exhibit remarkable potential on long-context tasks, achieving groundbreaking performance on various language understanding and generation benchmarks (Hendrycks et al., 2021; Bai

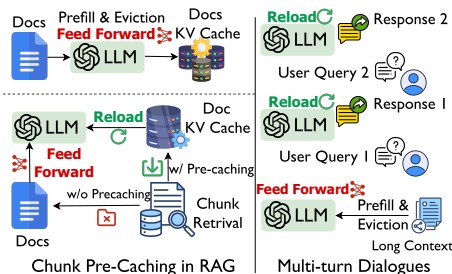

Figure 1: Question-independent KV cache eviction scenarios.

et al., 2024; Li et al., 2023; Ghazal et al., 2013). However, the Key-Value (KV) cache used during inference scales linearly with both sequence length and batch size, leading to substantial memory and computational overhead for long-sequence inference in LLMs (Zhang et al., 2023; Liu et al., 2024b).

Evidence from several studies (Liu et al., 2024b; Zhang et al., 2023; Mu et al., 2023) suggests that only a small subset of the KV cache contributes to the majority of the model's attention. As a result, many KV cache compression methods (Li et al., 2024b; Xiao et al., 2024b; Zhang et al., 2023) have been proposed, leveraging improved important metrics to identify and retain the most informative tokens. Most current KV cache selection approaches (Qin et al., 2025; Cai et al., 2024; Fu et al., 2024; Feng et al., 2024) are based on *observation window* (Li et al., 2024b) selection, which estimates the token importance based on the attention distribution of recent tokens. These methods achieve impressive results on several well-established benchmarks (such as Longbench (Bai et al., 2024)). However, in question-independent scenarios where question is unknown, such as chunks pre-caching

Figure 2: Accuracy without question on RULER benchmark.

Figure 3: Question-aware KV cache compression vs. question-independent KV cache cache compression.

in RAG (Yao et al., 2025) and multi-turn dialogues (Li et al., 2024a) as in Figure 1, we observe their significant performance drop in Figure 2. This motivates the following core questions:

*Why question-independent KV cache compression fails, and how to improve it?*

To answer these questions, our empirical analysis in Figure 4 reveals that the accurate KV entry selection heavily rely on the attention distribution induced by question tokens, which leads to the inadaptability in the question-independent scenarios. Thus, the fundamental challenge of this scenario lies in conducting well-founded estimation of token importance in the absence of target knowledge, *i.e.* without knowing exact questions or prompts the model will respond to, as shown in Figure 3.

To bridge this gap, we propose to find some alternatives to the exact future questions, which help estimate token importance. Recent report (Handa et al., 2025; Maslej et al., 2025) on AI economics observe that in large-scale real user dialogues with LLMs, user-asked question types exhibit strong statistical regularities. Moreover, for each type of question, the associated required information follows predictable distributional patterns (Maslej et al., 2025). Inspired by this, we introduce a method we call **OracleKV**. At a high level, we append a *oracle guidance* to the end of the long context as a substitute of the exact question. This guidance encodes the surface-level statistics about the distribution of future questions, such as the expected types of queries and categories of relevant information, and is designed based on prior user-preference analyses (Handa et al., 2025; Maslej et al., 2025). During inference, we estimate the importance of each context token by measuring its relevance with this oracle guidance. Tokens with low correspondence to the anticipated question distribution are progressively evicted until the retained KV cache size fits the memory budget.

In contrast to prior approaches that rely on token retention heuristics based on internal model-specific computational characteristics (Li et al., 2024b; Feng et al., 2024; Xiao et al., 2024a; Cai et al., 2024; Fu et al., 2024), OracleKV leverages external statistical priors about likely information requirements at the data level. This design makes it highly flexible and model-agnostic: OracleKV can be seamlessly integrated with existing KV cache compression frameworks Li et al. (2024b); Feng et al. (2024); Cai et al. (2024) to enhance their performance, especially under question-independent scenarios.

This paper makes following principal contributions. (1) We identify the root cause of challenges of question-independent KV cache compression (Section 3 and 4.1); (2) We build a theoretical model statistically illustrating the relationship between information induced by the question and required information to answer the question ((Section 4.1). Then, we present a data-level intervention technique, OracleKV, designed to address question-independent KV cache eviction (Section 4.2).(3) Our empirical evaluation shows that OracleKV results in a significant performance increase under the question-independent setting, on both single-turn (e.g., RAG pre-caching) and multi-turn long-context dialogue scenarios, suggesting that OracleKV introduces a useful inductive bias. (Section 5).

## 2 RELATED WORK

**KV Cache Eviction.** Leveraging the inherent sparsity in the self-attention mechanisms, early studies (Liu et al., 2024b; Zhang et al., 2023) propose maintaining a queue with a pre-allocated budget and progressively evicting unimportant cache entries during the inference. StreamingLLM (Xiao et al., 2024b) and LM-Infinite (Han et al., 2024) utilize the *attention sink* phenomenon to retain both initial

and most recent tokens. SnapKV (Li et al., 2024b) uses attention scores with recent tokens to estimate importance. PyramidKV (Cai et al., 2024), PyramidInfer (Yang et al., 2024c) and CAKE (Qin et al., 2025) dynamically adjust KV cache retaining ratio of different layers. DuoAttention (Xiao et al., 2024a) employs a learning-based method to identify *compression-insensitive* attention heads, while HeadKV (Fu et al., 2024) classifies heads based on their retrieval and reasoning utility. However, most existing methods (*e.g.* (Li et al., 2024b; Cai et al., 2024; Yang et al., 2024c; Qin et al., 2025; Feng et al., 2024; Fu et al., 2024; Hao et al., 2025)) rely heavily on importance metrics derived from the attention scores with given the exact question, limiting their robustness and applicability in real-world question-independent scenarios. In contrast, our approach operates at the data level, leveraging surface-level statistical regularities in the question distribution to affect attention behavior, making it compatible with existing methods and easily integrable into a broader range of applications.

**In-Context Learning/Instruction Following.** Early studies (Devlin et al., 2019; Liu et al., 2019) observed that language models can "learn" to perform a task from input-output examples provided at inference. (Xie et al., 2021) interprets the emergence of in-context learning by inferring the shared latent concept among demonstration examples. Based on these, OracleKV affects the attention distribution through in-context data manipulation, aiming to select instruction-correlated tokens.

**Recent Works.** Several recent works evaluate the importance of KV entries without the question. Feng et al. (2025) identify the value states within KV entries are critical, isolated with the attention matrices. However, their approach stem from perturbation analysis and is not specified for question-independent setting. KV-Distill (Chari et al., 2025) employs a distillation-based algorithm to select KV entries but need to retrain the model for days, and may overfit the training data. OracleKV offers more flexible KV cache management for question-independent scenarios, leading to better performance for both single and multi-turn applications. More related works are provided in Section I.

## 3 PRELIMINARY

**Revisit of KV Caching.** Modern LLMs (OpenAI, 2023; Anthropic, 2024; Touvron et al., 2023) typically perform transformer-based auto-regressive generation (Achiam et al., 2023). We begin to revisit the core self-attention (Vaswani et al., 2017) operation. For an attention layer parameterized by projection matrices $\mathbf{W}_Q, \mathbf{W}_K, \mathbf{W}_V$, the query, key, and value are computed by:

$$\mathbf{Q} = \mathbf{X}\mathbf{W}_Q, \quad \mathbf{K} = \mathbf{X}\mathbf{W}_K, \quad \mathbf{V} = \mathbf{X}\mathbf{W}_V, \tag{1}$$

with $\mathbf{X} \in \mathrm{R}^{L \times d}$ of length $L$ and dimension $d$. The self-attention is defined as (Vaswani et al., 2017):

$$\mathrm{Attention}(\mathbf{Q}, \mathbf{K}, \mathbf{V}) = \mathrm{Softmax}(\frac{\mathbf{Q}\mathbf{K}^{\mathrm{T}}}{\sqrt{d}})\mathbf{V} = \mathbf{A}\mathbf{V}, \tag{2}$$

where $\mathbf{A}$ denotes the attention matrix. During auto-regressive generation, each newly generated token $\mathbf{x}_i$ necessitates recalculating $\mathbf{Q}\mathbf{K}^{\mathrm{T}}$, which is inefficient. The goal of KV caching is to transform the recomputation of $\mathbf{Q}\mathbf{K}^{\mathrm{T}}$ to $\mathbf{q}_i\mathbf{K}^{\mathrm{T}}$ by caching the key and value state $\mathbf{K}$ and $\mathbf{V}$ (Pope et al., 2023):

$$\mathbf{K} = \mathrm{Concat}(\mathbf{K}, \mathbf{x}_i\mathbf{W}_K), \quad \mathbf{V} = \mathrm{Concat}(\mathbf{V}, \mathbf{x}_i\mathbf{W}_V) \tag{3}$$

where $\mathbf{q}_i = \mathbf{x}_i\mathbf{W}_Q$. Eq(3) highlights the length of KV cache grows linearly with the input sequence length, results in significant memory footprint and computational costs in long context inference.

**Attention-based KV Cache Eviction.** Generally, $\mathbf{X}$ is a concatenation of context $\mathbf{X}_{\mathrm{ctx}}$ and question $\mathbf{X}_{\mathrm{ques}}$ (or instruction). We denote its KV cache index set as $\Omega$ with sequence length $L = |\Omega|$. KV cache compression targets at exploring a subset of $\Omega$, denoted as $\mathcal{C} = \{l_i\}_{i=1}^{B}$ with size $B$, to maintain the model's capability. StreamingLLM (Xiao et al., 2024b) and LM-Infinite (Han et al., 2024) utilize a heuristic *attention sink* phenomenon to retain both initial and recent KV cache. Score-based methods (Li et al., 2024b; Cai et al., 2024; Qin et al., 2025) select the top $B$ entries based on the attention score within a $L_w$-long window (*i.e.* the observation window) in the tail of the context:

$$\arg\max_{\mathcal{C}} \sum_{i=L-L_w}^{L} \mathbf{A}[i, : -L_w], \quad s.t. \ |\mathcal{C}| = B. \tag{4}$$

**Question-independent Scenarios.** However, the success of these methods relies on using the attention scores of tokens within the observation window to identify relevant tokens. However, in

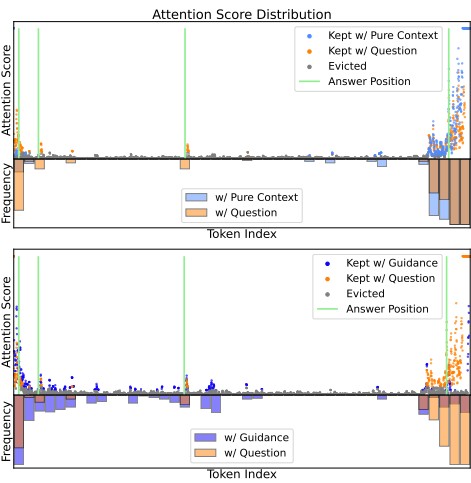

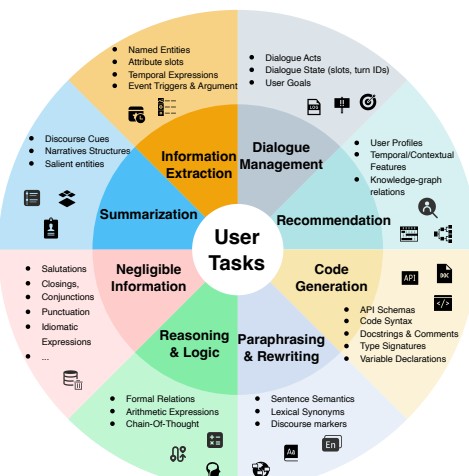

Figure 4: Visualization of attention distribution and kept KV entries (10% budget).

Figure 5: User preferences on task types and frequently involved information for each tasks.

question-independent scenario, question $\mathbf{X}_{\text{ques}}$ is not given during prefilling, *i.e.* $\mathbf{X} = \mathbf{X}_{\text{ctx}}$. When the exact question $\mathbf{X}_{\text{ques}}$ is not given, these methods fail to capture crucial information related to $\mathbf{X}_{\text{ques}}$ in the context $\mathbf{X}_{\text{ctx}}$ due to the question is missing in observation window. as shown in Figure 4.

# 4 ORACLEKV

## 4.1 AN INFORMATION RETAINING PERSPECTIVE OF KV CACHE EVICTION

In this section, we investigate how an LLM can answer a question from the perspective of *retaining* information, even when only a subset of cache entries is preserved. We visualize the attention distribution induced by pure context $\mathbf{X}_{\text{ctx}}$ and the query $\mathbf{X}_{\text{ques}}$, and highlight the top 10% scored kept tokens. Figure 4 reveals that accuracy degradation primarily results from a mismatch between: (1) the KV cache entries retained by a given eviction algorithm, and (2) the entries actually required to answer the question, as measured by their position and relevance with the question token.

In light of this, we build a statistical model to describe the relationship between question answering and the information retaining. Formally, for an answerable question $q$ (*i.e.* $\mathbf{X}_{\text{ques}}$), let $\mathcal{Q}$ denote the ideal set of token indexes that are critical for maintaining the model's ability to answer $q$. The predictive accuracy is positively correlated with $|\mathcal{Q} \cap \mathcal{C}|$, where $\mathcal{C}$ represents the set of retained cache entry indexes (defined in Sec 3). The objective of KV cache compression can be summarized as to optimize the retention process so that the retained entry indexes better align with $\mathcal{Q}$, effectively ensuring that the critical information required to answer the question is preserved. We begin by assume that the information contained by each tokens are associated with the semantic types.

**Assumption 4.1.** *Each KV entry* $\mathbf{KV}_i, i \in \Omega$*, its corresponding token retrained information belongs to one of $K$ semantic "types" (such as topics, concepts, sense, etc.).*

For required cache indexes $\mathcal{Q}$, the KV entries retrained information belongs to type $T_i$ account for:

$$P_{\mathcal{Q}}(T_i) = \frac{|\{\mathbf{KV}_j | \text{type of } \mathbf{KV}_j \in T_i, j \in \mathcal{Q}\}|}{|\mathcal{Q}|}. \tag{5}$$

On the other hand, the retained cache indexes $\mathcal{C}$, under a budget $B = |\mathcal{C}|$, exhibits a type distribution:

$$P_{\mathcal{C}}(T_i) = \frac{|\{\mathbf{KV}_j | \text{type of } \mathbf{KV}_j \in T_i, j \in \mathcal{C}\}|}{|\mathcal{C}|}, \;\; |\mathcal{C}| = B. \tag{6}$$

Our goal is to show that the index overlap of retrained caches and required caches $|\mathcal{Q} \cap \mathcal{C}| \uparrow$ as the semantic type distribution $P_{\mathcal{C}}$ aligns to $P_{\mathcal{Q}}$. Based on Assumption 4.1, we derive theorem:

**Theorem 4.2.** *Let the semantic type of cache entries with index $\mathcal{C}$ be a discrete variable $T_{\mathcal{C}}$, and the semantic type of cache entries with index set $\mathcal{Q}$ be a discrete variable $T_{\mathcal{Q}}$. The expected predictive*

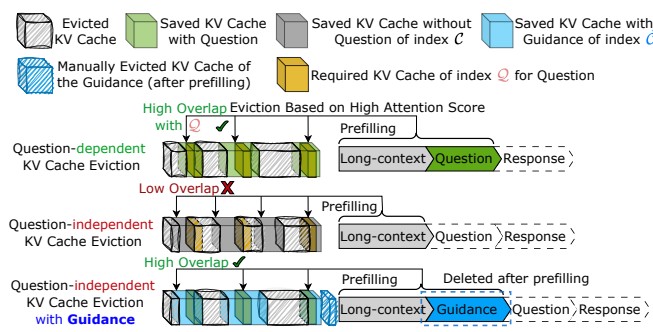

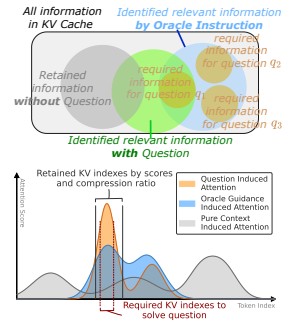

Figure 6: Overview of OracleKV. The KV entries of the context are evicted based on the attention score with oracle guidance.

Figure 7: Attention manipulation of OracleKV.

accuracy is positively correlated to:

$$\mathbb{E}_{T_\mathcal{C} \sim P_\mathcal{C}, T_\mathcal{Q} \sim P_\mathcal{Q}} \left( |\mathcal{Q} \cap \mathcal{C}| \right) \propto 1 - D_{\text{KL}}(P_\mathcal{Q} \,\|\, P_\mathcal{C}). \quad (7)$$

*Remark* 4.3. Theorem 4.2 indicates that the predictive accuracy is inversely correlated with the KL divergence between $P_\mathcal{Q}$ and $P_\mathcal{C}$. As $P_\mathcal{C}$ more closely matches $P_\mathcal{Q}$, the retained entries are more likely to be relevant to the query $q$, thereby improving predictive performance. See proofs in Section C.

## 4.2 ATTENTION MANIPULATION VIA DATA-LEVEL INTERVENTION

Building on the insights from Theorem 4.2, our objective is to align $P_\mathcal{C}(T_i)$ with $P_\mathcal{Q}(T_i)$ as closely as possible, ensuring that the retained entries effectively cover the semantic diversity required to answer the question. Result from Figure 4 provide a possibility to impose constraints on probability mass by manipulating attention over specific regions through data-level manipulation.

Based on the idea, we aim to control the distribution of attention across semantic types in the retained entries. An overview of our method is presented in Fig.6. We begin by manually designing an oracle guidance $\tilde{\mathcal{O}}$ of length $L_o$, which encodes surface statistics of prevalent user-preferred information types in large-scale dialogues. This oracle guidance $\tilde{\mathcal{O}}$ is then appended to the input context $\mathbf{X}_{\text{ctx}}$ as a substitute of $\mathbf{X}_{\text{ques}}$ (question $q$), allowing it to steer the attention distribution during the prefilling:

$$\mathbf{Q} = \mathbf{W}_Q \text{Concat}(\mathbf{X}_{\text{ctx}}, \tilde{\mathcal{O}}), \quad \mathbf{K} = \mathbf{W}_K \text{Concat}(\mathbf{X}_{\text{ctx}}, \tilde{\mathcal{O}}), \quad \mathbf{V} = \mathbf{W}_V \text{Concat}(\mathbf{X}_{\text{ctx}}, \tilde{\mathcal{O}}). \quad (8)$$

To ensure the representativeness of user-preferred information types, we resort to recent studies (Handa et al., 2025; Maslej et al., 2025) that indicates users exhibit distinct preferences across task types when interacting with large language models (LLMs). We summarize the dominant information types associated with common long-context tasks from (Handa et al., 2025; Maslej et al., 2025) in Figure 5 (detailed in Section D.2). Following the instruction format (Zhou et al., 2023), we design $\tilde{\mathcal{O}}$ to encourage the model to assign higher attention score to tokens containing specific information types as in Figure 7. We provide specific examples of $\tilde{\mathcal{O}}$ in Table A3, and we also explore the automation of $\tilde{\mathcal{O}}$ (LLM-as-Guidance) in Section G.2. Based on Figure 4, we make an assumption:

**Assumption 4.4.** *The attention matrix explicitly reflects the semantic correlation between KV entries.*

Our goal is to select the KV entries semantically correlated to the oracle guidance $\tilde{\mathcal{O}}$. Based on the Assumption 4.4, we focus on the attention scores within the cache entry window of $\tilde{\mathcal{O}}$, as illustrated in Fig. 6, and retain the top $B$ entries using attention scores with the oracle guidance (last $L_o$ tokens):

$$\arg\max_\mathcal{C} \sum_{i=L-L_o}^{L} \mathbf{A}[i, :-L_o], \quad \text{s.t. } |\mathcal{C}| = B. \quad (9)$$

Finally, we manually exclude the KV entries corresponding to $\tilde{\mathcal{O}}$ itself, as these surface statistics do not directly contribute to answering the query. Our method explicitly selects highly guidance-correlated cache entries. With Theorem 4.2 and Assumption 4.4, Our method results in a corollary:

**Corollary 4.5.** *Let $\tilde{\mathcal{C}}$ be the retrained index set with oracle guidance $\tilde{\mathcal{O}}$. The oracle guidance $\tilde{\mathcal{O}}$ constrains the probability mass of $P_{\tilde{\mathcal{C}}}$ over semantic regions $R_i$ ($R_i \cap R_j = \emptyset$, $i \neq j$) as follows:*

$$P_{\tilde{\mathcal{C}}}(R_i) = \sum_{T_i \in R_i} P_{\mathcal{C}|\tilde{\mathcal{O}}}(T_i) = \sum_{T_i \in R_i} P_\mathcal{Q}(T_i) = P_\mathcal{Q}(R_i), \quad (10)$$

Table 1: Average performance of various baselines across different LLMs in single-turn Bai et al. (2024) and multi-turn Li et al. (2024a) benchmarks. We compare OracleKV with baselines under 40% and 10% KV budget.

| Method | Budget | Single-turn LongBench | | | | | | Multi-turn SCBench | | | | | | |
|---|---|---|---|---|---|---|---|---|---|---|---|---|---|---|
| | | Sin.QA | Mul.QA | Sum. | Few.Shot. | Syn. | AVG. | M.C. | M.S. | M.F. | QA.En | QA.Ch | Sum. | AVG. |
| *LLaMA-3.1-8B-Instruct* | | | | | | | | | | | | | | |
| Full Cache | 100% | 44.3 | 47.0 | 29.2 | 51.1 | 55.6 | 45.7 | 5.7 | 40.0 | 28.0 | 20.9 | 27.5 | 30.9 | 25.8 |
| StreamingLLM | | 27.9 | 34.6 | 25.7 | 53.4 | 26.1 | 33.5 | 7.9 | 38.2 | 12.2 | 18.0 | 16.3 | 22.3 | 19.1 |
| SnapKV | | 35.3 | 44.8 | 26.5 | 53.6 | 54.5 | 42.9 | 6.1 | **44.4** | 22.5 | 17.8 | 23.5 | 27.7 | 23.7 |
| PyramidKV | | 34.1 | 35.3 | 25.7 | **55.3** | 51.5 | 40.4 | 5.5 | 42.8 | 19.5 | 18.6 | 19.9 | 23.5 | 21.6 |
| Ada SnapKV | 40% | 37.0 | 45.6 | 26.7 | 55.1 | 54.3 | 44.1 | 5.7 | 40.4 | 19.8 | 19.3 | 24.0 | 28.3 | 22.9 |
| DuoAttention | | **42.4** | 44.3 | 27.2 | 53.5 | 52.8 | 44.0 | **8.3** | 39.6 | 15.3 | 18.6 | 25.4 | 29.4 | 22.8 |
| **OracleKV** | | 40.9 | **46.4** | 27.4 | 54.0 | **56.1** | **45.0** | 5.2 | 44.2 | **23.3** | 20.5 | 25.7 | 29.8 | 24.8 |
| **Ada OracleKV** | | 42.0 | 45.5 | **27.6** | 53.4 | 55.8 | 44.9 | 5.7 | 43.0 | 22.9 | 20.5 | **27.2** | **30.5** | **24.9** |
| StreamingLLM | | 20.7 | 24.6 | 21.3 | 51.0 | 10.0 | 25.5 | 6.6 | 38.9 | 12.0 | 15.6 | 10.1 | 20.8 | 17.3 |
| SnapKV | | 22.6 | 29.6 | 21.7 | 51.6 | 29.5 | 30.1 | 6.1 | **50.7** | 21.2 | 15.2 | 14.5 | 22.3 | 21.7 |
| PyramidKV | | 21.3 | 24.0 | 21.7 | 51.8 | 27.1 | 29.2 | 5.2 | 48.8 | 20.6 | 13.6 | 11.0 | 21.4 | 20.1 |
| Ada SnapKV | 10% | 24.1 | 30.7 | 22.3 | 51.7 | 31.5 | 32.1 | 5.7 | 50.4 | 20.3 | 14.7 | 15.9 | 23.0 | 21.7 |
| DuoAttention | | 18.2 | 23.3 | 21.4 | 49.4 | 28.0 | 28.1 | **7.4** | 47.4 | 19.7 | 12.3 | 7.6 | 25.3 | 20.0 |
| **OracleKV** | | 24.5 | 33.2 | 23.5 | 56.3 | 45.3 | 36.5 | 6.1 | 45.2 | **21.3** | **18.5** | 18.1 | 25.9 | 22.5 |
| **Ada OracleKV** | | **28.8** | **34.0** | **23.8** | **58.3** | 49.3 | **38.8** | 5.7 | 50.4 | 21.0 | 17.0 | 19.3 | **26.3** | **23.3** |
| *Mistral-7B-Instruct-v0.2* | | | | | | | | | | | | | | |
| Full Cache | 100% | 32.1 | 24.3 | 27.7 | 55.4 | 38.5 | 35.6 | 11.4 | 64.1 | 5.7 | 6.1 | 10.1 | 24.0 | 20.2 |
| StreamingLLM | | 19.5 | 18.5 | 25.4 | 46.3 | 16.6 | 25.3 | 11.4 | 57.0 | **6.0** | 6.0 | 8.0 | 19.3 | 18.0 |
| SnapKV | | 23.9 | 19.4 | 25.2 | 53.9 | 37.9 | 32.0 | 10.5 | 59.3 | 3.7 | **6.8** | 9.1 | 23.1 | 18.7 |
| PyramidKV | | 23.9 | 20.8 | 25.1 | **54.7** | 34.0 | 31.7 | 11.4 | 61.1 | 5.7 | 6.7 | 9.5 | 22.5 | 19.5 |
| Ada SnapKV | 40% | 24.3 | 20.0 | 24.5 | 53.5 | 35.6 | 31.6 | 11.1 | 58.5 | 5.5 | 6.7 | 9.7 | 23.8 | 19.2 |
| DuoAttention | | 15.3 | 14.4 | 22.4 | 44.0 | 6.3 | 20.5 | 9.6 | 56.3 | 3.5 | 5.9 | 4.6 | 23.7 | 17.3 |
| **OracleKV** | | 25.4 | 21.1 | 26.3 | 53.6 | 37.2 | 32.7 | **11.9** | **62.6** | **5.7** | 6.4 | 9.5 | 23.9 | **20.0** |
| **Ada OracleKV** | | **26.4** | **21.3** | 26.2 | 53.7 | **37.8** | **33.1** | 11.5 | 61.5 | 4.5 | 6.4 | **10.3** | **24.3** | 19.7 |
| StreamingLLM | | 13.9 | 12.5 | 21.3 | 38.6 | 6.9 | 18.7 | **10.5** | 61.1 | 3.0 | 4.5 | 5.0 | 20.0 | 17.4 |
| SnapKV | | 15.1 | 13.5 | 21.8 | 49.5 | 23.0 | 24.6 | 9.6 | 59.4 | 3.2 | **5.7** | 6.2 | 20.5 | 17.4 |
| PyramidKV | | 14.6 | 13.7 | 21.6 | 49.2 | 21.6 | 24.1 | 8.3 | 60.2 | 6.8 | 5.2 | 6.5 | 20.3 | 17.9 |
| Ada SnapKV | 10% | 15.9 | **14.4** | 21.7 | **50.3** | 26.6 | 25.8 | 9.7 | 59.1 | 4.6 | **5.7** | 6.7 | 21.1 | 17.8 |
| DuoAttention | | 14.0 | 13.1 | 20.4 | 40.8 | 5.3 | 18.7 | 9.2 | 48.9 | 2.2 | 4.4 | 3.6 | 21.0 | 14.9 |
| **OracleKV** | | 17.6 | 13.7 | 23.3 | 49.1 | 26.9 | 26.1 | **10.5** | 59.9 | **6.0** | 5.5 | 8.8 | 21.1 | **18.6** |
| **Ada OracleKV** | | **18.3** | 14.1 | **23.5** | 48.3 | **32.6** | **27.4** | **10.9** | 59.7 | 4.3 | **5.7** | 9.1 | 21.9 | 18.6 |

*The following inequality holds:*

$$\mathbb{E}_{T_{\tilde{\mathcal{C}}} \sim P_{\tilde{\mathcal{C}}}, T_{\mathcal{Q}} \sim P_{\mathcal{Q}}} \left( |\mathcal{Q} \cap \tilde{\mathcal{C}}| \right) \geq \mathbb{E}_{T_{\mathcal{C}} \sim P_{\mathcal{C}}, T_{\mathcal{Q}} \sim P_{\mathcal{Q}}} \left( |\mathcal{Q} \cap \mathcal{C}| \right) \tag{11}$$

*Remark* 4.6. Corollary 4.5 shows that OracleKV improves predictive accuracy by introducing oracle guidance to redistribute probability mass over semantic types to better align with $P_{\mathcal{Q}}$, thereby enhancing the retention of relevant entries. The proofs are provided in Section C.

## 5 EXPERIMENT

**Datasets and Backbone LLMs.** We evaluate the performance of OracleKV using several well-established benchmarks. We choose Longbench (Bai et al., 2024), RULER (Hsieh et al., 2024) and Needle-In-A-Haystack (Kamradt, 2023) to evaluate the performance of OracleKV in single-turn dialogues (matching prefix-caching scenarios), and SCBench (Li et al., 2024a) for multi-turn dialogues (matching multi-turn dialogues shared prefix-caching). Our experiments are conducted on three state-of-the-art, open-source, instruction-tuned LLMs: Mistral-7B-Instruct-v0.2, Llama-3.1-8B-Instruct, and Qwen2.5-7B-Instruct, offering context window sizes of 32K, 128K, and 1M.

**Compared Baselines.** We compare OracleKV against several strong baselines categorized as: (1) *Progressive Eviction Methods*, including StreamingLLM (Xiao et al., 2024b) and H2O (Zhang et al., 2023); (2) *Selection-Based Methods*, exemplified by SnapKV (Li et al., 2024b); (3) *Layer-Level Methods*, represented by PyramidKV (Cai et al., 2024); and (4) *Head-Level Methods*, including DuoAttention (Xiao et al., 2024a) and AdaKV (Feng et al., 2024). These baselines offer a diverse range of approaches for comparison, ensuring a comprehensive evaluation of OracleKV. The detailed experiment settings are provided in Section D.1(dataset settings), Section D.2(guidance example), and Section D.3(implementation and justification of compared baselines). Except for ablation studies, we employ the same general oracle guidance for all experiments (second row in Table A3).

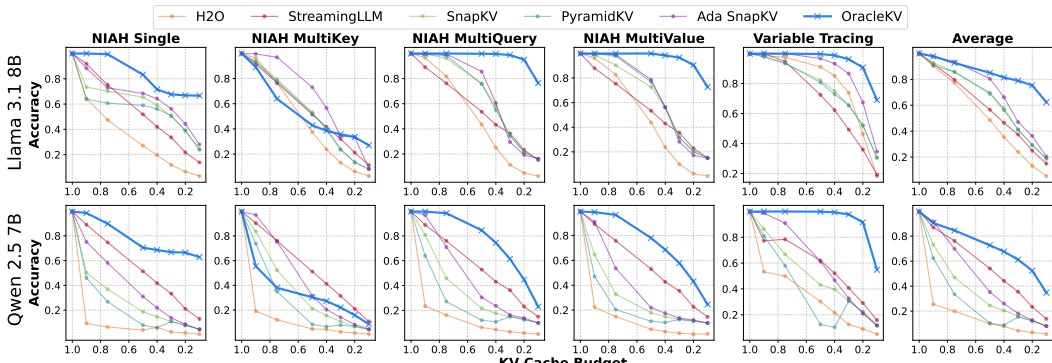

Figure 8: Performance comparison on RULER (Hsieh et al., 2024) benchmark. OracleKV provides superior KV budget and accuracy trade-off on most subtasks.

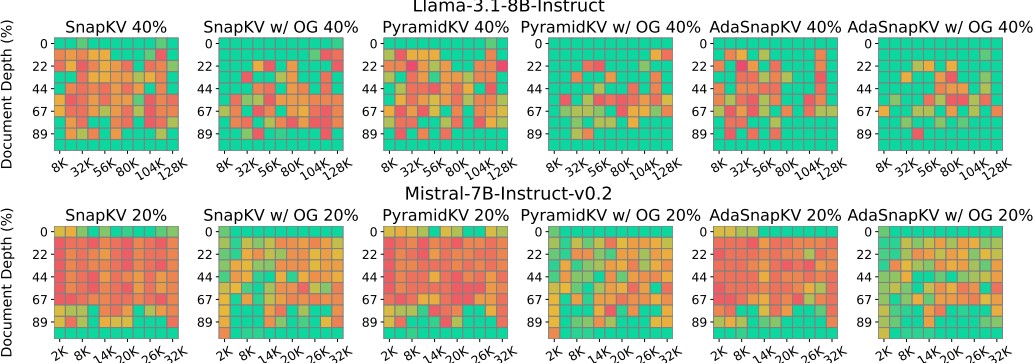

Figure 9: Oracle Guidance provides significant performance increase on the Needle-In-A-Haystack (Kamradt, 2023) pressure test while integrating into uniform (SnapKV(Li et al., 2024b)), layer-wise (PyramidKV (Cai et al., 2024)) and head-wise (Ada SnapKV (Feng et al., 2024)) methods.

## 5.1 ACCURACY EVALUATION

We evaluate OracleKV on four benchmarks: Longbench(Bai et al., 2024), RULER(Hsieh et al., 2024), Needle-In-A-Haystack(Kamradt, 2023), and SCBench (Li et al., 2024a). For Longbench and SCBench, we compare OracleKV against baselines under KV budgets of 40% and 10% with context length of 60K for Llama and 32K for Mistral. Additionally, we integrate the head-level adaptive allocation strategy from(Feng et al., 2024) into OracleKV, denoted as **Ada OracleKV**. Note that the original design of H2O (Zhang et al., 2023) is unsuitable for long-context scenarios due to its quadratic memory cost during the prefilling. We evaluate H2O on the RULER benchmark with 4K context length on both Llama and Qwen. The overall accuracy results are provided in Section E.

**LongBench(Bai et al., 2024)** is a comprehensive multi-task benchmark suite meticulously designed to evaluate the long-context understanding capabilities of LLMs across diverse scenarios. Table 1,A12,A13, and A14 present the performance of various methods across five task types with 14 datasets. OracleKV consistently demonstrates superior performance on most tasks. Notably, under a 10% cache budget, OracleKV and its variant significantly outperforms other methods across solid majority of all tasks on both models, achieving an average improvement of 6.7% for Llama and 1.6% for Mistral. This result highlights its superior adaptability under extreme memory constraints.

**RULER(Hsieh et al., 2024)** is specifically designed benchmark to evaluate a model's ability to identify and retrieve relevant information from long contexts, which includes complex needle-in-a-haystack tests. Figure 8, Table A15,Table A16,Table A17, and Table A18 illustrate the performance across five retrieval subtasks, comparing baselines with KV budgets ranging from 100% to 10%. OracleKV demonstrates an exceptional tradeoff between memory budget and accuracy across most subtasks, highlighting its strong retrieval capabilities. Notably, OracleKV experiences a minimal performance drop (less than 0.1) on three subtasks with only 30% of the KV cache budget.

**Needle-In-A-Haystack (NIAH)**(Kamradt, 2023) is a widely adopted benchmark designed to rigorously assess a model's ability to retrieve a specific string (the "needle") from a context (the

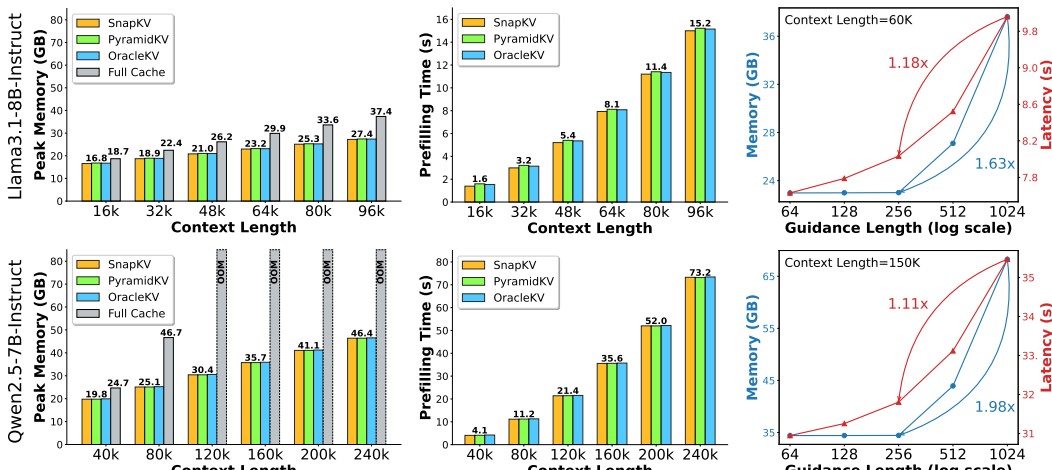

Figure 10: Prefilling latency and memory footprint of OracleKV comparing with other methods across different context length: (1) Comparison of peak memory usage. (2) Comparison of prefilling latency. (3) The computation cost scales with the length of the oracle guidance.

"haystack"). As shown in Figure 9, baseline methods struggle to extract the correct answer from contexts of varying depths. Notably, these methods (including head and layer-level methods) demonstrate significant improvement when integrated with OracleKV, highlighting its remarkable flexibility. This performance boost is particularly evident in deeper contexts, where other methods typically experience sharp declines in retrieval precision. OracleKV's robust handling of long-context scenarios thus proves crucial in improving the model's overall reliability in practical retrieval tasks.

**Multi-turn Benchmark.** To investigate performance in real-world multi-turn dialogues, we evaluate OracleKV along with all baselines on the multi-turn SCBench(Li et al., 2024a). SCBench is a challenging KV-centric multi-turn long-context benchmark that includes various tasks such as QA, choice, summary, and many-shot in-context learning, where each shared context involves at least four turns of dialogue. Table1 shows that OracleKV consistently outperforms all other baselines on most tasks, maintaining superior performance under the same KV budget on both models.

## 5.2 EFFICIENCY EVALUATION

We evaluate the prefilling latency and memory footprint of OracleKV on Llama-3.1-8B-Instruct for 96K context prefilling and Qwen2.5-7B-Instruct for 240K context prefilling. All experiments are conducted with a fixed 4K KV cache budget on a single A100 GPU. Since the primary goal of OracleKV is to compress KV cache in context-only scenarios without on-the-fly requirement, we do not emphasize its decoding efficiency in the main paper. Comprehensive efficiency evaluation results (decoding memory/latency and futher system-level evaluation) are provided in Section F.

**Peak Memory Usage.** As shown in Figure 10(1), OracleKV shows comparable memory savings with uniform budget allocation strategies (SnapKV (Li et al., 2024b)) and layer-pattern budget allocation strategies (PyramidKV (Cai et al., 2024)), both of which significantly reduce memory consumption compared to full attention. Notably, OracleKV saves 26.7% with on Llama model with 96K context length, while saving more than 62% memory on Qwen model with 120K context length.

**Prefilling Latency (Time-To-First-Token).** Figure 10(2) illustrates the prefilling latency for each method. OracleKV achieves comparable prefilling speed to PyramidKV (fixed-pattern allocation) while being marginally slower than SnapKV (uniform allocation), the latency increase accordingly with context length grows. This tradeoff reflects the efficient cache management of OracleKV.

**Computational Cost with Guidance Length.** To further investigate the computational efficiency of OracleKV, we examine how its memory footprint and prefilling latency scale with the guidance length, using context lengths of 64K and 150K. As shown in Figure 10(3), the memory usage increases significantly beyond a guidance length of 512 tokens. For a guidance length of 1K, peak memory usage increases by $1.63\times$ and $1.98\times$ for 64K and 150K contexts, respectively. Prefilling latency (TTFT) also increases, with a $1.18\times$ increase for the 64K context and a $1.11\times$ increase for the 150K context. These findings illustrate the tradeoffs between guidance length and computational efficiency in OracleKV, providing insights into optimal configuration choices for various scenarios.

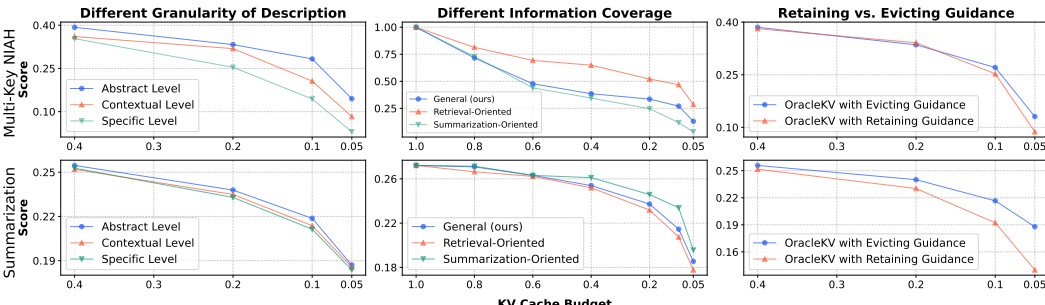

Figure 11: Ablation studies of OracleKV: (1) Comparison of OracleKV with varying descriptive granularity, showing the superior of high-level descriptive oracle guidance. (2) Analysis of OracleKV with guidance with different information coverage, showing the task-specific guidance results in performance increase on corresponding task but with the loss of generalization capability accordingly. (3). Comparison of retraining and evicting the KV entries of oracle guidance.

## 5.3 ABLATION STUDY

We perform ablation studies on the multi-key retrieval and summarization tasks to investigate the effect of various design choices in OracleKV, with Llama-3.1-8B-Instruct model of KV cache budget ranging from full cache to 5% budget. In Section G, we explore the question-dependent performance (Section G.1) of OracleKV, the automation of oracle guidance (LLM-as-Guidance, Section G.2) and further integration with other methods (Section G.3), and we also discuss the performance impact of retaining *vs.* evicting choice of oracle guidance (Section G.4) on specific task.

**Descriptive Granularity.** We examine the effect of descriptive granularity in oracle guidance with three levels: (1)*Abstract Level*: the oracle guidance provides generalized instructions, such as "Please remember the specific details." (2) *Contextual Level*: The guidance specifies information types as "Please remember the numerical information." (3) *Specific Level*: The guidance explicitly lists information types with examples, such as "Please remember numerical information, such as timelines, birthdays, and percentages." As shown in Figure 11(1), contrary to our initial expectations, the abstract-level guidance outperforms other two on both tasks, suggesting that concise, high-level instructions are more effective in guiding the model than overly detailed descriptions.

**Information Coverage of Oracle Guidance.** We further explore the impact of information coverage in oracle guidance by tailoring the guidance to the target task (e.g., specifying "This is a retrieval task."). Figure 11(2) shows that task-specific guidance significantly enhances performance on the corresponding task but leads to performance degradation on other tasks. In contrast, the default general (surface-level) guidance achieves balanced performance across all tasks. This finding indicates that task-specific oracle guidance can significantly boost performance when the task is explicitly known. However, it also impair the model's generalization capabilities, accordingly.

**Retraining vs. Evicting Oracle Guidance.** Finally, we investigate the effect of maintaining or evicting oracle guidance in the KV cache. As shown in Figure 11(3), retaining oracle guidance leads to a substantial performance drop as the KV cache budget decreases, and the gap widens in summarization task. This decline occurs because the oracle guidance is descriptive rather than factual, thus occupies valuable KV cache space without contributing directly to the answer. evicting the oracle guidance effectively mitigating the adverse impact of retaining invasive, non-essential guidance.

## 6 CONCLUSION

We present OracleKV, a data-level intervention approach designed for question-independent KV cache compression. OracleKV steers the attention distribution of by appending an oracle guidance to the pure context. It then selects KV entries that are semantically correlated with oracle guidance based on attention score. Comprehensive experiments across four well-established benchmarks demonstrate OracleKV results in a significant performance increase and acceleration under question-independent setting. We do not claim that OracleKV alone constitutes a state-of-art data-level solution for KV cache compression. Rather, we view it as a promising step toward more adaptive and context-aware cache compression. With extensive validation and development, OracleKV could serve as an useful component within a broader, more comprehensive framework for long-context inference.

## ETHICS STATEMENT

All datasets used in this work are publicly available and have been widely adopted in prior research. They do not involve any personally identifiable or sensitive information. Therefore, this work complies with the ethics requirements of ICLR. The purpose of this work is to improve the inference efficiency and robustness of large language models. The potential risks are consistent with those of general-purpose LLMs, and no additional ethical concerns are introduced. Existing mitigation strategies applicable to LLMs remain sufficient.

In particular, this work is intended to enhance the applicability of large language models in low-resource computational environments and to strengthen their service capability in specific scenarios. We hope this can contribute to broadening equitable access to advanced AI technologies.

We also acknowledge that large language models, including those optimized by our approach, may still inherit biases from training data and could be misused if applied irresponsibly. We encourage the research community to apply our methods with caution, and to further investigate fairness, transparency, and safety in downstream applications.

## REPRODUCIBILITY STATEMENT

All datasets used in this work are publicly available and can be directly downloaded. The preprocessing procedure follows prior research, and additional preprocessing details (e.g., test context length) are provided in Appendix Table A2. The models we used are all open-source, and the core algorithmic details are described in Section 4 of the main text. The most critical component (oracle guidance) is further summarized in Appendix Table A3.

We provide the source code in supplementary material. And we will release a well-organized version, along with experimental scripts and configuration files for open-source community, as soon as possible. All experiments were conducted using three NVIDIA L40S GPUs with 48GB memory (for accuracy experiments) and one NVIDIA A100 GPU with 80GB memory (for efficiency experiments). For each benchmark, compression ratio, and model, the runtime of a single experiment was less than one day.

We have reported comprehensive experimental results, ranging from per-benchmark results (Section 5) to per-task results (Table 1, Figure 8, Figure 9) to per-dataset results (Table A12, Table A13, Table A14, Table A15, Table A16, Table A17,T able A18). Negative results are also included and discussed in Section H.2.

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

APPENDIX

# A  LIMITATION AND BROADER IMPACT

**Limitation.** While OracleKV has been experimentally demonstrated to introduce useful inductive bias for KV cache eviction, it is not exempt from the "no free lunch" theorem–oracle guidance inevitably entails certain side effects. Specifically, we observe that in some specialized tasks, such as code generation, where issues related to token frequency are prominent, the general oracle guidance employed by OracleKV may fail to yield significant improvements. In some cases, it may even degrade the performance of the LLM on these tasks. Though, designing task-specific oracle guidance can enhance performance in such scenarios. However, as highlighted in our ablation study, this approach still suffers from side effects in other tasks.

Moreover, OracleKV involves the computation and explicitly return of the windowed attention matrix, which, to the best of our knowledge, cannot be optimized using memory-efficient techniques like flash attention (Dao et al., 2022; Dao, 2024). This limitation not only leads to notable latency but also results in high memory usage. In environments with rigorous memory peak requirements, such as edge devices, this necessitates carefully design within the length limit of oracle guidance for KV cache eviction. Nevertheless, as demonstrated in our ablation experiments, longer and more detailed oracle guidance does not always correlate with better performance.

**Broader Impact.** OracleKV provides a new perspective to guide KV cache eviction in large language models (LLMs) by leveraging data-level intervention to introduce inductive biases. While OracleKV demonstrates significant performance improvements in question-independent eviction settings, its potential extension to other setting (such as question-aware or task-oriented KV cache eviction) presents a exciting direction for future research. Additionally, the limitations of OracleKV, especially the task-specific nature of its benefits and the increased computational and memory overhead, highlight important trade-offs in the practical deployment. In particular, the increased latency and memory consumption could pose challenges for real-time applications on resource-constrained devices. Furthermore, the need for task-specific oracle design raises concerns about scalability and generalizability, potentially reinforcing disparities between well-resourced and low-resource tasks or domains. We believe that future work should explore more efficient and generalizable oracle guidance designs that balance interpretability, performance, and system efficiency, ensuring that such techniques can be equitably applied across a broad range of use cases.

# B  USE OF LLMS

Large Language Models (LLMs) were used in this work solely as general-purpose assistive tools. Specifically, we used ChatGPT (GPT-4/5), Claude-3.7 and Grok-3 for language polishing, grammar correction, and occasional translation of specific terminology. In some cases, we consulted LLMs for debugging advice (e.g., resolving minor Python errors), but all code implementations, experimental designs, and analyses were performed and verified independently by the authors.

LLMs were not used for research ideation, novelty, methodological design, or interpretation of results. All scientific contributions, claims, and conclusions of this paper are entirely the responsibility of the authors.

# C  COMPLETE ASSUMPTIONS/PROOFS FOR THEORETICAL RESULTS

**Assumption C.1.** type Each KV entry $\mathbf{KV}_i, i \in \Omega$, its retrained information belongs to one of $K$ semantic "types" (such as topics, concepts, etc.).

For required cache indexes $\mathcal{Q}$, the KV entries retrained information belongs to type $T_i$ account for:

$$P_{\mathcal{Q}}(T_i) = \frac{|\{\mathbf{KV}_j | \text{type}(\mathbf{KV}_j) \in T_i, j \in \mathcal{Q}\}|}{|\mathcal{Q}|}. \tag{12}$$

On the other hand, the retained cache indexes $\mathcal{C}$, under a budget $B = |\mathcal{C}|$, exhibits a type distribution given by:

$$P_{\mathcal{C}}(T_i) = \frac{|\{\mathbf{KV}_j | \text{type}(\mathbf{KV}_j) \in T_i, j \in \mathcal{C}\}|}{|\mathcal{C}|}, \quad |\mathcal{C}| = B \tag{13}$$

Our goal is to show that the index overlap of retrained caches and required caches $|\mathcal{Q} \cap \mathcal{C}| \uparrow$ as the semantic type distribution $P_\mathcal{C}$ aligns to $P_\mathcal{Q}$. Based on Assumption C.1, we derive the following theorem.

**Lemma C.2.** (Pinsker's inequality) Let $P$ and $Q$ be two distributions defined on a universe $U$, then

$$D_{\mathrm{KL}}(P \parallel Q) \geq \frac{1}{2}\|P - Q\|_1^2 \tag{14}$$

We first prove the above inequality for the special case of $U = \{0, 1\}$. Then we show how one can prove the general case, by reducing it to the binary case.

*Proof of Lemma C.2.* For the binary case:

$$P = \begin{cases} 1, & \text{w.p. } p \\ 0, & \text{w.p. } 1 - p \end{cases} \qquad Q = \begin{cases} 1, & \text{w.p. } q \\ 0, & \text{w.p. } 1 - q \end{cases} \tag{15}$$

We assume $p \geq q$ (proof of $q \geq p$ is similar), and let

$$f(p, q) = p \log \frac{p}{q} + (1 - p) \log \frac{1 - p}{1 - q} - \frac{1}{2 \ln 2}(2(p - q))^2. \tag{16}$$

Since

$$\frac{\partial f}{\partial q} = -\frac{p - q}{\ln 2}\left(\frac{1}{q(1 - q)} - 4\right) \leq 0, \tag{17}$$

and $f = 0$ when $q = p$, we conclude that $f(p, q) \geq 0$ where $q \leq p$. By change the logarithm base from 2 to $e$, we have $D_{\mathrm{KL}}(P \parallel Q) \geq \frac{1}{2}\|P - Q\|_1^2$ for this special case. We consider the general case. Let $P$ and $Q$ be distributions on $U$, let $A \subset U$ be

$$A = \{x \mid p(x) \geq q(x)\}. \tag{18}$$

And $P_A, Q_A$ be

$$P_A := \begin{cases} 1, & \text{w.p. } \sum_{x \in A} p(x) \\ 0, & \text{w.p. } \sum_{x \notin A} p(x) \end{cases} \qquad Q_A := \begin{cases} 1, & \text{w.p. } \sum_{x \in A} q(x) \\ 0, & \text{w.p. } \sum_{x \notin A} q(x) \end{cases} \tag{19}$$

Then,

$$\|P - Q\|_1 = \sum_x |p(x) - q(x)| \tag{20}$$

$$= \sum_{x \in A}(p(x) - q(x)) + \sum_{x \notin A}(q(x) - p(x)) \tag{21}$$

$$= \left|\sum_{x \in A} p(x) - \sum_{x \in A} q(x)\right| + \left|\left(1 - \sum_{x \notin A} p(x)\right) + \left(1 - \sum_{x \notin A} q(x)\right)\right| \tag{22}$$

$$= \|P_A - Q_A\|_1 \tag{23}$$

To caculate the KL-divergence, we define the random variable

$$Z = \begin{cases} 1, & \text{if } x \in A, \\ 0, & \text{if } x \notin A. \end{cases} \tag{24}$$

Since $Z$ is a function of $X$, we can also think of the two distributions $P$ and $Q$ as joint distributions for the random variables $(X, Z)$. Applying the chain rule for KL-divergence gives

$$D_{\mathrm{KL}}(P \parallel Q) = D_{\mathrm{KL}}(P(X, Z) \parallel Q(X, Z)) \tag{25}$$

$$= D_{\mathrm{KL}}(P(Z) \parallel Q(Z)) + D_{\mathrm{KL}}(P(X|Z) \parallel Q(X|Z)) \tag{26}$$

$$\geq D_{\mathrm{KL}}(P(Z) \parallel Q(Z)) \tag{27}$$

$$= D_{\mathrm{KL}}(P_A \parallel Q_A) \tag{28}$$

$$\geq \frac{1}{2}\|P_A - Q_A\|^2 \tag{29}$$

$$= \frac{1}{2}\|P - Q\|_1^2, \tag{30}$$

which completes the proof.

**Theorem C.3.** (Theorem 4.2) Let the semantic type of cache entries with index $\mathcal{C}$ be a discrete variable $T_{\mathcal{C}}$, and the semantic type of cache entries with index set $\mathcal{Q}$ be a discrete variable $T_{\mathcal{Q}}$. The lowerbound of expected predictive accuracy is positively correlated to:

$$\inf_{\mathcal{C} \subseteq \Omega} \mathbb{E}_{T_{\mathcal{C}} \sim P_{\mathcal{C}}, T_{\mathcal{Q}} \sim P_{\mathcal{Q}}} \left( |\mathcal{Q} \cap \mathcal{C}| \right) \propto 1 - D_{\mathrm{KL}}(P_{\mathcal{Q}} \,||\, P_{\mathcal{C}}). \tag{31}$$

*Proof.* Let $N_i = \{\mathrm{KV}_j \mid \mathrm{type}(\mathrm{KV}_j) \in T_i, j \in \Omega\}$ For a questions with required information type $T_i$, for a long-context, the model is possible to answer it correctly only when the retained KV caches contain the information of type $T_i$. *i.e.*

$$\mathbb{E}\left( |\mathcal{Q} \cap \mathcal{C}| \right) \propto \sum_i \frac{|\mathcal{Q}| P_{\mathcal{Q}}(T_i) \cdot B P_{\mathcal{C}}(T_i)}{N_i} = |\mathcal{Q}| \cdot B \sum_i \frac{P_{\mathcal{Q}}(T_i) P_{\mathcal{C}}(T_i)}{N_i}, \tag{32}$$

$$\propto \sum_{t \in \Omega^{\mathcal{T}}} P_{\mathcal{Q}}(t) P_{\mathcal{C}}(t) = \langle P_{\mathcal{Q}}, P_{\mathcal{C}} \rangle. \tag{33}$$

Where $\Omega^{\mathcal{T}}$ is the type space. Consider

$$\langle P_{\mathcal{Q}}, P_{\mathcal{C}} \rangle = 1 - \frac{1}{2} \sum_{t \in \Omega^{\mathcal{T}}} (P_{\mathcal{Q}}(t) - P_{\mathcal{C}}(t))^2 = 1 - \frac{1}{2} ||P_{\mathcal{Q}} - P_{\mathcal{C}}||_2^2, \tag{34}$$

since $||x||_2 \leq ||x||_1$, we have

$$||P_{\mathcal{Q}} - P_{\mathcal{C}}||_2^2 \leq ||P_{\mathcal{Q}} - P_{\mathcal{C}}||_1^2. \tag{35}$$

By Pinsker's inequality (Lemma C.2), we have

$$||P_{\mathcal{Q}} - P_{\mathcal{C}}||_1 \leq \sqrt{2 D_{\mathrm{KL}}(P_{\mathcal{Q}} \,||\, P_{\mathcal{C}})}, \tag{36}$$

then substitute Eq(36) to Eq(33) and we have

$$\mathbb{E}\left( |\mathcal{Q} \cap \mathcal{C}| \right) \propto \sum_{t \in \Omega^{\mathcal{T}}} P_{\mathcal{Q}}(t) P_{\mathcal{C}}(t) \geq 1 - D_{\mathrm{KL}}(P_{\mathcal{Q}} \,||\, P_{\mathcal{C}}), \tag{37}$$

which completes the proof.

**Corollary C.4.** *Let $\tilde{\mathcal{C}}$ be the retrained index set with oracle guidance $\tilde{\mathcal{O}}$. The oracle guidance $\tilde{\mathcal{O}}$ constrains the probability mass of $P_{\tilde{\mathcal{C}}}$ over specific semantic regions $R_i$ ($R_i \cap R_j = \emptyset$, $i \neq j$) as follows:*

$$P_{\tilde{\mathcal{C}}}(R_i) = \sum_{T_i \in R_i} P_{\mathcal{C}|\tilde{\mathcal{O}}}(T_i) = \sum_{T_i \in R_i} P_{\mathcal{Q}}(T_i) = P_{\mathcal{Q}}(R_i), \tag{38}$$

*The following inequality holds:*

$$\mathbb{E}_{T_{\tilde{\mathcal{C}}} \sim P_{\tilde{\mathcal{C}}}, T_{\mathcal{Q}} \sim P_{\mathcal{Q}}} \left( |\mathcal{Q} \cap \tilde{\mathcal{C}}| \right) \geq \mathbb{E}_{T_{\mathcal{C}} \sim P_{\mathcal{C}}, T_{\mathcal{Q}} \sim P_{\mathcal{Q}}} \left( |\mathcal{Q} \cap \mathcal{C}| \right) \tag{39}$$

*Proof of* Corollary C.4. Based on Theorem C.3, we show above inequality by proving $D_{\mathrm{KL}}(P_{\mathcal{Q}} \,||\, P_{\tilde{\mathcal{C}}}) \leq D_{\mathrm{KL}}(P_{\mathcal{Q}} \,||\, P_{\mathcal{C}})$. We first define a random variable $Z$ that indicates the region to which a type $T$ belongs to $Z = i$ if $T \in R_i$. Since the regions are disjoint and exhaustive, $Z$ is a deterministic function of $T$, The chain rule for KL-divergence allows us to express the divergence over the joint distribution of $(T, Z)$:

$$D_{\mathrm{KL}}(P(T) \,||\, Q(T)) = D_{\mathrm{KL}}(P(Z) \,||\, Q(Z)) + \sum_z D_{\mathrm{KL}}(P(T|Z=z) \,||\, Q(T|Z=z)), \tag{40}$$

where $P(Z)$ and $Q(Z)$ are the marginal distributions over the regions, and $P(T|Z=z)$ and $Q(T|Z=z)$ are the conditional distributions within region $z$. KL-divergence calculation:

$$D_{\mathrm{KL}}(P_{\mathcal{Q}} \,||\, P_{\mathcal{C}}) = D_{\mathrm{KL}}(P_{\mathcal{Q}}(Z) \,||\, P_{\mathcal{C}}(Z)) + \sum_i P_{\mathcal{Q}}(Z=i) D_{\mathrm{KL}}(P_{\mathcal{Q}}(T|Z=i) \,||\, P_{\mathcal{C}}(T|Z=i))$$

$$\tag{41}$$

where

$$P_{\mathcal{Q}}(Z = i) = P_{\mathcal{Q}}(R_i) = \sum_{t \in R_i} P_{\mathcal{Q}}(t) \tag{42}$$

$$P_{\mathcal{C}}(Z = i) = P_{\mathcal{C}}(R_i) = \sum_{t \in R_i} P_{\mathcal{C}}(t) \tag{43}$$

$$P_{\mathcal{Q}}(T = t|Z = i) = \frac{P_{\mathcal{Q}}(t)}{P_{\mathcal{Q}}(R_i)}, \text{ w.r.t. } t \in R \tag{44}$$

$$P_{\mathcal{C}}(T = t|Z = i) = \frac{P_{\mathcal{C}}(t)}{P_{\mathcal{C}}(R_i)}, \text{ w.r.t. } t \in R \tag{45}$$

And then we calculate

$$D_{\mathrm{KL}}(P_{\mathcal{Q}} \| P_{\tilde{\mathcal{C}}}) = D_{\mathrm{KL}}(P_{\mathcal{Q}}(Z) \| P_{\tilde{\mathcal{C}}}(Z)) + \sum_i P_{\mathcal{Q}}(Z = i) D_{\mathrm{KL}}(P_{\mathcal{Q}}(T|Z = i) \| P_{\tilde{\mathcal{C}}}(T|Z = i)) \tag{46}$$

Since the type distribution $P_{\mathcal{C}}$ and $P_{\tilde{\mathcal{C}}}$ are identical on regions $R_i$ in the sense of expectation. or

$$D_{\mathrm{KL}}[P_{\mathcal{Q}}(T|Z = i) \| P_{\tilde{\mathcal{C}}}(T|Z = i)] = D_{\mathrm{KL}}[P_{\mathcal{Q}}(T|Z = i) \| P_{\mathcal{C}}(T|Z = i)] \tag{47}$$

We compare the KL-divergence

$$D_{\mathrm{KL}}(P_{\mathcal{Q}} \| P_{\mathcal{C}}) - D_{\mathrm{KL}}(P_{\mathcal{Q}} \| P_{\tilde{\mathcal{C}}}) \tag{48}$$

$$= [D_{\mathrm{KL}}(P_{\mathcal{Q}}(Z) \| P_{\mathcal{C}}(Z)) - D_{\mathrm{KL}}(P_{\mathcal{Q}}(Z) \| P_{\tilde{\mathcal{C}}}(Z))] \tag{49}$$

$$+ [\sum_i P_{\mathcal{Q}}(Z = i) D_{\mathrm{KL}}(P_{\mathcal{Q}}(T|Z = i) \| P_{\mathcal{C}}(T|Z = i)) \tag{50}$$

$$- \sum_i P_{\mathcal{Q}}(Z = i) D_{\mathrm{KL}}(P_{\mathcal{Q}}(T|Z = i) \| P_{\tilde{\mathcal{C}}}(T|Z = i))] \tag{51}$$

Thus the second term vanishes. For the first term, since

$$D_{\mathrm{KL}}(P_{\mathcal{Q}}(Z) \| P_{\tilde{\mathcal{C}}}(Z)) = \sum_i P_{\mathcal{Q}}(Z = i) \ln \frac{P_{\mathcal{Q}}(R_i)}{P_{\mathcal{C}}(R_i)} \tag{52}$$

$$= \sum_i P_{\mathcal{Q}}(R_i) \ln \frac{P_{\mathcal{Q}}(R_i)}{P_{\mathcal{Q}}(R_i)} = 0, \tag{53}$$

thus we have

$$D_{\mathrm{KL}}(P_{\mathcal{Q}} \| P_{\mathcal{C}}) - D_{\mathrm{KL}}(P_{\mathcal{Q}} \| P_{\tilde{\mathcal{C}}}) = D_{\mathrm{KL}}(P_{\mathcal{Q}}(Z) \| P_{\mathcal{C}}(Z)) \geq 0, \tag{54}$$

which completes the proof.

## D IMPLEMENTATION DETAILS

### D.1 DATASET CONFIGURATION

We adopt four benchmark datasets for our experiments: LongBench(Bai et al., 2024), RULER(Hsieh et al., 2024), and Needle-In-A-Haystack(Kamradt, 2023). Detailed configurations of the datasets used are provided in TableA2.

For LongBench, we evaluate OracleKV and compare it with baseline methods on 14 tasks (excluding code-related tasks). Results for the code tasks are reported separately in the subsequent section. Due to the unavailability of head-level identification files for Qwen2.5(Yang et al., 2024a), we omit the DuoAttention(Xiao et al., 2024a) results on Qwen2.5.

Regarding H2O (Zhang et al., 2023), its original design is not well-suited for long-context inference (e.g., 32K tokens) due to the high memory cost of window attention, which leads to out-of-memory (OOM) errors for context lengths exceeding 11K tokens.

For RULER, we use a 4K context length to evaluate and compare the performance of OracleKV and other baselines on a subset of the tasks.

Table A2: Detailed dataset configuration of all experiments. Our experiments involve four datasets with comprehensive datasets and varying context length.

| Benchmark | Task | Dataset | Average Length | Test Length |
|---|---|---|---|---|
| LongBench (Bai et al., 2024) | Single-Document QA | NartvQA
Qasper
MultiFieldQA-En | 18409
3619
6701 | 60000(Llama)
32000(Mistral) |
| | Multi-Document QA | HotpotQA
2WikiMQA
Musique | 9151
4887
11214 | |
| | Summarization | GovReport
QMSum
MultiNews | 8734
10614
2113 | |
| | Few-shot Learning | TREC
TriviaQA
SAMSum | 5177
8209
6258 | |
| | Synthetic | Passage Count
Passage Retrieval | 11141
9289 | |
| | Code | Lcc
Repobench-p | 1235
4206 | |
| SCBench (Li et al., 2024a) | Multiple Choice | Multiple Choice | 188000 | |
| | In-Context Learning | Many Shot | 22000 | |
| | Math Find | Math Find | 120000 | |
| | Question Answering | QA.En
QA.Ch | 198000
1500000 | |
| | Summarization | Summarization | 18409 | |
| RULER (Hsieh et al., 2024) | NIAH Single | NIAH-Single-1
NIAH-Single-2
NIAH-Single-3 | 4000 | 4000 for Llama and Qwen |
| | NIAH Multikey | NIAH-Multikey-1
NIAH-Multikey-2
NIAH-Multikey-3 | | |
| | NIAH-Multiquery | Summarization | | |
| | NIAH-Multivalue | Summarization | | |
| | Variable Tracing | Variable Tracing | | |
| | Word Extraction | Common Word Extraction
Frequent Word Extraction | | |
| | Question Answering | QA.1
QA.2 | | |
| Needle-In-
-A-Haystack (Kamradt, 2023) | NIAH Single | Synthetic | depth
0%~100% | 2K~32K(Mistral)
8K~128K(Llama) |

## D.2  EXAMPLES OF ORACLE GUIDANCE

In OracleKV, the oracle guidance was designed based on an analysis of frequently queried information derived from prior studies (Handa et al., 2025; Maslej et al., 2025). Specifically, we examined user query patterns as detailed in (Handa et al., 2025) (Section 3.1, Figure 2) to identify semantic types highly relevant to common query types. For instance, tasks categorized as "data analysis" were found to be prevalent across domains such as life sciences, physical sciences, social sciences, finance, administration, and computing, indicating frequent queries involving semantic categories like numbers and entities. Similarly, Handa et al. (2025) (Section 3.2) highlights that content generation tasks, such as writing and marketing, account for over 10% of requests in the arts and media domains. These tasks often involve summarization, suggesting that semantic categories like main themes and timelines are critical. Additionally, we analyzed frequently requested skills (Section 3.2, Figure 3, Appendix D.1, Figures 12 and 13 in (Handa et al., 2025)) and inferred associated semantic types. For example, reading comprehension, which constitutes over 70% of total task requests, typically requires both global (e.g., main theme, topic) and local (e.g., entities such as person names, events) information for effective context understanding. Furthermore, insights from occupational categories (Appendix B.2, Figure 10 in (Handa et al., 2025)) provided additional information on relevant semantic types.

Table A3: Example of Oracle Guidance. The general oracle guidance can yield improvements on most of the tasks. The task-specific oracle guidance can yield significant performance increase on corresponding tasks.

| | |
|---|---|
| General | Next, you will be presented with a series of questions regarding the context above, including specific details about the narrative, content, and numerical information. Also, Do not forget the global information and the relations between entities. And note the structural cues. Please retain these details and provide accurate responses. |
| | Next, you will be presented with a series of questions regarding the context above. including specific details about the narrative, content, numerical information and key global information. The questions may also involve small fragmented relationships from the context, including term relationships, causal relations, and temporal relations. Please retain these details and provide accurate responses. Next, you will be presented with a series of questions regarding the context above. Please remember the following information: 1. specific details like names, places, and numbers; 2. main theme, like overall message; 3. relations, like family ties and event linkages, 4. semantic details, like grammar dependencies and narrative information between words. |
| | Analyze the given text carefully. Your tasks include: 1) Answering factual questions accurately. 2) Generating concise summaries. 3) Demonstrating in-context learning. 4) Writing code based on the text. 5) Counting paragraphs. 6) Retrieving specific strings. 7) Extracting numerical values. 8) Selecting correct answers in multiple-choice questions. 9) Calculating extreme values from arrays. Always ensure your answers are strictly based on the provided text. |
| | Carefully read and analyze the provided text. Your tasks involve multiple types of questions, each requiring precise information extraction from the text. Specifically, you will: 1) Answer factual questions by identifying accurate details directly from the text. 2) Generate concise and coherent summaries without introducing any external information. 3) Demonstrate in-context learning by recognizing patterns or concepts reflected in the text. 4) Write code accurately based on the textual instructions or examples. 5) Count the total number of paragraphs accurately. 6) Search and retrieve specific strings or terms mentioned in the text. 7) Extract and list numerical values, maintaining their original form. 8) Solve multiple-choice questions by selecting the most accurate answer based on the content. 9) Identify and calculate extreme values (maximum, minimum) from any given array of numbers in the text. Always ensure that your responses are strictly grounded in the provided text. Do not infer, assume, or generate information beyond what is explicitly stated. Maintain clarity, accuracy, and completeness in your answers. Stay focused on the input context and prioritize factual consistency. |
| Task-specific | Next, you will be asked with some questions about the context above. These questions includes: question answering, summarization, code completion, in-context learning, paragraph counting, retrival, etc. Please remember the relevant information and answering the question. |
| | Next, you will be asked with some questions about the context above. These questions will ask you to summarize the above context includes: question answering, summarization, code completion, in-context learning, paragraph counting, retrieval, etc. Please remember the relevant information and answering the question. |
| | Next, you will be asked to write a summary of all the contexts above. Please take care of the global information. |
| | Next, you will be asked to find a special number in the context above. Please take care of the relevant information. |

Furthermore, (Handa et al., 2025) reports that over 30% of Claude (Anthropic, 2024) dialogues are dedicated to code generation tasks. In artistic fields, approximately 10% of LLM usage is devoted to writing tasks, including technical writing, advertising copy, and archival work, while report writing and book processing account for 4–6% of total dialogues. Additionally, (Handa et al., 2025) emphasizes the significance of LLMs in supporting writing refinement (*e.g.*, rewriting) and academic reading tasks (*e.g.*, information retrieval and interpretation). Moreover, (Maslej et al., 2025) underscores that the widespread adoption of LLMs hinges on their advanced capabilities in processing long-context information—such as retrieval, reasoning, and summarization—frequently applied in complex scenarios like clinical note processing.

These findings, combined with industry trends reported in (Maslej et al., 2025) (e.g., Section 5 for medical applications and relevant tasks, Section 7 for educational applications and relevant tasks), guided our selection of semantic categories (e.g., numbers, topics, person names) for the oracle guidance templates. We build our oracle guidance based on these statistics.

Table A3 provides some examples of the oracle guidance. The general oracle guidance contain that we use for accuracy and efficiency experiments. The task-specific oracle guidance contain that we we use for ablation experiments.

### D.3 DETAILS OF EXPERIMENTAL ENVIRONMENT AND BASELINE

We use PyTorch 2.3.1 as our primary experimental platform. Our implementation is based on the NVIDIA KVPress repository (NVIDIA, 2025), which also serves as the codebase for most baseline methods.

All experiments are conducted using a server equipped with an AMD EPYC 7742 64-Core Processor, 256 GB of CPU memory, and four GPUs: three NVIDIA L40 GPUs for accuracy evaluations and ablation studies, and one NVIDIA A100 GPU for efficiency experiments.

For H2O(Zhang et al., 2023), since the original algorithm is not designed to perform KV cache eviction during the prefilling stage, we modify it to begin eviction only after prefilling is complete. For DuoAttention(Xiao et al., 2024a), whose effective compression ratio varies dynamically with input length, we adaptively set the head compression ratio per input to maintain a fixed KV cache budget. Additionally, to ensure a fair comparison, we disable all on-the-fly decoding mechanisms across all baselines.

## E SUPPLEMENTARY ACCURACY EVALUATION RESULTS

### E.1 SUPPLEMENTARY RESULTS FOR LONGBENCH

We present the detailed results on LongBench (Bai et al., 2024) in Table A12, Table A13, and Table A14. These include extended evaluations of Qwen2.5-7B-Instruct-1M on LongBench, as well as two additional code datasets that are not reported in the main paper.

We observe that LLaMA-3.1-8B-Instruct is generally more robust than both Mistral-7B-Instruct-v0.2 and Qwen2.5-7B-Instruct-1M across tasks. Notably, the performance of KV cache compression methods on multi-document QA datasets—such as HotpotQA and MuSiQue (Table A12)—tends to be unstable. In some cases, the compressed models actually outperform the full model. This is especially pronounced with Qwen2.5-7B-Instruct-1M, where several KV compression methods exceed the full-model performance on both single-document QA (e.g., NarrativeQA) and multi-document QA (e.g., MuSiQue).

Interestingly, OracleKV leads to slight degradation on some multi-document QA datasets (e.g., 2WikiMQA) compared to head-level compression methods such as DuoAttention(Xiao et al., 2024a) and Ada SnapKV(Feng et al., 2024), under the 40% KV cache budget. However, under the extreme condition of a 10% KV cache budget, OracleKV consistently outperforms all other methods across nearly all single- and multi-document QA datasets, highlighting its strong adaptability in low-memory scenarios.

As shown in Table A13, most KV cache compression methods lead to performance improvements on few-shot learning and in-context learning tasks across all three models—particularly on the

Table A4: Performance results on subsets of Longbench (Bai et al., 2024), keep ratio=0.1, using LLama-3.1-8B-Instruct.

| Method | NrtvQA | Qasper | MF-En | HotPotQA | GovReport | MultiNews | TREC | TriviaQA | PRe |
|--------|--------|--------|-------|----------|-----------|-----------|------|----------|-----|
| CAKE | 27.0 | 35.9 | 43.0 | 57.0 | 31.2 | **25.5** | **35.5** | 85.2 | 99.5 |
| OmniKV | 23.4 | 30.8 | 29.7 | 47.4 | 30.9 | 25.2 | 31.9 | **91.5** | 99.5 |
| OracleKV | **27.8** | **42.3** | **51.3** | **58.2** | **32.4** | 25.4 | 35.0 | 85.6 | 99.5 |

TREC dataset. This suggests that longer contexts may degrade few-shot performance. Additionally, streaming-based methods such as StreamingLLM(Xiao et al., 2024b) and DuoAttention(Xiao et al., 2024a) show a notable advantage on TriviaQA. Again, OracleKV demonstrates a superior accuracy–memory trade-off on both summarization and few-shot learning tasks, particularly under the 10

For passage count and passage retrieval tasks, as shown in Table A14, OracleKV delivers significant performance gains on LLaMA-3.1-8B-Instruct under both 10% and 40% KV cache budgets. For Mistral-7B-Instruct-v0.2, OracleKV outperforms all baselines on passage retrieval and achieves competitive performance with head-level methods on passage count at the 40% budget. On Qwen2.5-7B-Instruct-1M, OracleKV surpasses most methods at 40% for both passage count and retrieval tasks, and outperforms all baselines under the 10% KV cache budget for both tasks.

Table A14 highlights the side effect of OracleKV on code generation tasks, specifically on LCC and RepoBench-P. Notably, OracleKV introduces significant performance degradation on these tasks, with the most pronounced drop observed on the LCC dataset.

We hypothesize that this degradation stems from the oracle-guided attention redistribution, which may interfere with the inherent structural and syntactic regularity of code. Unlike natural language, code relies heavily on precise token dependencies and hierarchical structures. The intervention of OracleKV, though beneficial for semantic understanding tasks, may distort these structural patterns, leading to suboptimal generation quality in code-oriented scenarios.

### E.2 SUPPLEMENTARY RESULTS FOR RULER

We present the extended results on RULER(Hsieh et al., 2024) in TableA15, Table A16, Table A17, and Table A18. These tables report performance across 8 tasks from 13 datasets, evaluated under varying KV cache budgets (from 100% down to 10%) using two models: LLaMA-3.1-8B-Instruct and Qwen2.5-7B-Instruct-1M.

As shown in Table A15 and Table A16, OracleKV achieves a favorable accuracy–memory trade-off on the single-key Needle-in-a-Haystack (NIAH) task, demonstrating strong retrieval capabilities on both models. However, for multi-key NIAH, OracleKV shows noticeable performance degradation when compared to Ada SnapKV(Feng et al., 2024) and StreamingLLM(Xiao et al., 2024b). That said, the performance gap narrows as the KV cache budget decreases, indicating OracleKV's stronger adaptability under constrained memory conditions.

Table A16 and Table A18 further demonstrate OracleKV's strength in multi-value and multi-query variants of the NIAH task, as well as in variable tracing tasks. In these settings, OracleKV significantly outperforms all baselines across various KV cache budgets. Notably, under a 10% KV cache budget, OracleKV achieves average scores of 72.7 on LLaMA and 34.1 on Qwen, far surpassing the best-performing baselines (averaging 22.0 on LLaMA and 15.1 on Qwen). These results underscore OracleKV's effectiveness in complex retrieval and memory-intensive tasks, even under extreme memory constraints.

However, OracleKV also exhibits certain side effects on the RULER benchmark. Specifically, in word extraction (e.g., identifying the most frequently occurring words) and QA-style tasks, OracleKV underperforms relative to baseline methods, as shown in Table A16 and Table A18. This degradation suggests that OracleKV struggles with counting-oriented tasks or frequency-based reasoning. A possible explanation is that the general-purpose oracle guidance used by OracleKV does not effectively capture the inductive biases required for such tasks—biases that are rarely emphasized in large scale dialogues.

Table A5: Throughput results of decoding, measured in token per second (token/s), batch size=1, cache budget=1024, using Llama-3.1-8B-Instruct.

| Method | 2K | 4K | 8K | 16K | 32K |
|--------|------|------|------|------|------|
| Original | 34.90 | 31.37 | 22.22 | 14.00 | 8.02 |
| SnapKV | 35.41 | 35.30 | 34.97 | 34.63 | **35.37** |
| PyramidKV | 35.37 | 35.16 | 35.21 | 34.84 | 34.83 |
| OracleKV | 35.12 | **35.31** | **35.22** | **35.00** | 34.18 |

Table A6: KV Cache memory footprint results of decoding, measured in gigabyte(GB), batch size=1, cache budget=1024, using Llama-3.1-8B-Instruct.

| Method | 2K | 4K | 8K | 16K | 32K |
|--------|------|------|------|------|------|
| Original | 15.46 | 15.92 | 16.87 | 18.73 | 22.46 |
| SnapKV | 15.33 | 15.55 | 16.00 | 16.89 | 18.66 |
| PyramidKV | 15.33 | 15.55 | 16.00 | 16.89 | 18.66 |
| OracleKV | 15.33 | 15.55 | 16.00 | 16.89 | 18.66 |

Table A7: Results for guidance length latency-memory tradeoff, KV budget=1K, using Llama-3.1-8B-Instruct.

| Exp.Settings | KV.Mem.(GB) | Pref.Latency(s) | I/O Latency(s) | Dec.Latency(s) | Infer.w/o.Loading(s) | Infer.w/Loading(s) |
|--------------|-------------|-----------------|----------------|----------------|----------------------|--------------------|
| Original,64k,bs=1 | 7.83 | 11.75 | 1.31~7.83 | 17.77 | 29.52 | 19.08~25.60 |
| Original,32k,bs=2 | 7.84 | 4.14 | 1.31~7.84 | 8.40 | 12.54 | 9.71~16.24 |
| Original,16k,bs=4 | 7.87 | 1.72 | 1.31~7.87 | 4.90 | 6.62 | 6.21~12.77 |
| OralceKV,64k,bs=1,L=128 | 0.49 | 11.87 | 0.08~0.49 | 14.76 | 26.63 | 14.84~15.25 |
| OralceKV,32k,bs=2,L=128 | 0.49 | 4.41 | 0.08~0.49 | 7.30 | 11.71 | 7.38~7.79 |
| OralceKV,16k,bs=4,L=128 | 0.49 | 1.88 | 0.08~0.49 | 4.71 | 6.59 | 4.79~5.20 |
| OralceKV,64k,bs=1,L=256 | 0.49 | 12.32 | 0.08~0.49 | 15.20 | 27.52 | 15.28~15.69 |
| OralceKV, 32k, bs=2, L=256 | 0.49 | 4.63 | 0.08~0.49 | 7.51 | 12.14 | 7.59~8.00 |
| OralceKV, 16k, bs=4, L=256 | 0.49 | 2.01 | 0.08~0.49 | 4.83 | 6.84 | 4.91~5.32 |
| OralceKV, 64k, bs=1, L=512 | 0.49 | 13.98 | 0.08~0.49 | 16.31 | 30.29 | 16.39~16.80 |
| OralceKV, 32k, bs=2, L=512 | 0.49 | 5.20 | 0.08~0.49 | 8.08 | 13.28 | 8.16~8.57 |
| OralceKV, 16k, bs=4, L=512 | 0.49 | 2.30 | 0.08~0.49 | 5.14 | 7.44 | 5.22~5.63 |

### E.3 ADDITIONAL COMPARISON WITH MORE ADVANCED METHODS

We compare OracleKV with recent advanced methods (Qin et al., 2025; Hao et al., 2025) on subsets of Longbench in Table A4. OracleKV outperforms CAKE and OmniKV in question-independent scenarios.

## F SUPPLEMENTARY EFFICIENCY RESULTS

### F.1 DECODING LATENCY

During the decoding stage, questions are typically provided by users, rendering question-independent algorithms less applicable. However, to further investigate the applicability of OracleKV under such conditions, we conducted additional experiments to evaluate OracleKV's memory footprint and latency during the decoding phase, particularly for challenging edge cases. As shown in Table A5 and Table A6, OracleKV's decoding memory footprint and latency remain comparable (cache memory costs are identical since the budget is fixed) to SnapKV and PyramidKV across varying context lengths. This is primarily because the main computational overhead in OracleKV during decoding stems from the calculation of the window attention matrix.

### F.2 EFFICIENCY-ACCURACY TRADEOFF WITH GUIDANCE LENGTH

To highlight critical trade-off between memory savings and computational/latency overhead in real-time systems, we conducted experiments across varying guidance lengths in two settings:

- Decoding without loading CPU KV cache: re-prefilling and decoding;
- Decoding with loading CPU KV cache: loading KV cache with CPU-GPU I/O and decoding.

We measured KV cache memory usage, decoding throughput, prefilling latency, and I/O latency, calculated based on typical consumer-grade server sequential read speeds (1GB/s for NVMe SSDs with PCIe 3.0 x4 to 6GB/s for PCIe 4.0 x4). Total latency for generating 100 tokens was computed as: the w/o loading setting, and I/O latency + decoding latency for the loading setting. Results are reported in Table A7. OracleKV outperforms the full-cache method in systems with KV cache

Table A8: Results for guidance length latency-memory tradeoff on a subset of Longbench (Bai et al., 2024), KV budget=1K, using Llama-3.1-8B-Instruct.

| Length | NrtvQA | Qasper | MF-En | HotPotQA | GovReport | MultiNews | TREC | TriviaQA | PRe |
|---|---|---|---|---|---|---|---|---|---|
| 73 | 31.2 | 44.6 | 49.3 | 56.1 | 32.8 | 25.1 | 38.0 | 85.6 | 99.5 |
| 122 | 28.2 | 40.8 | 49.9 | 56.6 | 32.7 | 25.3 | 33.0 | 84.6 | 99.5 |
| 231 | 29.1 | 39.1 | 49.1 | 55.5 | 32.2 | 24.3 | 32.5 | 79.3 | 99.5 |

Table A9: Question-dependent performance of OracleKV on RULER (Hsieh et al., 2024) benchmark. We compare OracleKV with SnapKV using general surface-level oracle guidance.

| Method | Budget | NIAH-S | NIAH-M | MV | MQ | VT | CWE | FWE | QA | Avg |
|---|---|---|---|---|---|---|---|---|---|---|
| SnapKV | 50% | 95.67 | **98.80** | 99.60 | **99.95** | 99.88 | **98.04** | 91.33 | **75.00** | 94.78 |
| OracleKV | 50% | **100.00** | 97.60 | **99.70** | 99.90 | **99.92** | 97.28 | **93.00** | 74.70 | **95.26** |
| SnapKV | 40% | 88.80 | **94.87** | 99.45 | **99.90** | 99.88 | **96.06** | 90.00 | **74.90** | 92.98 |
| OracleKV | 40% | **98.93** | 91.53 | **99.70** | **99.90** | **99.92** | 94.88 | **92.00** | **74.90** | **93.97** |
| SnapKV | 30% | 78.40 | **84.93** | 99.50 | **99.85** | 99.88 | 88.46 | 87.07 | **74.80** | 89.11 |
| OracleKV | 30% | **94.47** | 80.67 | **99.65** | **99.85** | 99.84 | **89.14** | **89.40** | **74.80** | **90.98** |
| SnapKV | 20% | 68.67 | **72.73** | 98.80 | **99.80** | 98.96 | 72.60 | 81.80 | 74.10 | 83.43 |
| OracleKV | 20% | **79.80** | 69.47 | **99.50** | **99.80** | **99.64** | **74.52** | **84.40** | **74.70** | **85.23** |
| SnapKV | 10% | 67.13 | **65.87** | 77.70 | 99.20 | 94.92 | 40.62 | 64.73 | 73.40 | 72.95 |
| OracleKV | 10% | **67.93** | **65.87** | **99.15** | **99.30** | **97.12** | **41.82** | **74.80** | **74.00** | **77.50** |

loading when guidance length is $\leq 512$. In systems with prefilling, OracleKV achieves significant speedup only when guidance length is $\leq 128$. For guidance length = 256, speedup is observed for longer contexts (>32K), with greater benefits as context length increases. However, at guidance length = 512, negative performance is observed in prefilling systems for contexts between 16K and 64K, as longer guidance requires computing and explicitly returning the attention matrix during prefilling, which cannot leverage hardware acceleration (e.g., FlashAttention (Dao et al., 2022; Dao, 2024)), incurring significant overhead. We also explore the accuracy with varying guidance length. Table A8 shows that longer guidance length(>128) generally underperforms, suggesting a practical heuristic: guidance length should be kept $\leq 128$ for better performance.

# G  SUPPLEMENTARY ABLATION RESULTS

## G.1  QUESTION-DEPENDENT KV CACHE EVICTION PERFORMANCE

We explore the design choices of OracleKV under a question-dependent setting. As shown in Table A9, OracleKV yields small but consistent improvements over question-independent KV cache eviction when guided only by surface-level oracle guidance. This suggests that OracleKV can introduce a useful inductive bias, even when the question is already known. Additionally, as the KV cache budget decrease, OracleKV can enhance the average performance significantly (77.50 *vs.* 72.95 under 10% KV budget)

## G.2  LLM AS ORACLE GUIDANCE

In this section, we explore the automation of oracle guidance in multi-task pipeline. In real-world deployment scenarios, one viable approach is to predefine guidance templates for each domain and train a classifier to map user inputs to the corresponding domain. Upon receiving a user request, the system can automatically select and apply the appropriate domain-specific guidance.

To validate the feasibility of this concept, we conducted preliminary experiments **using LLM itself** to predict the types of questions users might ask based on the provided context and to retain key information. Specifically, we evicted KV cache with guidance: Based on the context above, please predict the questions that the user might ask about the above context. Please remember the most

Table A10: Performance results on subsets of Longbench (Bai et al., 2024), keep ratio=0.1, using Mistral-7B-Instruct-v0.2 and LLama-3.1-8B-Instruct.

| Method | NrtvQA | Qasper | MF-En | HotPotQA | GovReport | MultiNews | TREC | TriviaQA | PRe |
|---|---|---|---|---|---|---|---|---|---|
| *Mistral-7B-Instruct-v0.2* | | | | | | | | | |
| Original | 20.8 | 29.3 | 46.0 | 35.1 | 32.1 | 26.81 | 50.8 | 76.1 | 74.2 |
| SnapKV | 12.8 | 9.1 | 23.5 | 20.1 | 24.4 | 20.9 | 31.8 | **78.8** | 42.8 |
| Statistics-as-Guidance | **13.7** | **13.0** | **26.0** | **20.8** | 26.1 | **22.5** | **32.8** | 77.5 | 49.9 |
| LLM-as-Guidance | 13.4 | 10.9 | 23.5 | 20.2 | **26.4** | 21.8 | 31.0 | 77.7 | **54.3** |
| *Llama-3.1-8B-Instruct* | | | | | | | | | |
| Original | 29.7 | 47.6 | 55.7 | 58.8 | 35.5 | 27.2 | 28.0 | 86.2 | 100.00 |
| SnapKV | 24.2 | **21.1** | 26.9 | 46.4 | 25.6 | 20.5 | 33.0 | 82.9 | 56.0 |
| Statistics-as-Guidance | **26.8** | 20.2 | 26.6 | **48.3** | 27.6 | 21.5 | **45.5** | **84.1** | 79.5 |
| LLM-as-Guidance | 23.6 | 17.7 | **28.2** | 47.6 | **27.9** | 21.5 | 39.5 | 84.0 | **84.0** |

Table A11: Performance results on subsets of Longbench (Bai et al., 2024), cache budget=0.1, using Llama-3.1-8B-Instruct.

| Method | NrtvQA | Qasper | MF-En | HotPotQA | GovReport | MultiNews | TREC | TriviaQA | PRe |
|---|---|---|---|---|---|---|---|---|---|
| PyramidKV | 26.9 | 33.2 | 42.1 | 47.2 | 29.7 | 24.9 | 49.0 | 86.3 | 93.8 |
| Pyramid OracleKV | 27.2 | 33.5 | 47.6 | 55.0 | 29.9 | 23.6 | 44.0 | 85.3 | 99.5 |
| OracleKV | 29.1 | 42.3 | 51.3 | 58.2 | 32.8 | 25.4 | 35.0 | 86.3 | 99.5 |

important and relevant information and answer the question. We refer to this preliminary approach as LLM-as-Guidance.

As shown in the Table A10, even without fine-tuning or incorporating prior information, LLM-as-Guidance achieves competitive performance across various tasks (best performance on some tasks) compared to baseline methods. The results demonstrate the potential of leveraging learned models (e.g. LLMs) to predict future user queries based on context, thereby automating the generation of oracle guidance. We believe optimizing automated guidance strategies is a interesting direction for future research.

### G.3 FUTHER INTEGRATION ANALYSIS WITH OTHER LAYER/HEAD-LEVEL METHODS

Current layer/head-level methods (e.g., PyramidKV, AdaKV, CAKE) use heuristics for budget allocation, relying on SnapKV's observation window algorithm (Li et al., 2024b). OracleKV, however, selects entries based on oracle guidance attention scores, enabling seamless integration with these methods. Table 1 shows that integrating OracleKV with AdaKV improves performance on most tasks. To further enrich the integration analysis, we conducted additional experiments integrating OracleKV with PyramidKV, with results in the Table A11. As shown, integrating with PyramidKV slightly degrades OracleKV's performance on most tasks but still outperforms PyramidKV alone. This further demonstrates the superiority of guidance-based selection over observation window-based selection in question-independent scenarios.

### G.4 MORE DISCUSSION ABOUT RETAINING *vs.* EVICTING GUIDANCE

In Section 5.3, we conduct experiments and conclude that retaining the guidance cache degrades performance in most cases, especially with low KV cache budgets, as it consumes space without contributing relevant information, leading to the eviction of critical tokens. This affects tasks like summarization and QA.

Specifically, evicting the guidance cache is generally more effective, but in few-shot learning tasks on LongBench, particularly TREC, we observe that retaining it outperforms eviction,

Figure A12: Results on Longbench few-shot learning dataset TREC.

| Exp.Setting | Accuracy.TREC |
|---|---|
| Original | 28.0 |
| SnapKV | 32.5 |
| OracleKV w/ Evicting Guidance | 34.5 |
| OracleKV w/ Retaining Guidance | 67.5 |

showing a significant performance gap over OracleKV and other baselines, with KV compression

methods also surpassing full cache. Results in few-shot learning datasets TREC are shown in Table A12.

## H  SUPPLEMENTARY ANALYSIS

### H.1  VISUALIZATION OF ATTENTION DISTRIBUTION

Our main paper (Section 4) highlights the mismatch in attention distributions induced by the question and the pure context, which leads to discrepancies in the indices of retained tokens. In this section, we further investigate the differences in attention distribution and token retention across **layers** and **tasks**.

**Layer-wise visualization.** We observe a clear layer-wise pattern in the attention distributions. As shown in Figures A1–A14, the distributions induced by the question and pure context exhibit substantial overlap in the first three layers (Layers 0, 1, and 2) across various tasks. Notably, the attention distribution in the first layer appears relatively stable, and its attention scores are significantly higher than those in other layers.

**Task-wise visualization.** We also observe clear task-specific patterns in the attention distributions. Figures A1–A14 demonstrate that the overlap between attention scores induced by the question and pure context are higher in tasks such as word extraction, variable tracing, and question answering, compared to NIAH tasks.

### H.2  SIDE EFFECTS OF ORACLEKV

We observe side effects of OracleKV on certain datasets across benchmarks (e.g., code generation in LongBench (Bai et al., 2024), and word extraction and QA in RULER (Hsieh et al., 2024)). However, consistent with the no-free-lunch principle, OracleKV generally performs well across a majority of tasks, serving as an effective approach for KV cache eviction.

## I  MORE RELATED WORK

### I.1  KV CACHE EVICTION

Previous research has highlighted the inherent sparsity in the self-attention mechanisms of large language models (LLMs). Leveraging this property, early studies (Liu et al., 2024b; Zhang et al., 2023) propose maintaining a queue with a pre-allocated budget and progressively evicting unimportant cache entries during the inference. Subsequent works focus on exploiting fixed attention patterns within the input sequence. StreamingLLM (Xiao et al., 2024b) and LM-Infinite (Han et al., 2024) utilize the *attention sink* phenomenon to retain both initial and most recent tokens. Recently, SnapKV (Li et al., 2024b) introduces an attention-based strategy that uses attention scores with recent tokens to estimate importance. Building on this foundation, several approaches (Hao et al., 2025; Qin et al., 2025; Cai et al., 2024; Yang et al., 2024c) incorporate layer-wise cache budget allocation. PyramidKV (Cai et al., 2024) and PyramidInfer (Yang et al., 2024c) discard more KV entries from deeper layers, motivated by the *pyramidal information funneling* hypothesis. Similarly, CAKE (Qin et al., 2025) analyzes layer-wise preferences using spatial and temporal attention dynamics to optimize cache retention. In parallel, the discovery of *retrieval heads* in attention mechanisms (Wu et al., 2024) fuel a new line of research in head-level cache eviction (Fu et al., 2024; Xiao et al., 2024a; Feng et al., 2024). DuoAttention (Xiao et al., 2024a) employs a learning-based method to identify *compression-insensitive* attention heads (*i.e.*, streaming heads), while HeadKV (Fu et al., 2024) classifies heads based on their retrieval and reasoning utility (R2 heads).

Most recent research has introduced a variety of strategies for managing the Key-Value (KV) cache in large language models (LLMs), focusing on eviction and compression techniques to enhance memory efficiency without compromising performance. Eviction methods like NaCl (Chen et al., 2024) combine attention-based statistics with randomized strategies to retain crucial tokens, achieving significant cache reduction while maintaining high performance. HashEvict (Liu et al., 2024a) employs locality-sensitive hashing to identify and replace tokens with low relevance, reducing computational

overhead. Compression approaches have also evolved; GEAR (Kang et al., 2024) integrates ultra-low precision quantization, low-rank approximation, and sparse matrices to achieve near-lossless 4-bit compression, enhancing throughput. RazorAttention (Tang et al., 2024) differentiates between retrieval and non-retrieval heads, maintaining full cache for the former while discarding distant tokens for the latter. FastGen (Jacobs et al., 2023) introduces a plug-and-play adaptive compression method that profiles attention modules to selectively retain or discard tokens based on their contextual importance, significantly reducing GPU memory usage. Additionally, methods like BalanceKV (Han et al., 2025) utilize vector balancing theory for geometric sampling, and LoRC (Zhang et al., 2024) applies low-rank approximations with progressive compression strategies. These methodologies collectively advance the efficiency of LLM inference by intelligently managing KV cache resources.

Despite these approaches achieve impressive performance on several long-context benchmarks, most existing methods (*e.g.* (Li et al., 2024b; Cai et al., 2024; Yang et al., 2024c; Qin et al., 2025; Feng et al., 2024; Fu et al., 2024; Hao et al., 2025)) rely heavily on importance metrics derived from the attention scores with the current question, limiting their robustness and applicability in real-world scenarios without question. In contrast, our approach operates at the data level, leveraging surface-level statistical regularities in the question distribution, making it compatible with existing methods and easily integrable into a broader range of applications.

## I.2    IN-CONTEXT LEARNING

Early studies (Devlin et al., 2019; Liu et al., 2019) observed that language models can "learn" to perform a task from a few shot input-output examples provided in context at inference. (Xie et al., 2021) interprets the emergence of in-context learning by inferring the shared latent concept among demonstration examples. Based on these, OracleKV affects the attention behavior through in-context data manipulation, aiming to select instruction-correlated tokens.

Recently, (Bai et al., 2023) provided theoretical evidence that transformers can implement a broad class of machine learning algorithms in-context, including least squares and Lasso, and can adaptively select among them based on input sequences. Further empirical analysis (Kossen et al., 2023) revealed that ICL predictions are heavily influenced by in-context labels and that models can learn novel tasks in-context, although they may retain biases from pre-training data. For the algorithmic reasoning, (Zhou et al., 2022) introduced *algorithmic prompting*, teaching LLMs to perform multi-step reasoning tasks by formulating algorithms as composable skills, leading to significant performance improvements. Additionally, (Kirsch et al., 2022) explored meta-learning approaches, showing that transformers can be trained to act as general-purpose in-context learners, capable of adapting to diverse tasks without explicit training loss definitions.

Table A12: Detailed results of LongBench Bai et al. (2024), including Single-Document QA datasets(NartvQA, Qasper, and MF-en) and Multi-Documents QA datasets(HotpotQA, 2WikiMQA, and Musique).

| Method | Budget | NartvQA | Qasper | MF-en | HotpotQA | 2WikiMQA | Musique |
|---|---|---|---|---|---|---|---|
| *LLaMA-3.1-8B-Instruct* | | | | | | | |
| Full Cache | 100% | 31.99 | 48.20 | 57.18 | 57.56 | 48.92 | 28.19 |
| StreamingLLM | | 23.44 | 30.69 | 29.64 | 47.29 | 33.94 | 22.43 |
| SnapKV | | 27.04 | 35.66 | 43.22 | 56.79 | 47.50 | 30.06 |
| Ada SnapKV | | 27.28 | 39.60 | 44.25 | 56.11 | 50.05 | 30.66 |
| PyramidKV | 40% | 26.85 | 33.16 | 42.14 | 47.17 | 39.22 | 19.60 |
| DuoAttention | | 28.55 | 41.08 | 50.68 | 54.80 | 48.91 | 29.24 |
| OracleKV | | 29.10 | 42.33 | 51.34 | 58.16 | 48.51 | 32.61 |
| Ada OracleKV | | 28.51 | 44.44 | 53.04 | 57.47 | 47.41 | 31.47 |
| StreamingLLM | | 20.59 | 18.44 | 22.95 | 37.24 | 22.40 | 14.01 |
| SnapKV | | 23.64 | 20.77 | 23.29 | 44.62 | 24.22 | 19.87 |
| Ada SnapKV | | 24.24 | 21.14 | 26.88 | 46.35 | 26.95 | 18.80 |
| PyramidKV | 10% | 19.63 | 21.45 | 22.89 | 34.25 | 24.28 | 13.36 |
| DuoAttention | | 15.13 | 14.21 | 25.22 | 35.02 | 22.31 | 12.59 |
| OracleKV | | 26.77 | 20.18 | 26.59 | 48.31 | 29.19 | 21.95 |
| Ada OracleKV | | 28.12 | 23.86 | 34.48 | 50.00 | 29.40 | 22.50 |
| *Mistral-7B-Instruct-v0.2* | | | | | | | |
| Full Cache | 100% | 20.84 | 29.34 | 45.99 | 35.11 | 20.73 | 16.95 |
| StreamingLLM | | 13.75 | 17.09 | 27.75 | 26.87 | 17.31 | 11.44 |
| SnapKV | | 15.31 | 19.71 | 36.59 | 28.89 | 15.79 | 13.39 |
| Ada SnapKV | | 17.29 | 19.07 | 36.39 | 30.74 | 16.16 | 13.24 |
| PyramidKV | 40% | 15.86 | 19.68 | 36.27 | 30.84 | 18.72 | 12.70 |
| DuoAttention | | 11.51 | 9.09 | 25.27 | 21.51 | 15.03 | 6.76 |
| OracleKV | | 18.61 | 21.21 | 36.24 | 29.84 | 18.71 | 14.70 |
| Ada OracleKV | | 19.13 | 22.04 | 37.94 | 31.29 | 17.88 | 14.81 |
| StreamingLLM | | 10.01 | 10.55 | 21.02 | 17.82 | 12.33 | 7.41 |
| SnapKV | | 12.75 | 9.08 | 23.54 | 20.06 | 12.20 | 8.12 |
| Ada SnapKV | | 14.10 | 9.98 | 23.49 | 21.08 | 13.06 | 8.91 |
| PyramidKV | 10% | 10.43 | 9.08 | 24.38 | 21.26 | 12.64 | 7.28 |
| DuoAttention | | 9.01 | 7.98 | 24.96 | 19.68 | 13.56 | 6.01 |
| OracleKV | | 13.70 | 13.01 | 25.96 | 20.79 | 13.06 | 7.39 |
| Ada OracleKV | | 13.85 | 13.65 | 27.49 | 21.31 | 13.36 | 7.59 |
| *Qwen2.5-7B-Instruct-1M* | | | | | | | |
| Full Cache | 100% | 20.23 | 49.72 | 52.53 | 62.91 | 56.35 | 33.74 |
| StreamingLLM | | 17.37 | 31.62 | 28.82 | 44.61 | 42.68 | 24.66 |
| SnapKV | | 24.57 | 38.34 | 38.23 | 58.65 | 47.50 | 34.08 |
| Ada SnapKV | | 26.05 | 38.79 | 41.63 | 59.30 | 48.24 | 33.77 |
| PyramidKV | 40% | 17.16 | 27.13 | 29.79 | 47.10 | 37.66 | 23.27 |
| OracleKV | | 24.88 | 39.55 | 42.67 | 59.58 | 52.34 | 34.36 |
| Ada OracleKV | | 25.08 | 40.32 | 44.64 | 60.07 | 53.86 | 34.87 |
| StreamingLLM | | 13.20 | 18.51 | 21.32 | 32.83 | 32.40 | 13.54 |
| SnapKV | | 19.78 | 15.95 | 24.83 | 41.15 | 30.89 | 23.40 |
| Ada SnapKV | | 22.44 | 17.91 | 25.89 | 42.57 | 31.27 | 22.52 |
| PyramidKV | 10% | 16.93 | 16.18 | 24.69 | 36.70 | 31.01 | 15.76 |
| OracleKV | | 24.99 | 18.88 | 31.39 | 51.31 | 33.96 | 27.68 |
| Ada OracleKV | | 24.97 | 20.73 | 30.33 | 47.04 | 35.03 | 28.94 |

Table A13: Detailed results of LongBench Bai et al. (2024), including Summarization datasets(Gov Report, QMSum, Multi News) and Few-shot Learning datasets(TREC, Trivia QA, and SAMSum).

| Method | Budget | GovReport | QMSum | MultiNews | TREC | TriviaQA | SAMSum |
|---|---|---|---|---|---|---|---|
| *LLaMA-3.1-8B-Instruct* | | | | | | | |
| Full Cache | 100% | 35.49 | 25.06 | 27.15 | 28.00 | 86.21 | 39.16 |
| StreamingLLM | | 30.41 | 21.69 | 24.97 | 31.50 | 91.40 | 37.21 |
| SnapKV | | 30.96 | 23.37 | 25.07 | 34.50 | 85.13 | 41.04 |
| Ada SnapKV | | 30.98 | 23.47 | 25.64 | 38.00 | 86.38 | 41.05 |
| PyramidKV | 40% | 29.72 | 22.65 | 24.86 | 49.00 | 86.28 | 30.76 |
| DuoAttention | | 30.70 | 25.00 | 24.80 | 34.00 | 90.19 | 36.21 |
| OracleKV | | 32.78 | 24.11 | 25.39 | 35.00 | 86.28 | 40.57 |
| Ada OracleKV | | 32.85 | 24.40 | 25.64 | 33.50 | 87.21 | 39.49 |
| StreamingLLM | | 24.81 | 19.10 | 20.09 | 28.00 | 90.66 | 34.38 |
| SnapKV | | 25.21 | 20.04 | 19.91 | 34.00 | 82.21 | 38.64 |
| Ada SnapKV | | 25.57 | 20.87 | 20.52 | 33.00 | 82.88 | 39.18 |
| PyramidKV | 10% | 24.84 | 20.13 | 19.98 | 34.00 | 85.38 | 36.05 |
| DuoAttention | | 23.70 | 17.70 | 22.83 | 25.00 | 86.92 | 36.21 |
| OracleKV | | 27.61 | 21.28 | 21.45 | 45.50 | 84.11 | 39.31 |
| Ada OracleKV | | 27.90 | 21.97 | 21.44 | 49.50 | 85.59 | 39.67 |
| *Mistral-7B-Instruct-v0.2* | | | | | | | |
| Full Cache | 100% | 32.13 | 24.15 | 26.81 | 50.75 | 76.14 | 39.32 |
| StreamingLLM | | 30.23 | 21.34 | 24.54 | 49.50 | 52.78 | 36.48 |
| SnapKV | | 28.77 | 22.12 | 24.65 | 44.75 | 77.92 | 39.11 |
| Ada SnapKV | | 27.73 | 22.37 | 24.53 | 44.15 | 77.10 | 39.10 |
| PyramidKV | 40% | 28.43 | 22.21 | 24.55 | 45.75 | 79.23 | 39.12 |
| DuoAttention | | 23.79 | 20.35 | 23.04 | 24.62 | 72.36 | 35.08 |
| OracleKV | | 30.28 | 23.38 | 25.35 | 44.05 | 78.06 | 38.80 |
| Ada OracleKV | | 30.03 | 23.31 | 25.36 | 45.25 | 77.17 | 38.78 |
| StreamingLLM | | 24.72 | 20.25 | 19.00 | 34.50 | 45.69 | 35.49 |
| SnapKV | | 24.39 | 20.04 | 20.86 | 31.75 | 78.81 | 37.99 |
| Ada SnapKV | | 24.06 | 20.56 | 20.44 | 37.50 | 79.14 | 38.18 |
| PyramidKV | 10% | 23.97 | 20.05 | 20.64 | 31.75 | 78.93 | 36.82 |
| DuoAttention | | 20.89 | 18.38 | 21.87 | 22.62 | 66.06 | 33.69 |
| OracleKV | | 26.10 | 21.14 | 22.51 | 32.75 | 77.46 | 36.97 |
| Ada OracleKV | | 26.04 | 21.74 | 22.74 | 31.50 | 75.68 | 37.61 |
| *Qwen2.5-7B-Instruct-1M* | | | | | | | |
| Full Cache | 100% | 35.45 | 24.59 | 25.97 | 69.50 | 86.53 | 37.21 |
| StreamingLLM | | 32.35 | 20.91 | 24.03 | 59.00 | 48.12 | 24.61 |
| SnapKV | | 33.04 | 21.71 | 23.62 | 61.00 | 86.25 | 37.22 |
| Ada SnapKV | | 32.50 | 21.98 | 23.81 | 64.50 | 86.45 | 36.54 |
| PyramidKV | 40% | 29.46 | 20.10 | 23.16 | 49.50 | 82.40 | 39.06 |
| OracleKV | | 33.71 | 23.14 | 23.64 | 69.00 | 86.52 | 38.40 |
| Ada OracleKV | | 33.51 | 23.50 | 23.84 | 72.00 | 86.88 | 36.44 |
| StreamingLLM | | 26.66 | 19.25 | 18.46 | 47.00 | 40.82 | 20.98 |
| SnapKV | | 27.51 | 18.88 | 19.01 | 42.25 | 86.42 | 35.84 |
| Ada SnapKV | | 27.65 | 18.63 | 19.37 | 45.50 | 85.97 | 36.09 |
| PyramidKV | 10% | 26.40 | 18.45 | 19.36 | 42.50 | 85.16 | 35.31 |
| OracleKV | | 29.50 | 20.68 | 19.66 | 61.50 | 86.16 | 35.51 |
| Ada OracleKV | | 29.43 | 20.58 | 19.81 | 58.25 | 85.68 | 35.46 |

Table A14: Detailed results of LongBench Bai et al. (2024), including Sythetic datasets(Passage Count, Passage Retrieval) and Code Generation datasets(Lcc and RepoBench-P).

| Method | Budget | Passage Count | Passage Retrieval | Lcc | RepoBench-P |
|---|---|---|---|---|---|
| *LLaMA-3.1-8B-Instruct* | | | | | |
| Full Cache | 100% | 11.20 | 100.00 | 54.09 | 47.28 |
| StreamingLLM | | 6.70 | 45.50 | 50.63 | 49.28 |
| SnapKV | | 11.05 | 98.00 | 53.23 | 47.52 |
| Ada SnapKV | | 11.55 | 97.00 | 48.21 | 42.85 |
| PyramidKV | 40% | 9.22 | 93.75 | 56.77 | 56.93 |
| DuoAttention | | 6.00 | 99.50 | 55.09 | 53.09 |
| OracleKV | | 12.65 | 99.50 | 48.35 | 43.89 |
| Ada OracleKV | | 12.15 | 99.50 | 50.23 | 46.10 |
| StreamingLLM | | 4.00 | 16.00 | 52.29 | 52.88 |
| SnapKV | | 5.00 | 54.00 | 51.04 | 48.78 |
| Ada SnapKV | | 7.00 | 56.00 | 46.36 | 44.92 |
| PyramidKV | 10% | 7.50 | 46.75 | 51.40 | 52.26 |
| DuoAttention | | 6.00 | 50.00 | 55.09 | 53.09 |
| OracleKV | | 11.10 | 79.50 | 38.73 | 43.05 |
| Ada OracleKV | | 9.50 | 89.00 | 36.76 | 43.72 |
| *Mistral-7B-Instruct-v0.2* | | | | | |
| Full Cache | 100% | 2.81 | 74.17 | 51.25 | 50.74 |
| StreamingLLM | | 2.14 | 31.01 | 43.95 | 46.21 |
| SnapKV | | 3.37 | 72.40 | 48.54 | 48.49 |
| Ada SnapKV | | 3.23 | 67.87 | 48.41 | 48.35 |
| PyramidKV | 40% | 3.36 | 64.54 | 51.56 | 50.83 |
| DuoAttention | | 2.08 | 10.54 | 45.24 | 47.28 |
| OracleKV | | 3.41 | 71.02 | 46.66 | 47.14 |
| Ada OracleKV | | 2.63 | 72.93 | 45.93 | 48.21 |
| StreamingLLM | | 3.64 | 10.20 | 46.05 | 47.29 |
| SnapKV | | 3.16 | 42.77 | 49.31 | 50.06 |
| Ada SnapKV | | 3.26 | 50.02 | 47.24 | 48.35 |
| PyramidKV | 10% | 4.41 | 38.83 | 48.81 | 50.73 |
| DuoAttention | | 1.88 | 8.64 | 42.78 | 48.46 |
| OracleKV | | 3.87 | 49.93 | 37.99 | 47.92 |
| Ada OracleKV | | 3.97 | 61.20 | 38.44 | 47.28 |
| *Qwen2.5-7B-Instruct-1M* | | | | | |
| Full Cache | 100% | 8.50 | 99.00 | 63.14 | 59.08 |
| StreamingLLM | | 5.00 | 34.00 | 56.46 | 54.51 |
| SnapKV | | 8.50 | 98.00 | 62.13 | 57.71 |
| Ada SnapKV | | 9.00 | 99.00 | 59.83 | 55.30 |
| PyramidKV | 40% | 8.50 | 63.00 | 61.76 | 58.54 |
| OracleKV | | 9.50 | 98.00 | 55.64 | 56.65 |
| Ada OracleKV | | 8.00 | 98.00 | 54.31 | 56.25 |
| StreamingLLM | | 2.50 | 10.50 | 55.10 | 54.96 |
| SnapKV | | 4.50 | 40.50 | 57.98 | 56.39 |
| Ada SnapKV | | 4.50 | 36.50 | 55.61 | 53.42 |
| PyramidKV | 10% | 4.50 | 31.00 | 57.92 | 56.47 |
| OracleKV | | 9.00 | 71.00 | 43.89 | 55.60 |
| Ada OracleKV | | 8.00 | 75.00 | 42.59 | 54.68 |

Table A15: Detailed results of RULER (Hsieh et al., 2024) on Llama-3.1-8B-Instruct, including the NIAH-Single and NIAH-MultiKey, both of which consists of three datasets.

| Method | Budget | NIAH-Single | | | NIAH-MultiKey | | |
|---|---|---|---|---|---|---|---|
| | | S-1 | S-2 | S-3 | MK-1 | MK-2 | MK-3 |
| *LLaMA-3.1-8B-Instruct* | | | | | | | |
| Full Cache | 100% | 100.00 | 100.00 | 100.00 | 99.80 | 100.00 | 99.80 |
| H2O | 90% | 84.00 | 74.20 | 34.60 | 97.40 | 100.00 | 91.40 |
| StreamingLLM | | 92.60 | 91.60 | 91.40 | 89.00 | 93.60 | 92.20 |
| SnapKV | | 99.20 | 100.00 | 21.00 | 99.60 | 98.20 | 83.80 |
| PyramidKV | | 70.20 | 100.00 | 21.40 | 99.20 | 96.60 | 83.00 |
| Ada SnapKV | | 100.00 | 100.00 | 65.00 | 99.80 | 100.00 | 99.00 |
| OracleKV | | 100.00 | 100.00 | 100.00 | 99.80 | 91.20 | 76.20 |
| H2O | 75% | 76.60 | 47.80 | 18.00 | 86.60 | 94.00 | 48.60 |
| StreamingLLM | | 75.20 | 75.20 | 75.00 | 77.40 | 78.20 | 74.40 |
| SnapKV | | 99.60 | 99.80 | 11.20 | 99.00 | 84.20 | 55.20 |
| PyramidKV | | 71.00 | 100.00 | 11.20 | 98.60 | 84.20 | 51.00 |
| Ada SnapKV | | 99.80 | 100.00 | 18.80 | 99.80 | 99.20 | 91.60 |
| OracleKV | | 100.00 | 100.00 | 98.40 | 99.80 | 68.00 | 24.00 |
| H2O | 50% | 46.20 | 23.60 | 11.40 | 51.00 | 46.40 | 15.20 |
| StreamingLLM | | 50.20 | 50.00 | 55.40 | 55.60 | 49.80 | 48.80 |
| SnapKV | | 95.40 | 95.20 | 5.60 | 85.40 | 53.60 | 20.60 |
| PyramidKV | | 74.80 | 98.00 | 4.20 | 83.40 | 55.80 | 17.20 |
| Ada SnapKV | | 99.00 | 98.60 | 7.60 | 88.80 | 82.80 | 47.40 |
| OracleKV | | 100.00 | 100.00 | 49.80 | 99.80 | 26.60 | 2.00 |
| H2O | 40% | 34.00 | 16.20 | 8.80 | 33.60 | 28.00 | 9.40 |
| StreamingLLM | | 40.80 | 41.80 | 43.40 | 46.80 | 38.80 | 39.20 |
| SnapKV | | 90.60 | 83.00 | 3.80 | 68.40 | 39.80 | 9.60 |
| PyramidKV | | 72.60 | 92.20 | 3.60 | 71.00 | 41.40 | 10.20 |
| Ada SnapKV | | 97.80 | 90.40 | 4.60 | 71.60 | 67.00 | 31.20 |
| OracleKV | | 100.00 | 100.00 | 14.80 | 99.80 | 14.80 | 1.00 |
| H2O | 30% | 21.80 | 6.80 | 6.80 | 17.80 | 16.60 | 5.00 |
| StreamingLLM | | 30.20 | 34.00 | 36.60 | 39.20 | 29.20 | 27.60 |
| SnapKV | | 82.60 | 65.60 | 2.80 | 43.80 | 22.80 | 4.40 |
| PyramidKV | | 83.80 | 65.80 | 2.80 | 43.60 | 23.40 | 4.00 |
| Ada SnapKV | | 95.40 | 70.60 | 2.60 | 44.60 | 44.20 | 14.00 |
| OracleKV | | 100.00 | 100.00 | 3.00 | 99.80 | 7.40 | 0.00 |
| H2O | 20% | 13.40 | 2.00 | 4.00 | 7.60 | 8.80 | 2.20 |
| StreamingLLM | | 19.60 | 22.40 | 23.20 | 27.00 | 19.00 | 17.60 |
| SnapKV | | 73.40 | 41.40 | 2.40 | 27.20 | 12.00 | 1.00 |
| PyramidKV | | 72.80 | 41.80 | 2.40 | 27.40 | 12.20 | 1.00 |
| Ada SnapKV | | 91.80 | 38.80 | 2.40 | 24.80 | 21.60 | 3.60 |
| OracleKV | | 100.00 | 100.00 | 0.60 | 96.00 | 4.60 | 0.00 |
| H2O | 10% | 5.40 | 1.20 | 2.40 | 3.60 | 3.60 | 0.40 |
| StreamingLLM | | 10.40 | 15.40 | 15.60 | 17.60 | 9.00 | 7.00 |
| SnapKV | | 56.20 | 13.40 | 2.40 | 17.40 | 6.20 | 0.40 |
| PyramidKV | | 56.40 | 13.40 | 2.40 | 17.40 | 6.20 | 0.40 |
| Ada SnapKV | | 70.20 | 11.40 | 2.40 | 17.00 | 7.80 | 0.60 |
| OracleKV | | 100.00 | 100.00 | 0.00 | 77.80 | 3.00 | 0.00 |

Table A16: Detailed results of RULER (Hsieh et al., 2024) on LLama-3.1-8B-Instruct, including the NIAH-MultiValue(MV), NIAH-MultiQuery(MQ), Varaiable Tracing(VT), Common Words Extraction(CWE), Frequent Words Extraction(FWE), Question Answering(QA-1, QA-2).

| Method | Budget | MV | MQ | VT | Word Extraction | | QA | |
| | | | | | CWE | FWE | QA-1 | QA-2 |
|---|---|---|---|---|---|---|---|---|
| *LLaMA-3.1-8B-Instruct* | | | | | | | | |
| Full Cache | 100% | 99.90 | 99.90 | 99.88 | 99.62 | 94.80 | 87.80 | 62.80 |
| H2O | 90% | 96.15 | 96.55 | 98.68 | 99.64 | 94.60 | 87.60 | 61.60 |
| StreamingLLM | | 87.90 | 89.05 | 100.00 | 99.70 | 95.40 | 87.00 | 59.00 |
| SnapKV | | 98.80 | 99.60 | 97.80 | 99.56 | 94.80 | 87.20 | 61.40 |
| PyramidKV | | 99.70 | 99.20 | 97.96 | 99.78 | 94.40 | 80.60 | 54.40 |
| Ada SnapKV | | 99.90 | 99.85 | 99.92 | 99.70 | 94.67 | 87.20 | 62.40 |
| OracleKV | | 99.90 | 99.90 | 99.92 | 99.48 | 94.73 | 85.00 | 62.20 |
| H2O | 75% | 82.70 | 81.75 | 97.40 | 99.66 | 94.27 | 86.80 | 59.80 |
| StreamingLLM | | 75.35 | 76.20 | 94.68 | 99.62 | 94.07 | 87.00 | 55.00 |
| SnapKV | | 90.35 | 96.05 | 93.36 | 99.42 | 94.27 | 83.20 | 58.80 |
| PyramidKV | | 97.85 | 98.20 | 93.52 | 99.58 | 93.47 | 78.20 | 53.00 |
| Ada SnapKV | | 98.95 | 99.45 | 99.92 | 99.58 | 94.27 | 84.80 | 59.00 |
| OracleKV | | 99.85 | 99.90 | 99.92 | 99.00 | 93.47 | 78.60 | 60.00 |
| H2O | 50% | 43.85 | 43.55 | 91.12 | 99.66 | 93.87 | 86.80 | 56.20 |
| StreamingLLM | | 53.40 | 53.60 | 72.40 | 53.38 | 91.60 | 87.40 | 49.60 |
| SnapKV | | 72.65 | 75.75 | 82.16 | 98.38 | 92.33 | 75.60 | 52.00 |
| PyramidKV | | 76.80 | 75.85 | 80.68 | 90.26 | 88.93 | 67.20 | 43.80 |
| Ada SnapKV | | 78.60 | 85.60 | 96.68 | 99.28 | 94.73 | 77.20 | 52.80 |
| OracleKV | | 99.90 | 99.70 | 99.64 | 95.54 | 89.87 | 58.00 | 52.40 |
| H2O | 40% | 23.80 | 25.10 | 85.16 | 99.58 | 93.33 | 85.80 | 52.60 |
| StreamingLLM | | 43.00 | 43.25 | 62.28 | 26.86 | 91.93 | 87.40 | 47.00 |
| SnapKV | | 56.15 | 57.85 | 75.12 | 96.34 | 90.87 | 69.20 | 48.60 |
| PyramidKV | | 56.30 | 54.85 | 72.88 | 82.18 | 85.67 | 68.80 | 44.80 |
| Ada SnapKV | | 59.65 | 60.55 | 93.28 | 99.06 | 94.47 | 73.20 | 47.20 |
| OracleKV | | 98.35 | 99.60 | 98.76 | 91.50 | 87.40 | 49.80 | 47.20 |
| H2O | 30% | 9.90 | 11.45 | 73.84 | 98.70 | 92.13 | 85.00 | 47.60 |
| StreamingLLM | | 35.55 | 36.40 | 49.28 | 13.50 | 92.60 | 88.00 | 41.80 |
| SnapKV | | 32.00 | 34.40 | 65.32 | 90.80 | 87.47 | 60.60 | 40.80 |
| PyramidKV | | 31.55 | 34.75 | 65.56 | 90.22 | 87.47 | 60.60 | 40.60 |
| Ada SnapKV | | 28.20 | 29.60 | 86.64 | 97.78 | 93.33 | 64.60 | 43.60 |
| OracleKV | | 96.60 | 98.70 | 96.20 | 80.68 | 83.67 | 41.40 | 44.80 |
| H2O | 20% | 2.30 | 4.90 | 46.28 | 95.46 | 89.60 | 79.00 | 40.80 |
| StreamingLLM | | 22.85 | 23.25 | 35.96 | 1.60 | 92.80 | 88.60 | 36.40 |
| SnapKV | | 20.65 | 21.45 | 51.72 | 73.02 | 81.60 | 48.60 | 34.60 |
| PyramidKV | | 20.05 | 21.50 | 52.28 | 72.92 | 81.87 | 48.40 | 34.60 |
| Ada SnapKV | | 17.15 | 19.30 | 67.44 | 93.44 | 90.20 | 55.60 | 36.80 |
| OracleKV | | 90.55 | 95.05 | 90.68 | 60.62 | 78.27 | 31.20 | 36.40 |
| H2O | 10% | 0.30 | 2.45 | 18.28 | 78.28 | 77.93 | 68.40 | 32.00 |
| StreamingLLM | | 15.15 | 15.25 | 19.12 | 0.44 | 87.60 | 74.60 | 29.80 |
| SnapKV | | 15.05 | 16.05 | 30.44 | 16.08 | 67.80 | 32.20 | 24.80 |
| PyramidKV | | 15.00 | 16.20 | 30.68 | 16.40 | 67.67 | 32.00 | 25.00 |
| Ada SnapKV | | 14.90 | 16.25 | 34.60 | 44.54 | 78.87 | 35.60 | 27.20 |
| OracleKV | | 72.65 | 76.35 | 69.00 | 30.70 | 65.20 | 19.20 | 27.60 |

Table A17: Detailed results of RULER (Hsieh et al., 2024) on Qwen2.5-7B-Instruct-1M, including the NIAH-Single and NIAH-MultiKey, both of which consists of three datasets.

| Method | Budget | NIAH-Single | | | NIAH-MultiKey | | |
|---|---|---|---|---|---|---|---|
| | | S-1 | S-2 | S-3 | MK-1 | MK-2 | MK-3 |
| *Qwen2.5-7B-Instruct-1M* | | | | | | | |
| Full Cache | 100% | 100.00 | 99.00 | 99.80 | 100.00 | 99.80 | 99.40 |
| H2O | 90% | 39.00 | 98.00 | 13.80 | 86.80 | 80.40 | 84.00 |
| StreamingLLM | | 32.00 | 96.40 | 9.60 | 77.00 | 73.20 | 70.80 |
| SnapKV | | 92.00 | 86.40 | 88.60 | 89.00 | 92.40 | 89.60 |
| PyramidKV | | 86.20 | 98.40 | 40.60 | 99.20 | 99.00 | 92.60 |
| Ada SnapKV | | 18.00 | 9.60 | 0.20 | 25.40 | 30.40 | 1.20 |
| OracleKV | | 100.00 | 98.80 | 96.40 | 99.80 | 22.40 | 44.20 |
| H2O | 75% | 24.80 | 79.80 | 6.20 | 55.60 | 61.40 | 39.80 |
| StreamingLLM | | 17.60 | 59.40 | 3.40 | 37.00 | 44.60 | 23.20 |
| SnapKV | | 75.00 | 74.40 | 74.60 | 77.40 | 77.80 | 72.80 |
| PyramidKV | | 66.80 | 96.40 | 11.40 | 81.20 | 90.20 | 54.60 |
| Ada SnapKV | | 14.60 | 4.40 | 0.20 | 18.80 | 17.40 | 0.20 |
| OracleKV | | 100.00 | 99.00 | 70.40 | 97.80 | 3.60 | 12.60 |
| H2O | 50% | 15.00 | 38.40 | 2.80 | 25.20 | 29.40 | 8.80 |
| StreamingLLM | | 9.40 | 11.80 | 2.40 | 13.40 | 10.40 | 1.40 |
| SnapKV | | 49.40 | 49.40 | 55.20 | 55.20 | 49.60 | 48.60 |
| PyramidKV | | 31.00 | 58.00 | 4.00 | 37.00 | 45.20 | 12.60 |
| Ada SnapKV | | 10.20 | 1.80 | 0.00 | 8.00 | 6.20 | 0.00 |
| OracleKV | | 100.00 | 99.20 | 12.20 | 87.00 | 1.40 | 2.60 |
| H2O | 40% | 14.00 | 27.40 | 2.60 | 22.60 | 19.60 | 4.40 |
| StreamingLLM | | 8.40 | 6.60 | 2.40 | 12.00 | 7.00 | 0.60 |
| SnapKV | | 40.80 | 41.60 | 43.00 | 46.60 | 38.80 | 38.40 |
| PyramidKV | | 22.00 | 41.00 | 2.80 | 25.40 | 29.60 | 6.20 |
| Ada SnapKV | | 8.80 | 1.60 | 0.00 | 3.80 | 0.00 | 9.40 |
| OracleKV | | 100.00 | 99.80 | 5.80 | 79.20 | 1.20 | 1.40 |
| H2O | 30% | 11.80 | 18.40 | 2.40 | 17.20 | 13.80 | 1.60 |
| StreamingLLM | | 12.00 | 18.40 | 2.40 | 17.40 | 4.40 | 1.60 |
| SnapKV | | 30.20 | 33.60 | 36.20 | 39.00 | 29.00 | 26.40 |
| PyramidKV | | 15.60 | 23.00 | 2.60 | 19.40 | 18.40 | 4.60 |
| Ada SnapKV | | 6.00 | 1.20 | 0.00 | 4.60 | 2.60 | 0.00 |
| OracleKV | | 100.00 | 99.80 | 0.60 | 65.20 | 1.00 | 0.00 |
| H2O | 20% | 11.00 | 10.00 | 2.40 | 12.40 | 7.60 | 0.40 |
| StreamingLLM | | 10.80 | 10.00 | 2.40 | 12.40 | 7.60 | 0.40 |
| SnapKV | | 19.60 | 22.20 | 21.60 | 26.80 | 19.00 | 17.40 |
| PyramidKV | | 12.40 | 11.20 | 2.40 | 13.60 | 10.20 | 1.00 |
| Ada SnapKV | | 4.00 | 1.00 | 0.00 | 2.40 | 2.00 | 0.00 |
| OracleKV | | 100.00 | 99.00 | 0.40 | 46.60 | 1.00 | 0.00 |
| H2O | 10% | 8.40 | 3.20 | 2.40 | 10.40 | 2.40 | 0.00 |
| StreamingLLM | | 8.40 | 3.20 | 2.40 | 10.40 | 2.40 | 0.00 |
| SnapKV | | 10.40 | 15.40 | 13.00 | 16.20 | 9.00 | 7.00 |
| PyramidKV | | 7.60 | 2.80 | 2.40 | 10.20 | 3.60 | 0.00 |
| Ada SnapKV | | 1.40 | 0.60 | 0.00 | 1.00 | 0.60 | 0.00 |
| OracleKV | | 99.40 | 89.00 | 0.00 | 25.20 | 0.60 | 0.00 |

Table A18: Detailed results of RULER (Hsieh et al., 2024) on Qwen2.5-7B-Instruct-1M, including the NIAH-MultiValue(MV), NIAH-MultiQuery(MQ), Varaiable Tracing(VT), Common Words Extraction(CWE), Frequent Words Extraction(FWE), Question Answering(QA-1, QA-2).

| Method | Budget | MV | MQ | VT | Word Extraction | | QA | |
| | | | | | CWE | FWE | QA-1 | QA-2 |
|---|---|---|---|---|---|---|---|---|
| *Qwen2.5-7B-Instruct-1M* | | | | | | | | |
| Full Cache | 100% | 99.30 | 99.60 | 99.92 | 95.24 | 85.87 | 85.80 | 60.40 |
| H2O | | 64.65 | 80.75 | 86.32 | 95.40 | 85.93 | 85.60 | 58.60 |
| StreamingLLM | | 47.10 | 63.95 | 81.00 | 95.94 | 87.80 | 84.40 | 57.40 |
| SnapKV | 90% | 89.25 | 88.75 | 77.32 | 95.82 | 89.60 | 86.20 | 60.20 |
| PyramidKV | | 91.00 | 96.70 | 98.60 | 95.06 | 85.93 | 85.60 | 58.60 |
| Ada SnapKV | | 22.35 | 23.40 | 53.28 | 84.34 | 83.87 | 83.40 | 55.80 |
| OracleKV | | 98.80 | 99.50 | 99.92 | 95.18 | 84.60 | 84.40 | 58.80 |
| H2O | | 32.90 | 45.85 | 66.76 | 94.98 | 86.33 | 82.80 | 58.20 |
| StreamingLLM | | 20.40 | 27.10 | 57.96 | 95.20 | 87.20 | 80.20 | 52.80 |
| SnapKV | 75% | 75.75 | 75.95 | 78.36 | 94.12 | 88.60 | 86.20 | 55.20 |
| PyramidKV | | 53.60 | 71.00 | 90.92 | 95.16 | 86.20 | 84.20 | 57.40 |
| Ada SnapKV | | 14.80 | 16.20 | 49.84 | 83.28 | 83.73 | 82.20 | 53.80 |
| OracleKV | | 96.60 | 98.25 | 99.92 | 94.40 | 82.27 | 77.60 | 55.40 |
| H2O | | 17.90 | 21.60 | 43.16 | 92.42 | 85.33 | 73.00 | 48.00 |
| StreamingLLM | | 11.00 | 12.05 | 12.48 | 45.52 | 79.00 | 50.20 | 33.00 |
| SnapKV | 50% | 53.00 | 52.85 | 61.96 | 88.60 | 85.93 | 86.60 | 49.40 |
| PyramidKV | | 21.95 | 30.40 | 61.24 | 93.70 | 85.67 | 75.80 | 50.80 |
| Ada SnapKV | | 4.60 | 6.25 | 30.24 | 82.54 | 85.20 | 79.00 | 49.40 |
| OracleKV | | 78.00 | 84.50 | 99.84 | 89.42 | 78.00 | 61.40 | 50.80 |
| H2O | | 14.70 | 17.65 | 39.72 | 89.28 | 84.00 | 68.40 | 46.60 |
| StreamingLLM | | 10.05 | 10.80 | 10.32 | 42.86 | 77.33 | 64.20 | 36.00 |
| SnapKV | 40% | 42.70 | 43.10 | 52.12 | 87.62 | 85.93 | 87.40 | 47.00 |
| PyramidKV | | 17.50 | 23.70 | 46.72 | 92.54 | 85.60 | 69.40 | 48.00 |
| Ada SnapKV | | 2.70 | 4.50 | 21.56 | 81.34 | 84.07 | 79.20 | 46.20 |
| OracleKV | | 68.60 | 74.45 | 99.32 | 84.04 | 76.27 | 55.40 | 48.20 |
| H2O | | 12.45 | 14.95 | 30.68 | 84.44 | 82.27 | 59.00 | 43.80 |
| StreamingLLM | | 12.60 | 15.05 | 30.64 | 38.64 | 82.33 | 58.80 | 43.40 |
| SnapKV | 30% | 35.50 | 36.30 | 40.76 | 84.50 | 83.53 | 86.60 | 44.40 |
| PyramidKV | | 13.60 | 16.35 | 32.72 | 88.86 | 84.67 | 61.20 | 43.60 |
| Ada SnapKV | | 1.40 | 2.75 | 12.44 | 77.02 | 82.07 | 73.80 | 44.00 |
| OracleKV | | 57.85 | 61.45 | 97.68 | 74.80 | 75.00 | 45.60 | 42.80 |
| H2O | | 11.25 | 12.65 | 22.44 | 73.58 | 80.33 | 52.00 | 37.00 |
| StreamingLLM | | 11.25 | 12.60 | 22.08 | 73.80 | 80.40 | 52.20 | 37.20 |
| SnapKV | 20% | 22.65 | 23.10 | 29.04 | 80.10 | 83.87 | 87.60 | 37.20 |
| PyramidKV | | 12.05 | 13.80 | 20.84 | 81.44 | 82.13 | 53.80 | 37.60 |
| Ada SnapKV | | 0.70 | 1.85 | 9.04 | 69.70 | 77.60 | 72.40 | 40.20 |
| OracleKV | | 43.10 | 44.75 | 91.56 | 56.32 | 69.40 | 36.60 | 39.00 |
| H2O | | 9.60 | 9.90 | 11.80 | 54.48 | 72.73 | 35.20 | 25.80 |
| StreamingLLM | | 9.60 | 9.90 | 11.68 | 54.62 | 72.60 | 35.20 | 26.60 |
| SnapKV | 10% | 14.55 | 14.90 | 15.80 | 65.90 | 76.87 | 73.20 | 28.80 |
| PyramidKV | | 9.55 | 9.90 | 11.52 | 62.22 | 76.93 | 36.20 | 27.60 |
| Ada SnapKV | | 0.90 | 0.90 | 5.00 | 45.86 | 69.40 | 58.60 | 32.60 |
| OracleKV | | 24.55 | 23.00 | 54.76 | 30.60 | 60.73 | 27.80 | 28.60 |

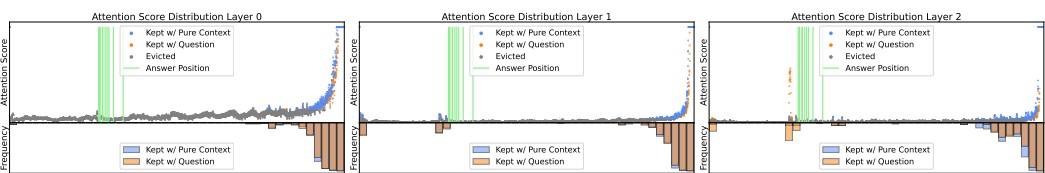

Figure A13: Attention distribution of first three layers in Common Words Extraction (CWE) task.

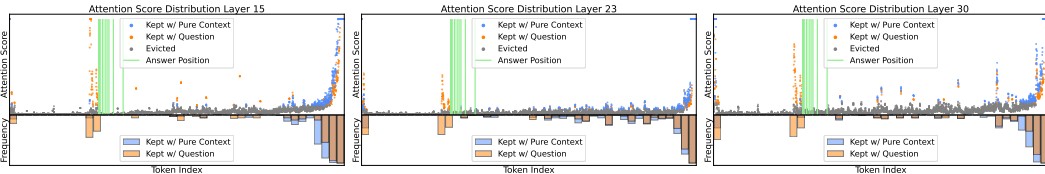

Figure A14: Attention distribution of layer 15, 23, 30 in Common Words Extraction (CWE) task.

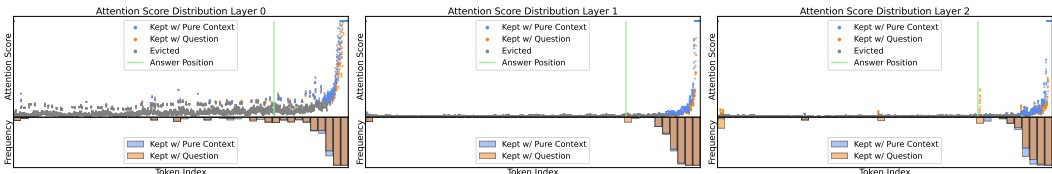

Figure A15: Attention distribution of first three layers in multi-key needle in a haystack (NIAH) task.

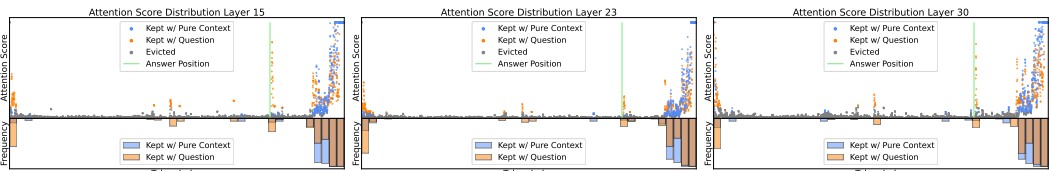

Figure A16: Attention distribution of first layer 15, 23, 30 in multi-key needle in a haystack (NIAH) task.

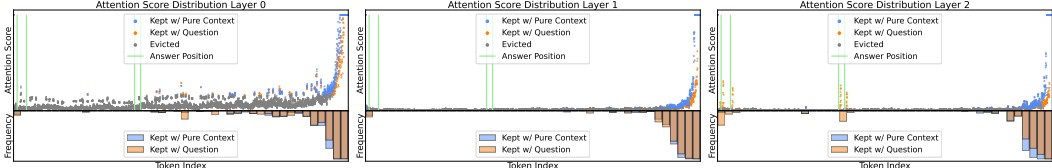

Figure A17: Attention distribution of first three layers in multi-query needle in a haystack (NIAH) task.

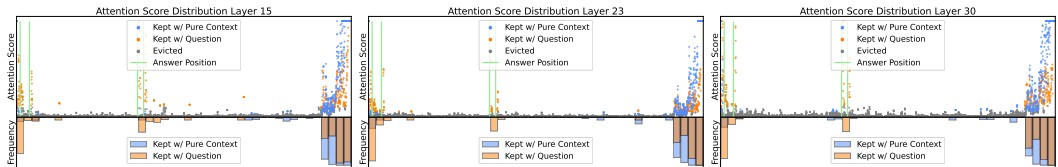

Figure A18: Attention distribution of first layer 15, 23, 30 in multi-query needle in a haystack (NIAH) task.

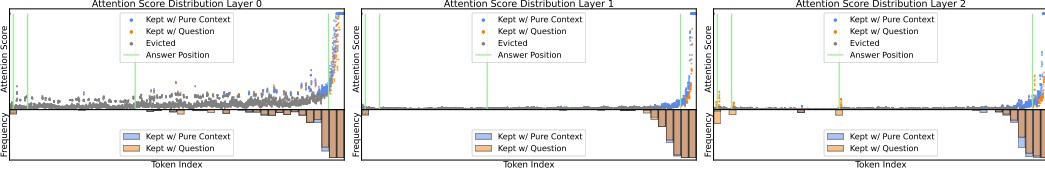

Figure A19: Attention distribution of first three layers in multi-value needle in a haystack (NIAH) task.

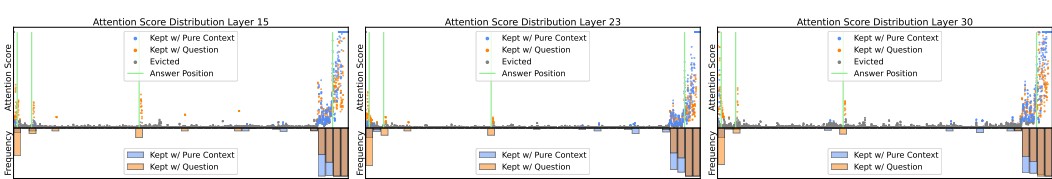

Figure A20: Attention distribution of first layer 15, 23, 30 in multi-value needle in a haystack (NIAH) task.

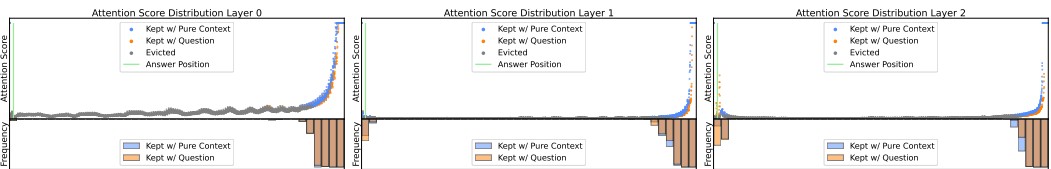

Figure A21: Attention distribution of first three layers in single needle in a haystack (NIAH) task.

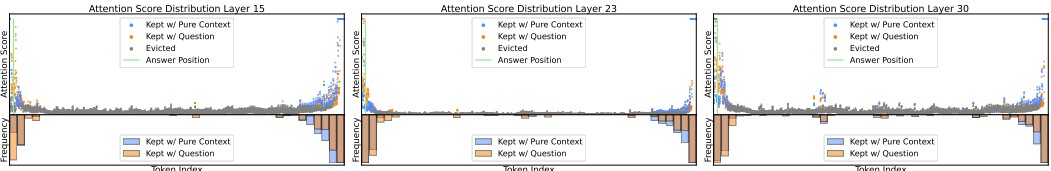

Figure A22: Attention distribution of first layer 15, 23, 30 in single needle in a haystack (NIAH) task.

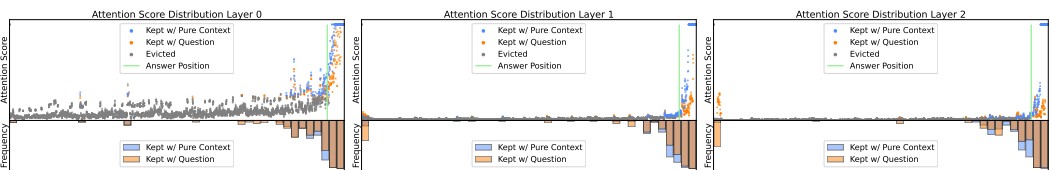

Figure A23: Attention distribution of first three layers in question-answering (QA) task.

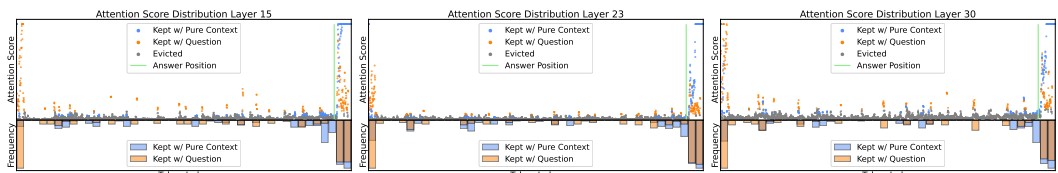

Figure A24: Attention distribution of first layer 15, 23, 30 in question-answering (QA) task.

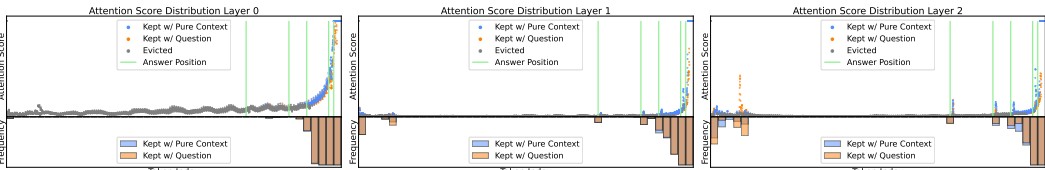

Figure A25: Attention distribution of first three layers in variable tracing (VT) task.

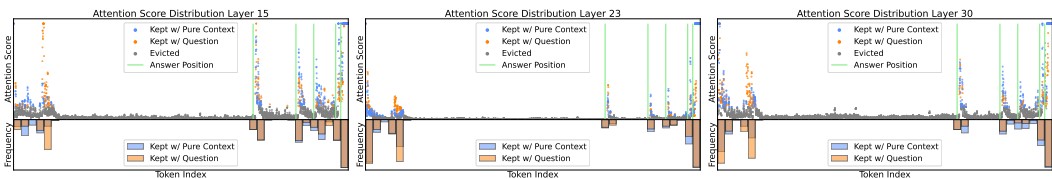

Figure A26: Attention distribution of first layer 15, 23, 30 in variable tracing (VT) task.

