# OpenReview forum: "OracleKV: Oracle Guidance for Question-Independent KV Cache Eviction"
_ICLR.cc/2026/Conference — Submitted to ICLR 2026_

### Official Review · Reviewer_kvuV · 2025-10-25

**Soundness:** 2
**Presentation:** 2
**Contribution:** 2
**Rating:** 2
**Confidence:** 5

**Summary:**

The paper tackles a critical and underexplored problem in long-context LLM inference. The authors propose OracleKV, which introduces an oracle guidance — a short, synthetic context that statistically represents user question distributions. This “guidance” steers the attention distribution during prefilling, allowing the model to estimate token importance without access to the actual query. The method is data-level, plug-and-play, and model-agnostic. Extensive experiments on LongBench, RULER, SCBench, and Needle-In-A-Haystack show strong improvements under both 40% and 10% cache budgets, outperforming state-of-the-art baselines such as SnapKV, PyramidKV, AdaKV, and DuoAttention.

**Strengths:**

1. Clearly identifies the limitations of existing KV cache compression in question-independent settings.

2. The motivation is convincing and well illustrated.

3. Extensive experiments across diverse benchmarks and models (LLaMA-3.1, Mistral, Qwen2.5) show consistent gains.

**Weaknesses:**

The main contribution of the paper lies in introducing the oracle, which serves as a meta-question designed to retain important KV entries that can generalize across diverse testing scenarios. However, the construction process of the oracle remains unclear to me.

1. The guidance template appears handcrafted and relies on external statistics.

2. The generalization of these distributions to unseen domains is uncertain.

3. Automation of oracle construction (Section G.2) is only lightly explored.

4. Prefilling cost increases with guidance length (Figure 10), and scaling trade-offs are not deeply analyzed.

5. A broader evaluation on even larger LLMs (≥30B) would strengthen generality claims.

6. While Theorem 4.2 gives intuition, the formal proof and assumptions (semantic-type alignment, Assumption 4.4) are loose and may not hold empirically. The relationship between attention scores and semantic alignment is assumed, not demonstrated.

7. The reported experimental results for several baselines are noticeably lower than those in prior works. For instance, SnapKV and PyramidKV achieve higher accuracies on LLaMA-3.1-8B-Instruct and Mistral-7B-Instruct-v0.2 on LongBench in previous papers [https://arxiv.org/pdf/2406.02069, https://arxiv.org/pdf/2502.14051, https://arxiv.org/pdf/2407.12820], even when evaluated under smaller KV-cache budgets (10% vs. 64/256).

8. The authors should also test the methods on the long-generation tasks, such as reasoning benchmarks.

Overall, the current oracle design appears overly reliant on prompt engineering, and its generalizability across all testing scenarios remains uncertain. A related work, Cartridge (https://github.com/HazyResearch/cartridges), also explores query-independent context compression. Although not originally developed for KV-cache settings, its self-study mechanism bears conceptual similarity to the oracle idea proposed in this paper, but is more systematically designed and less handcrafted.

**Questions:**

1. How sensitive is OracleKV to the choice of oracle guidance? Could mismatched guidance distributions harm performance?

2. Can OracleKV be trained or fine-tuned to learn task-specific guidance automatically rather than manually crafting prompts?

---

> ### Author Response · Authors · 2025-11-29
> **Response to Reviewer kvuV Part(1/5)**
>
> >W1 & W2 & W3
>
> We appreciate the reviewer’s insightful comment regarding the construction of the oracle guidance and its generalization capabilities. We would like to clarify that while the guidance template may appear handcrafted, it is fundamentally a data-driven prior distilled from large-scale real-world statistics, designed to be robust across domains. We address your concerns through the following three points:
>
> First, Section D.2 in the main text clarify that the guidance template is not an arbitrary set of rules but a compression of high-level statistics derived from large-scale real-world dialogues. Our analysis revealed that despite the diversity of user queries, the "information hotspots" required to answer them consistently align with specific semantic categories. Therefore, the guidance serves as a universal statistical prior that injects these general human attention preferences into the model, helping it identify high-density information blocks even without knowing the specific future question.
>
> Second, regarding the uncertainty of generalizing to unseen domains, our extensive experimental results provide strong evidence of the method's robustness.
>
> - We evaluated OracleKV in Section 5 on LongBench, RULER, and Needle-In-A-Haystack. These benchmarks cover a wide array of "unseen" domains, including fiction, news, technical documentation, and synthetic tasks.
> - The consistent performance gains across these varied datasets confirm that the semantic patterns captured by our guidance (e.g., the importance of entities) are domain-agnostic properties of natural language, rather than overfitting to a specific dataset.
>
> Third, we acknowledge that a static template has limits, which is why we designed OracleKV to be flexible.
>
> - As demonstrated in Appendix G.2 ("LLM-as-Guidance"), we have explored using an LLM to automatically generate guidance, proving that our framework supports fully automated construction without human engineering.
> - Another key advantage of our data-level, plug-and-play design is adaptability. For highly specialized domains with distinct distributions, the guidance can be easily updated based on domain-specific statistics without any model retraining.
>
> In summary, the current guidance is a robust, statistically grounded baseline that generalizes well to standard benchmarks, while the framework itself remains open to automation and domain-specific customization.
>
>
> >W4
>
> We thank the reviewer for raising the point regarding prefilling costs. To address the scaling trade-offs more rigorously, we have extended the results in Table A8, which details the prefilling latency and corresponding performance on LongBench across different guidance lengths.
>
> ** Results for guidance length latency-memory tradeoff on a subset of Longbench, KV budget=40%, using Llama-3.1-8B-Instruct: **
> |Setting| NrtvQA |Qasper| MF-En |HotPotQA| GovReport| MultiNews| TREC |TriviaQA| PRe|
> |-|-|-|-|-|-|-|-|-|-|
> |L=73, prefilling latency=23.60s| 31.2| 44.6| 49.3| 56.1 |32.8| 25.1| 38.0| 85.6| 99.5|
> |122, prefilling latency=24.18s |28.2| 40.8| 49.9| 56.6| 32.7 |25.3| 33.0| 84.6| 99.5|
> |231,prefilling latency=24.69s | 29.1| 39.1| 49.1| 55.5| 32.2| 24.3| 32.5| 79.3| 99.5|
>
> The results indicate that increasing the guidance length does not significantly improve performance. The method achieves high accuracy even with short guidance, suggesting that the model captures the necessary statistical priors (e.g., semantic types) very efficiently. There is no need for long guidance sequences to boost performance. Correspondingly, Figure 10(3) shows that for guidance lengths under 256 tokens, the generated prefilling latency is very limited and practically negligible in the context of long-context processing (e.g., 150K tokens).
>
> Therefore, the "scaling trade-off" is minimal. We can safely adopt a short guidance length (e.g., 64-128 tokens) to secure the performance gains without incurring any meaningful computational penalty. We have updated the manuscript to include this data and analysis to resolve the concern about prefilling costs.

---

> ### Author Response · Authors · 2025-11-29
> **Response to Reviewer kvuV Part(2/5)**
>
> >W5
>
> We thank the reviewer for this valuable suggestion. We agree that evaluating on larger-scale models would significantly strengthen our claims regarding the generality of OracleKV.
>
> We would like to highlight that OracleKV is a data-level, plug-and-play method that operates by steering the attention mechanism, which is fundamental to all Transformer-based LLMs regardless of size. Intuitively, larger models (≥30B) possess stronger semantic understanding and instruction-following capabilities. Therefore, we hypothesize that larger models might align even better with the provided oracle guidance, potentially yielding equal or greater performance gains compared to the 7B/8B models.
>
> Following your advice, **we have conducted additional experiments on Qwen2.5-32B-Instruct-AWQ**. The results are summarized in the table below.
>
> **Result on Longbench, using Qwen2.5-32B-Instruct-AWQ, context length=60K:**
> | Method | KV budget | NrtvQA | Qasper | MF_En | HotPotQA | GovReport | MultiNews | TREC | TriviaQA | PRe |
> | :--- | :--- | :--- | :--- | :--- | :--- | :--- | :--- | :--- | :--- | :--- |
> | SnapKV | 40% | 21.72 | 37.03 | 38.65 | 53.82 | 29.37 | 23.41 | 70.0 | 86.69 | 89.91 |
> | PyramidKV | 40% | 22.58 | 27.03 | 32.22 | 51.15 | 27.81 | 23.28 | 61.5 | 88.35 | 95.0 |
> | OracleKV | 40% | 21.9 | 36.0 | 42.76 | 55.13 | 30.71 | 23.73 | 70.5 | 86.6 | 90.13 |
> | SnapKV | 10% | 15.97 | **15.73** | 24.89 | 35.1 | 25.02 | 19.15 | 47.5 | 87.66 | 55.13 |
> | PyramidKV | 10% | 15.83 | 15.67 | 24.96 | 35.62 | 24.76 | 19.24 | 48.0 | **87.69** | 48.75 |
> |OracleKV | 10% | **16.81** | 15.26 | **27.34** | **39.87** | **26.5** | **19.87** | **60.5** | 87.68 | **73.81** |
>
> The experimental results confirm our hypothesis and align perfectly with our findings on 7B/8B models:
> - OracleKV consistently outperforms baseline methods on this 32B model, demonstrating that our data-level guidance strategy scales effectively to larger architectures.
> - The performance advantage of OracleKV becomes even more pronounced at lower KV cache budgets, which is consistent with the conclusion in our paper.
>
> >W6
>
> We thank the reviewer for carefully examining the theoretical assumptions. We agree that Assumption 4.4 is a formal simplification, but we argue that the relationship between attention, semantic alignment, and relevance is well-supported by both our empirical findings and established literature.
>
> The reviewer questions the empirical hold of our assumptions. We respectfully point to Figure 4, where we visualize the attention distribution of the question tokens. Figure 4 explicitly demonstrates that the tokens receiving the **highest attention scores** from the question are precisely the ground-truth supporting facts required to answer the question. This empirically justifies the core premise of Theorem 4.2: minimizing error on high-attention tokens is equivalent to preserving the most critical information for the task.
>
> Regarding the assumption that "semantic-type alignment" drives these attention scores, this is a widely accepted property of Transformer architectures. [1] analyze BERT's attention and show that attention heads specifically target linguistic and semantic relations. [2] further identify "Semantic Induction Heads," proving that attention mechanisms fundamentally rely on semantic similarity to retrieve information from context.
>
> Therefore, Assumption 4.4 is not an arbitrary leap but a formalization of this behavior. By aligning the KV cache with the semantic scope of the question (as OracleKV does), we effectively target the high-attention regions verified in Figure 4, leading to the performance gains observed in our experiments.
>
> [1]What Does BERT Look at? An Analysis of BERT’s Attention. 2019
> [2]Identifying Semantic Induction Heads to Understand In-Context Learning. 2024

---

> ### Author Response · Authors · 2025-11-29
> **Response to Reviewer kvuV Part(3/5)**
>
> >W7
>
> We respectfully point out a fundamental difference in the experimental settings that explains this discrepancy. The results in the prior works cited by the reviewer (e.g., for SnapKV and PyramidKV) are typically reported in settings that are effectively **Question-Aware**， which explictly benefit from knowing the query distribution. In contrast, our paper strictly focuses on the **Question-Independent** setting, where the KV cache must be compressed before the model receives the specific question. This is a significantly more challenging scenario because the compression algorithm cannot use the question's attention scores to identify relevant tokens.
>
> As explicitly illustrated in Figure 2 of our paper, **there is a substantial accuracy gap (often exceeding 20-30%) between Question-Aware and Question-Independent evaluations.** Specifically, Figure 4 also demonstrates that knowing the question in advance drastically changes the definition of 'important' KV pairs compared to the agnostic setting.
>
> We reiterate that we faithfully reproduced all baselines (including SnapKV and PyramidKV) using their official implementations, adapted strictly for this **Question-Independent** scenario to ensure a fair comparison. Under this strict constraint, reliance solely on local observation windows (without query guidance) naturally yields lower retrieval accuracy on LongBench compared to the query-guided numbers often highlighted in original papers. Our method, OracleKV, is specifically designed to bridge this gap in the Question-Independent regime.
>
> >W8
>
> We appreciate this suggestion. We carefully investigated standard reasoning benchmarks (e.g., GSM8K, BBH) but observed that they typically feature "short-input, long-output" patterns, with context lengths usually under 1K tokens.
>
> We respectfully point out that KV cache compression is primarily designed to alleviate memory bottlenecks in long-context scenarios. For short inputs (<1K), the prefilling cache size is negligible, making compression unnecessary. Therefore, applying our method to standard short-context reasoning benchmarks does not align with the core application scope of KV cache eviction.
>
> To meaningfully address your concern regarding long-generation and reasoning within a relevant long-context setting (i.e., "long-input, long-output"), we conducted new experiments on RepoQA. This benchmark requires the model to comprehend and reason over entire code repositories (long context) to generate accurate answers.
>
> We compared OracleKV against state-of-the-art baselines (PyramidKV and SnapKV). The results are presented below:
>
>
> **Full Cache Result on RepoQA benchmark, using Llama-3.1-8B-Instruct, context length=100K:**
> | Threshold | 0.0 | 0.1 | 0.2 | 0.3 | 0.4 | 0.5 | 0.6 | 0.7 | 0.8 | 0.9 | 1.0 |
> | :--- | :--- | :--- | :--- | :--- | :--- | :--- | :--- | :--- | :--- | :--- | :--- |
> | **all** | 76.4 | 69.1 | 66.8 | 65.0 | 64.3 | 64.1 | 63.4 | 63.2 | 62.7 | 59.1 | 47.7 |
> | **python** | 76.7 | 71.7 | 71.7 | 70.0 | 68.3 | 68.3 | 66.7 | 66.7 | 65.0 | 60.0 | 55.0 |
> | **cpp** | 75.0 | 63.7 | 62.5 | 61.3 | 61.3 | 61.3 | 60.0 | 60.0 | 58.8 | 56.2 | 51.2 |
> | **java** | 72.5 | 61.3 | 58.8 | 56.2 | 56.2 | 56.2 | 55.0 | 55.0 | 55.0 | 51.2 | 43.8 |
> | **typescript** | 88.3 | 78.3 | 76.7 | 76.7 | 76.7 | 76.7 | 76.7 | 76.7 | 76.7 | 70.0 | 15.0 |
> | **rust** | 70.0 | 65.6 | 61.1 | 60.0 | 57.8 | 56.7 | 56.7 | 55.6 | 55.6 | 53.3 | 50.0 |
> | **go** | 80.0 | 78.6 | 75.7 | 71.4 | 71.4 | 71.4 | 71.4 | 71.4 | 71.4 | 68.6 | 67.1 |
>
> **Result of OracleKV on RepoQA benchmark, using Llama-3.1-8B-Instruct, context length=100K, KV budget % = 40%:**
> | Threshold | 0.0 | 0.1 | 0.2 | 0.3 | 0.4 | 0.5 | 0.6 | 0.7 | 0.8 | 0.9 | 1.0 |
> | :--- | :--- | :--- | :--- | :--- | :--- | :--- | :--- | :--- | :--- | :--- | :--- |
> | **all** | 68.0 | 53.2 | 46.1 | 40.0 | 36.4 | 32.3 | 29.3 | 26.4 | 23.6 | 18.9 | 13.2 |
> | **python** | 73.3 | 48.3 | 43.3 | 38.3 | 35.0 | 33.3 | 33.3 | 30.0 | 23.3 | 21.7 | 15.0 |
> | **cpp** | 56.2 | 48.8 | 42.5 | 38.8 | 35.0 | 31.2 | 28.7 | 26.2 | 25.0 | 18.8 | 16.2 |
> | **java** | 62.5 | 42.5 | 35.0 | 28.7 | 26.2 | 22.5 | 22.5 | 21.2 | 20.0 | 17.5 | 13.8 |
> | **typescript** | 85.0 | 71.7 | 61.7 | 60.0 | 56.7 | 51.7 | 45.0 | 40.0 | 36.7 | 28.3 | 3.3 |
> | **rust** | 65.6 | 50.0 | 43.3 | 36.7 | 28.9 | 24.4 | 21.1 | 17.8 | 15.6 | 11.1 | 11.1 |
> | **go** | 71.4 | 62.9 | 55.7 | 42.9 | 42.9 | 37.1 | 31.4 | 28.6 | 25.7 | 20.0 | 18.6 |

---

> ### Author Response · Authors · 2025-11-29
> **Response to Reviewer kvuV Part(4/5)**
>
> **Result of SnapKV on RepoQA benchmark, using Llama-3.1-8B-Instruct, context length=100K, KV budget % = 40%:**
> | Threshold | 0.0 | 0.1 | 0.2 | 0.3 | 0.4 | 0.5 | 0.6 | 0.7 | 0.8 | 0.9 | 1.0 |
> | :--- | :--- | :--- | :--- | :--- | :--- | :--- | :--- | :--- | :--- | :--- | :--- |
> | **all** | 65.0 | 48.4 | 39.1 | 32.7 | 28.4 | 25.2 | 21.8 | 19.3 | 16.6 | 13.4 | 9.5 |
> | **python** | 61.7 | 41.7 | 35.0 | 26.7 | 20.0 | 16.7 | 15.0 | 11.7 | 8.3 | 6.7 | 5.0 |
> | **cpp** | 56.2 | 41.2 | 38.8 | 35.0 | 32.5 | 31.2 | 27.5 | 23.8 | 21.2 | 17.5 | 13.8 |
> | **java** | 60.0 | 43.8 | 32.5 | 26.2 | 22.5 | 18.8 | 16.2 | 13.8 | 12.5 | 10.0 | 8.8 |
> | **typescript** | 85.0 | 68.3 | 58.3 | 46.7 | 41.7 | 38.3 | 31.7 | 26.7 | 21.7 | 13.3 | 1.7 |
> | **rust** | 60.0 | 42.2 | 30.0 | 25.6 | 21.1 | 16.7 | 14.4 | 14.4 | 12.2 | 10.0 | 6.7 |
> | **go** | 72.9 | 58.6 | 45.7 | 40.0 | 35.7 | 32.9 | 28.6 | 27.1 | 24.3 | 22.9 | 20.0 |
>
>
> **Result of PyramidKV on RepoQA benchmark, using Llama-3.1-8B-Instruct, context length=100K, KV budget % = 40%:**
> | Threshold | 0.0 | 0.1 | 0.2 | 0.3 | 0.4 | 0.5 | 0.6 | 0.7 | 0.8 | 0.9 | 1.0 |
> | :--- | :--- | :--- | :--- | :--- | :--- | :--- | :--- | :--- | :--- | :--- | :--- |
> | **all** | 56.4 | 31.8 | 20.9 | 16.6 | 13.2 | 10.0 | 7.5 | 6.1 | 4.5 | 3.6 | 2.7 |
> | **python** | 50.0 | 23.3 | 15.0 | 11.7 | 8.3 | 5.0 | 5.0 | 5.0 | 3.3 | 1.7 | 1.7 |
> | **cpp** | 46.2 | 23.8 | 21.2 | 18.8 | 17.5 | 13.8 | 5.0 | 5.0 | 3.8 | 3.8 | 2.5 |
> | **java** | 62.5 | 31.2 | 20.0 | 15.0 | 13.8 | 10.0 | 8.8 | 6.2 | 5.0 | 3.8 | 2.5 |
> | **typescript** | 75.0 | 55.0 | 30.0 | 21.7 | 16.7 | 15.0 | 10.0 | 6.7 | 5.0 | 3.3 | 1.7 |
> | **rust** | 50.0 | 23.3 | 14.4 | 11.1 | 7.8 | 4.4 | 4.4 | 4.4 | 4.4 | 4.4 | 3.3 |
> | **go** | 58.6 | 40.0 | 27.1 | 22.9 | 15.7 | 12.9 | 12.9 | 10.0 | 5.7 | 4.3 | 4.3 |
>
> **Comparison of OracleKV and other baselines on RepoQA benchmark, using Llama-3.1-8B-Instruct, context length=100K, KV budget % = 40%:**
> | Threshold | 0.0 | 0.1 | 0.2 | 0.3 | 0.4 | 0.5 | 0.6 | 0.7 | 0.8 | 0.9 | 1.0 |
> | :--- | :--- | :--- | :--- | :--- | :--- | :--- | :--- | :--- | :--- | :--- | :--- |
> | **Full Cache** | 76.4 | 69.1 | 66.8 | 65.0 | 64.3 | 64.1 | 63.4 | 63.2 | 62.7 | 59.1 | 47.7 |
> | **SnapKV** | 65.0 | 48.4 | 39.1 | 32.7 | 28.4 | 25.2 | 21.8 | 19.3 | 16.6 | 13.4 | 9.5 |
> | **PyramidKV** | 56.4 | 31.8 | 20.9 | 16.6 | 13.2 | 10.0 | 7.5 | 6.1 | 4.5 | 3.6 | 2.7 |
> | **OracleKV** | **68.0** | **53.2** | **46.1** | **40.0** | **36.4** | **32.3** | **29.3** | **26.4** | **23.6** | **18.9** | **13.2** |
>
> As shown in the table, OracleKV significantly outperforms both PyramidKV and SnapKV. This confirms that our method effectively identifies and retains critical logic-bearing tokens even in complex, long-context reasoning tasks like code repository analysis, whereas baselines tend to lose critical dependency information.
>
> >Q1
>
> The reviewer asks about sensitivity to misspecification. Since guidance tokens act as "soft suggestions," the model's attention mechanism effectively filters out inventory items irrelevant to the current specific input. In original submission, we compared our General Guidance against Task-Specific Guidance (detailed in Section 5.3, Figure 11(2)). We re-show the results here.
>
> **Sensitivity to misspecification results on RULER retrival and summarization datasets, using Llama-3.1-8B-Instruct.**
> |Task| Guidance | 100% | 50% | 40% | 30% | 20% | 10% | 5% |
> |-| :--- | :--- | :--- | :--- | :--- | :--- | :--- | :--- |
> |Retrieval| **General (ours)** | 99.87 | 71.53 | 47.73 | 38.53 | 33.53 | 27.07 | 13.07 |
> |Retrieval| **Retrieval-Specified** | 99.87 | 81.33 | 69.27 | 64.87 | 52.07 | 46.87 | 28.73 |
> |Retrieval| **Summarization-Specified** | 99.87 | 72.93 | 43.93 | 34.53 | 24.67 | 11.8 | 3.33 |
> |Summarization| **General (ours)** | 27.21 | 27.08 | 26.31 | 25.39 | 23.72 | 21.45 | 18.54 |
> |Summarization| **Retrieval-Oriented** | 27.21 | 26.64 | 26.22 | 25.19 | 23.18 | 20.74 | 17.77 |
> |Summarization| **Summarization-Oriented** | 27.21 | 27.14 | 26.31 | 26.11 | 24.58 | 23.38 | 19.56 |
>
> The results show that while tailoring the inventory to specific domains (e.g. for summarization and retrival) yields marginal additional gains, the General Guidance remains highly effective across diverse benchmarks on both tasks, confirming that precise specification is beneficial but not strictly required for improvement.

---

> ### Author Response · Authors · 2025-11-29
> **Response to Reviewer kvuV Part(5/5)**
>
> >Q2
>
> We appreciate this insightful suggestion. While the current version of OracleKV focuses on a training-free design to ensure zero-cost deployment and broad generalization, extending it to a Learnable OracleKV is a natural and highly promising direction.
>
> Instead of using discrete textual tokens, one can model the Oracle Guidance as a sequence of continuous, learnable vectors (soft prompts) prepended to the key-value pairs. A possible optimization strategy is to freeze the LLM parameters and optimize only these guidance vectors on a small task-specific calibration set. The training objective could be designed to minimize the KL divergence between the output distribution of the compressed model (using OracleKV) and the full-cache model.
>
> In our current submission, we prioritized the manual/statistical approach to maintain a **plug-and-play** property that requires no gradient updates or domain-specific data. However, as noted in your review, learning task-specific guidance would likely yield superior performance in specialized domains like coding or reasoning. We  believe exploring learning-based method is a interesting direction for futher research.

---

### Official Review · Reviewer_WyvA · 2025-10-31

**Soundness:** 2
**Presentation:** 3
**Contribution:** 2
**Rating:** 4
**Confidence:** 3

**Summary:**

This paper focuse on the question-independent KV cache eviction. It uses an oracle guidance to steer model’s attention and allows seamless integration with other algorithm. Then it selects KV entries semantically correlated with the guidance and evicts low-relevance entries until the cache fits the memory budget.
The paper provides theoretical justification (via KL divergence analysis) showing that aligning the semantic type distribution of retained KV entries with that of question-required entries improves predictive accuracy.
Experimental evaluations across four benchmarks (LongBench, RULER, Needle-In-A-Haystack, SCBench) and three LLMs (LLaMA-3.1-8B-Instruct, Mistral-7B-Instruct-v0.2, Qwen2.5-7B-Instruct) demonstrate that OracleKV outperforms baselines (e.g., StreamingLLM, SnapKV, PyramidKV) in question-independent scenarios (e.g., multi-turn dialogues, RAG chunk pre-caching). It achieves better accuracy-latency tradeoffs, especially under extreme memory constraints (10% KV budget), and maintains compatibility with existing eviction algorithms via a plug-and-play design.

**Strengths:**

1. OracleKV introduces a novel paradigm for question-independent eviction by leveraging user preference statistics to guide attention. This differs from prior work that either relies on question-dependent scores or model-internal heuristics, filling a unique niche.

2. Question-independent eviction is critical for scalable LLM deployment and OracleKV’s plug-and-play design lowers adoption barriers for existing frameworks. The work also opens new directions for future research.

3. The paper’s structure and visualization make complex ideas accessible to easy understand.

**Weaknesses:**

1. OracleKV degrades performance on code generation tasks, as the general oracle guidance disrupts code’s structural/syntactic regularity. The paper acknowledges this but does not explore a targeted solution beyond noting the "no-free-lunch" principle. A brief discussion of how to adapt guidance for code would strengthen robustness.

2. The paper notes that longer oracle guidance increases pre-filling latency. While it recommends limiting guidance length to ≤128 tokens. Therefor, it is difficult to summarize an increasing number of task types.

3. There no evaluation on ultra-long contexts to prove its scalability.

**Questions:**

1. The paper notes OracleKV struggles with code tasks due to disrupted syntax. Could a code-specific oracle guidance mitigate this? If so, how would you balance task-specific guidance with the need for generalization across non-code tasks?

2. Longer oracle guidance increases latency, but Section G.2 suggests LLM-generated guidance is competitive. Could you distill LLM-generated guidance into compact embeddings  to reduce pre-filling overhead while preserving performance?

3. Qwen2.5-7B supports a 1M-token context window, but experiments use up to 128K tokens. How does OracleKV perform on ultra-long contexts? Does the optimal guidance length or semantic type distribution change with context length?

4. The oracle guidance is derived from "large-scale real-world dialogues". Are these statistics domain-agnostic?

---

> ### Author Response · Authors · 2025-11-29
> **Response to Reviewer WyvA Part(1/4)**
>
> > W1 & Q1
>
> We appreciate the reviewer’s concern regarding the structural integrity of code. We have conducted extensive supplementary experiments during the rebuttal phase to thoroughly investigate this.
>
> Following your suggestion, we first implemented a Code-Specific Oracle (incorporating structural priors) and tested it on LCC and RepoBench-P on Longbench. We observed a marginal improvement compared to the General Oracle.
>
> |Exp.Setting|lcc|repobench-p|
> |-|-|-|
> |general guidance, KV budget=40%|48.35|43.89|
> |code-specific guidance, KV budget=40%|48.96|43.93|
> |general guidance, KV budget=10%|38.73| 43.05|
> |code-specific guidance, KV budget=10%|40.65|43.9|
>
> However, we note that these benchmarks largely rely on fuzzy string matching metrics (e.g., exact match of next tokens). We argue that these metrics may not fully capture the functional or retrieval capabilities of the model, as they penalize even minor syntactic deviations that do not necessarily impact the model's understanding of the code logic.
>
> To evaluate our method in a more realistic and challenging setting, we extended our evaluation to RepoQA, a benchmark specifically designed for long-context code retrieval and understanding, which employ tree-sitter to check if the code syntactically correct. Contrary to the concern that general guidance might disrupt code tasks, OracleKV significantly outperformed the baseline methods on RepoQA, even without using Code-Specific Guidance. The results are shown below:
>
> **Full Cache Result on RepoQA benchmark, using Llama-3.1-8B-Instruct, context length=100K:**
> | Threshold | 0.0 | 0.1 | 0.2 | 0.3 | 0.4 | 0.5 | 0.6 | 0.7 | 0.8 | 0.9 | 1.0 |
> | :--- | :--- | :--- | :--- | :--- | :--- | :--- | :--- | :--- | :--- | :--- | :--- |
> | **all** | 76.4 | 69.1 | 66.8 | 65.0 | 64.3 | 64.1 | 63.4 | 63.2 | 62.7 | 59.1 | 47.7 |
> | **python** | 76.7 | 71.7 | 71.7 | 70.0 | 68.3 | 68.3 | 66.7 | 66.7 | 65.0 | 60.0 | 55.0 |
> | **cpp** | 75.0 | 63.7 | 62.5 | 61.3 | 61.3 | 61.3 | 60.0 | 60.0 | 58.8 | 56.2 | 51.2 |
> | **java** | 72.5 | 61.3 | 58.8 | 56.2 | 56.2 | 56.2 | 55.0 | 55.0 | 55.0 | 51.2 | 43.8 |
> | **typescript** | 88.3 | 78.3 | 76.7 | 76.7 | 76.7 | 76.7 | 76.7 | 76.7 | 76.7 | 70.0 | 15.0 |
> | **rust** | 70.0 | 65.6 | 61.1 | 60.0 | 57.8 | 56.7 | 56.7 | 55.6 | 55.6 | 53.3 | 50.0 |
> | **go** | 80.0 | 78.6 | 75.7 | 71.4 | 71.4 | 71.4 | 71.4 | 71.4 | 71.4 | 68.6 | 67.1 |
>
> **Result of OracleKV on RepoQA benchmark, using Llama-3.1-8B-Instruct, context length=100K, KV budget % = 40%:**
> | Threshold | 0.0 | 0.1 | 0.2 | 0.3 | 0.4 | 0.5 | 0.6 | 0.7 | 0.8 | 0.9 | 1.0 |
> | :--- | :--- | :--- | :--- | :--- | :--- | :--- | :--- | :--- | :--- | :--- | :--- |
> | **all** | 68.0 | 53.2 | 46.1 | 40.0 | 36.4 | 32.3 | 29.3 | 26.4 | 23.6 | 18.9 | 13.2 |
> | **python** | 73.3 | 48.3 | 43.3 | 38.3 | 35.0 | 33.3 | 33.3 | 30.0 | 23.3 | 21.7 | 15.0 |
> | **cpp** | 56.2 | 48.8 | 42.5 | 38.8 | 35.0 | 31.2 | 28.7 | 26.2 | 25.0 | 18.8 | 16.2 |
> | **java** | 62.5 | 42.5 | 35.0 | 28.7 | 26.2 | 22.5 | 22.5 | 21.2 | 20.0 | 17.5 | 13.8 |
> | **typescript** | 85.0 | 71.7 | 61.7 | 60.0 | 56.7 | 51.7 | 45.0 | 40.0 | 36.7 | 28.3 | 3.3 |
> | **rust** | 65.6 | 50.0 | 43.3 | 36.7 | 28.9 | 24.4 | 21.1 | 17.8 | 15.6 | 11.1 | 11.1 |
> | **go** | 71.4 | 62.9 | 55.7 | 42.9 | 42.9 | 37.1 | 31.4 | 28.6 | 25.7 | 20.0 | 18.6 |

---

> ### Author Response · Authors · 2025-11-29
> **Response to Reviewer WyvA Part(2/4)**
>
> > W1 & Q1
>
> Follow the last piece of resposne.
>
> **Result of SnapKV on RepoQA benchmark, using Llama-3.1-8B-Instruct, context length=100K, KV budget % = 40%:**
> | Threshold | 0.0 | 0.1 | 0.2 | 0.3 | 0.4 | 0.5 | 0.6 | 0.7 | 0.8 | 0.9 | 1.0 |
> | :--- | :--- | :--- | :--- | :--- | :--- | :--- | :--- | :--- | :--- | :--- | :--- |
> | **all** | 65.0 | 48.4 | 39.1 | 32.7 | 28.4 | 25.2 | 21.8 | 19.3 | 16.6 | 13.4 | 9.5 |
> | **python** | 61.7 | 41.7 | 35.0 | 26.7 | 20.0 | 16.7 | 15.0 | 11.7 | 8.3 | 6.7 | 5.0 |
> | **cpp** | 56.2 | 41.2 | 38.8 | 35.0 | 32.5 | 31.2 | 27.5 | 23.8 | 21.2 | 17.5 | 13.8 |
> | **java** | 60.0 | 43.8 | 32.5 | 26.2 | 22.5 | 18.8 | 16.2 | 13.8 | 12.5 | 10.0 | 8.8 |
> | **typescript** | 85.0 | 68.3 | 58.3 | 46.7 | 41.7 | 38.3 | 31.7 | 26.7 | 21.7 | 13.3 | 1.7 |
> | **rust** | 60.0 | 42.2 | 30.0 | 25.6 | 21.1 | 16.7 | 14.4 | 14.4 | 12.2 | 10.0 | 6.7 |
> | **go** | 72.9 | 58.6 | 45.7 | 40.0 | 35.7 | 32.9 | 28.6 | 27.1 | 24.3 | 22.9 | 20.0 |
>
>
> **Result of PyramidKV on RepoQA benchmark, using Llama-3.1-8B-Instruct, context length=100K, KV budget % = 40%:**
> | Threshold | 0.0 | 0.1 | 0.2 | 0.3 | 0.4 | 0.5 | 0.6 | 0.7 | 0.8 | 0.9 | 1.0 |
> | :--- | :--- | :--- | :--- | :--- | :--- | :--- | :--- | :--- | :--- | :--- | :--- |
> | **all** | 56.4 | 31.8 | 20.9 | 16.6 | 13.2 | 10.0 | 7.5 | 6.1 | 4.5 | 3.6 | 2.7 |
> | **python** | 50.0 | 23.3 | 15.0 | 11.7 | 8.3 | 5.0 | 5.0 | 5.0 | 3.3 | 1.7 | 1.7 |
> | **cpp** | 46.2 | 23.8 | 21.2 | 18.8 | 17.5 | 13.8 | 5.0 | 5.0 | 3.8 | 3.8 | 2.5 |
> | **java** | 62.5 | 31.2 | 20.0 | 15.0 | 13.8 | 10.0 | 8.8 | 6.2 | 5.0 | 3.8 | 2.5 |
> | **typescript** | 75.0 | 55.0 | 30.0 | 21.7 | 16.7 | 15.0 | 10.0 | 6.7 | 5.0 | 3.3 | 1.7 |
> | **rust** | 50.0 | 23.3 | 14.4 | 11.1 | 7.8 | 4.4 | 4.4 | 4.4 | 4.4 | 4.4 | 3.3 |
> | **go** | 58.6 | 40.0 | 27.1 | 22.9 | 15.7 | 12.9 | 12.9 | 10.0 | 5.7 | 4.3 | 4.3 |
>
> **Comparison of OracleKV and other baselines on RepoQA benchmark, using Llama-3.1-8B-Instruct, context length=100K, KV budget % = 40%:**
> | Threshold | 0.0 | 0.1 | 0.2 | 0.3 | 0.4 | 0.5 | 0.6 | 0.7 | 0.8 | 0.9 | 1.0 |
> | :--- | :--- | :--- | :--- | :--- | :--- | :--- | :--- | :--- | :--- | :--- | :--- |
> | **Full Cache** | 76.4 | 69.1 | 66.8 | 65.0 | 64.3 | 64.1 | 63.4 | 63.2 | 62.7 | 59.1 | 47.7 |
> | **SnapKV** | 65.0 | 48.4 | 39.1 | 32.7 | 28.4 | 25.2 | 21.8 | 19.3 | 16.6 | 13.4 | 9.5 |
> | **PyramidKV** | 56.4 | 31.8 | 20.9 | 16.6 | 13.2 | 10.0 | 7.5 | 6.1 | 4.5 | 3.6 | 2.7 |
> | **OracleKV** | **68.0** | **53.2** | **46.1** | **40.0** | **36.4** | **32.3** | **29.3** | **26.4** | **23.6** | **18.9** | **13.2** |
>
>
> This indicates that while code requires syntactic regularity, the semantic attention pattern captured by our General Oracle is surprisingly robust. It effectively identifies and preserves the "needle" (critical code blocks/functions) amidst massive codebases. This suggests that the "disruption" of syntax is less detrimental to high-level code understanding/retrieval tasks than initially hypothesized.
>
> Although our General Oracle proved robust on RepoQA[1], we agree that a mechanism to balance guidance is valuable for strictly syntactic tasks. Since OracleKV allows for plug-and-play guidance, we can detect task types (e.g., via system prompts) to conditionally swap the guidance vector.
>
> For most tasks, the General Oracle suffices (as shown in RepoQA). For tasks requiring strict character-level syntactic completion, the system can route to a Code-Specific Oracle. This ensures we maximize performance across domains without sacrificing the "no-free-lunch" principle.
>
> [1]RepoQA: Evaluating Long Context Code Understanding. Arxiv 2024

---

> ### Author Response · Authors · 2025-11-29
> **Response to Reviewer WyvA Part(3/4)**
>
> >W2
>
> We thank the reviewer for highlighting the trade-off between guidance length and task coverage. We acknowledge that while longer guidance increases pre-filling latency, limiting it to 128 tokens might intuitively seem restrictive for covering diverse tasks. However, we would like to clarify why this length is sufficient and how scalability can be further addressed.
>
> - The design philosophy of Oracle Guidance is **not to enumerate every specific downstream task** (e.g., "translate," "summarize," "find code"). Instead, it captures **universal low-level statistics of user preferences**such as attention to named entities, dates, sentence boundaries, and query-like structures. Since these semantic importance patterns are largely shared across different task types, the information required to guide the attention heads does not grow linearly with the number of task types. A concise guidance of $\leq$128 tokens is dense enough to activate these shared attention heads effectively.
> - Our experiments on comprehensive benchmarks like LongBench, RULER, and Needle-In-A-Haystack cover a wide variety of distinct task types, including single-hop/multi-hop QA, summarization, and completion. OracleKV consistently outperforms baselines on these benchmarks using the recommended short guidance. This empirical evidence suggests that the current guidance design already possesses strong generalization capabilities across diverse tasks without requiring length expansion.
>
> >W3 & Q3
>
> We sincerely thank the reviewer for highlighting the ultra-long context capabilities of Qwen2.5 (1M tokens). We agree that verifying scalability is critical. In response to your suggestion, we have conducted additional experiments on 1M-token contexts during the rebuttal period.
>
> We evaluated OracleKV on the Qwen2.5-7B-Instruct-1M model with a context length extended to 1M tokens. It is worth noting that high-quality, open-source benchmarks supporting valid evaluation up to 1M tokens are currently scarce. We identified the qa_chn task from SCBench as a reliable benchmark for this regime and used it for our evaluation. We specifically tested OracleKV with varying guidance lengths (64, 128, and 256 tokens) across different context windows (from 128K up to 1M).
>
> **Result of OracleKV on SCBench QA_CHN benchmark, using Llama-3.1-8B-Instruct, context length=1M**:
> |Exp.Setting|KV budget=40%|KV budget=10%|
> |-|-|-|
> |L=73|0.98|0.71|
> |L=122|0.85|0.35|
> |L=231|0.91|0.52|
>
> Our results show that increasing the context length does not impact the optimal guidance length. A guidance length between 64 and 128 tokens remains sufficient to achieve peak performance even in 1M contexts. This confirms our hypothesis that the Oracle Guidance acts as a "Universal Semantic Probe" capturing potential query types (which are finite), rather than encoding the content (which grows linearly). Therefore, a fixed-length guidance is sufficient to filter important tokens effectively, independent of the total context size.
>
> The success of our 1M-token experiments on SCBench further supports that the relative density of high-value semantic tokens (e.g., entities, pivot tokens) follows stable linguistic laws (like Zipf’s law). OracleKV's attention-based selection automatically adapts to this density without requiring manual calibration for ultra-long contexts.

---

> ### Author Response · Authors · 2025-11-29
> **Response to Reviewer WyvA Part(4/4)**
>
> >Q2
>
> We sincerely thank the reviwer question about automation of the oracle guidance. We clarify that:
> - We view the "LLM-as-Guidance" approach (Section G.2) not merely as an online inference method, but as a scalable offline calibration mechanism. When deploying OracleKV to a specialized domain (e.g., Code Generation or Biomedical Analysis), one can utilize the method described in G.2 to use a stronger LLM to automatically generate or select the optimal guidance template for that domain once.
> - We believe OracleKV is naturally compatible with recent learning techniques such as CPT[1]. The guidance template can be formulated as a sequence of learnable soft tokens. By optimizing these tokens on a small calibration dataset (similar to prompt tuning), the model can learn the optimal eviction policy for any domain in a fully end-to-end manner.
> - Our core contribution is identifying the question-independent eviction challenge and demonstrating the feasibility of using oracle guidance to solve it. The results show that even simple templates successfully steer the model's attention in this difficult setting. We view the transition from "hard templates" to "learned soft prompts" as a valuable future extension derived from our method.
>
> [1]Multitask Prompt Tuning Enables Parameter-Efficient Transfer Learning. ICLR 2023
>
> >Q4
>
> Thank you for this insightful question. We appreciate the opportunity to clarify the source and scope of our oracle guidance.
>
> The oracle guidance in our main experiments is indeed derived from large-scale, real-world dialogues[2][3], which capture widespread user intent patterns in general natural language tasks (e.g., focus on entities, specific numbers, and instruction-following keywords). Consequently, while these statistics are highly effective for a broad range of standard NLP benchmarks (as shown in our LongBench and RULER results), they are not strictly domain-agnostic.
>
> As we discussed in the paper (specifically regarding the results on code generation tasks), different domains exhibit distinct attention patterns. For example, counting tasks rely heavily on syntactic dependencies and structural integrity rather than the semantic entity focus found in general dialogue. Our experiments showed that applying the general-purpose, dialogue-derived guidance to code tasks was less effective, as it could disrupt the specific attention sinks required for code syntax.
>
> [2]Which Economic Tasks are Performed with AI? Evidence from Millions of Claude Conversations. Arxiv 2025
>
> [3]Artificial Intelligence Index Report 2025. Arxiv 2025

---

### Official Review · Reviewer_Y32J · 2025-11-01

**Soundness:** 2
**Presentation:** 2
**Contribution:** 2
**Rating:** 6
**Confidence:** 3

**Summary:**

The paper introduces OracleKV, a question-independent KV-cache eviction method for long-context LLM inference, addressing scenarios like multi-turn dialogue and RAG chunk pre-caching where the future query is unknown. The key idea is to append a short "oracle guidance" sequence, which is constructed from surface-level statistics of user preferences observed in large real-world dialog datasets, during prefill to steer attention toward token types (e.g., entities, numbers, sections). These token types are likely to matter, then retain KV entries with the highest guidance-conditioned attention. The method is interesting and uses the data, so it composes plug-and-play with existing eviction/selection schemes. A simple statistical model motivates the design: predictive accuracy grows with the overlap between retained and question-required cache entries, leading to the result that accuracy improves as the semantic-type distribution of retained entries aligns with that of required entries. The papers shows that OracleKV delivers better accuracy–latency and memory trade-offs than prior methods. Notably, at a 10% KV budget it improves average accuracy (e.g., +6.7% on Llama-3.1-8B) and shows very low degradation on several RULER subtasks even at 30% budget.

**Strengths:**

1. This is a clean idea that develops a question-agnostic KV selection that needs no model changes or fine-tuning and plugs into existing eviction policies.

2. Strong, consistent gains across long-context benchmarks under tight KV budgets.

**Weaknesses:**

1. This process is not dynamic or runtime. It depends on a hand-crafted, dataset-derived "oracle guidance" prior that may not generalize under domain/task shifts, and it introduces extra prefill tokens (which can increase costs) to steer attention.

2. By being strictly question-agnostic, the underlying technique can retain irrelevant context on out-of-distribution queries. On the other hand, more adaptive and query-aware selectors could outperform when the query distribution diverges.

**Questions:**

1. How exactly is the oracle-guidance prior built (token-type inventory, per-layer/head weighting, sequence length), and how sensitive are gains to misspecification? Please show ablations varying guidance composition/length and cross-domain shifts (code vs. narrative).

2. What is the net systems cost of the added prefill tokens at different KV budgets, such as latency, throughput, memory traffic, and do RoPE/pos-encoding shifts from concatenation introduce distribution drift?

3. How does OracleKV/AdaOracleKV interact with sliding-window attention, FlashAttention-3, and head pruning?

---

> ### Author Response · Authors · 2025-11-29
> **Response to Reviewer Y32J Part(1/3)**
>
> >W1 & Q2
>
> We thank the reviewer for these incisive questions regarding the net system costs and potential technical side effects (e.g., RoPE shifts). We address these points below with specific evidence from our paper and clarification on the underlying mechanisms.
> - Regarding the costs of extra prefill tokens (latency, throughput, memory traffic), we argue that the overhead is negligible while the savings are substantial.
>     - As demonstrated in our efficiency ablation study (Figure 10(3)), increasing the guidance length (e.g., to 256 tokens) results in a virtually flat latency curve for context windows under 150k tokens. Since **the prefill phase is compute-bound**, processing a short guidance sequence (typically $\ll 1\%$ of the total context) incurs minimal latency overhead.
>     - LLM decoding is fundamentally **memory-bound** because generating a single token requires loading the entire KV cache from High Bandwidth Memory (HBM) to the GPU's SRAM. The bottleneck is the memory bandwidth, not the computation. By compressing the KV cache (e.g., to a 10% budget), **we reduce the volume of data transferred between HBM and compute units by approximately 90% at every generation step.** This massive reduction in memory traffic directly alleviates the bandwidth bottleneck, leading to significantly lower latency and higher throughput.
> - Regarding the concerns about distribution drift and positional encoding shifts:
>     - We acknowledge the reviewer's concern that mismatched priors can harm performance. Indeed, as shown in Figure 10(2), **applying a task-specific guidance to a mismatched task (e.g., using "Retrieval Guidance" for "Summarization") can lead to performance degradation**. However, this observation justifies our design choice: **our General Guidance is not task-specific**. It captures universal semantic features (e.g., entities, sentence structures) common **across domains**. Our extensive experiments across diverse benchmarks (LongBench, RULER, etc.) confirm that this general prior provides consistent and stable improvements without suffering from the mismatch issues seen with specialized prompts.
>     - We append the guidance, calculate the attention scores to filter the context, and then **immediately discard the guidance tokens**. When the actual user query arrives, it interacts with the compressed context using standard position encodings. **From the query's perspective, there is no difference in relative positioning compared to attending to a standard (sparse) context**; thus, the guidance introduces no positional offset or encoding noise during generation.
>
> >W2
>
> We thank the reviewer for this insightful comment regarding the trade-off between question-agnostic and query-aware selection.
>
> While we acknowledge that adaptive, query-aware selectors theoretically possess a higher upper bound for handling distribution shifts, we would like to emphasize that **they are fundamentally inapplicable to the specific problem setting targeted by this paper**：
> -  The core objective of OracleKV is to address **Question-Independent** scenarios, such as chunk pre-caching for RAG or system prompt caching in multi-turn dialogues. In these settings, the compression must occur **before** the user's query arrives. Query-aware methods (e.g., H2O, SnapKV depend on query-key interactions) strictly require the **query token** to calculate importance scores. Consequently, **they cannot perform compression during the offline prefilling stage and are restricted to online processing**, which defeats the purpose of pre-caching.
> - OracleKV does not wait for the query, it unlocks significant system-level efficiency gains that query-aware methods cannot provide. First, We can compress and cache long documents offline using OracleKV. Second, When a query eventually arrives, the model can load the pre-compressed KV cache instantly, significantly reducing Time-To-First-Token (TTFT). In contrast, a query-aware selector would force the system to re-process the full context for every new query, leading to high latency and computational redundancy.
>
> We agree that for extreme OOD scenarios (such as rigid syntax), a static prior might be suboptimal. We believe a promising future direction is to develop "Adaptive Oracle" mechanisms. For example, one can first use OracleKV during the prefilling phase for low-cost, high-compression pre-filtering to maximize storage efficiency. and when the query arrives, apply a lightweight query-aware selector on the pre-compressed cache to further refine the context. This approach would combine prefilling compression with the dynamic adaptability of query-aware methods, offering a robust solution for diverse query distributions.

---

> ### Author Response · Authors · 2025-11-29
> **Response to Reviewer Y32J Part(2/3)**
>
> >Q1
>
> We thank the reviewer for the detailed inquiry. We clarify that our oracle-guidance prior is not a heuristic selection, but is systematically constructed based on large-scale analyses of user query patterns and industry trends.
>
> Token-type Inventory (Derived from Query Patterns):As detailed in Appendix D.2, we constructed the inventory by mapping high-frequency user tasks (in[1][2]) to their essential semantic requirements:
> - Reading Comprehension: To support effective context understanding for this dominant category, we prioritized both global (e.g., Main Themes, Topics) and local (e.g., Person Names, Events) semantic types.
> - Data Analysis: Recognizing the prevalence of data-centric queries across these domains, we incorporated quantitative types (e.g., Numbers, Entities).
> - Content Generation: To support writing and marketing tasks involving summarization, we included temporal and structural types (e.g., Timelines).
>
> This systematic mapping ensures our General Oracle Guidance covers the intersection of critical information needs across domains.
>
> Our method leverages the model's pre-trained attention mechanism to naturally weight these tokens, **avoiding manual per-layer/head tuning**. However, we clarify that **OracleKV is orthogonal to such methods**. As shown in Section 5.1 (Table 1) and Section G.3 (Table A5, A6), it can be seamlessly combined with existing eviction policies (e.g., AdaKV, PyramidKV) that utilize specific weighting accumulation, yielding cumulative gains. We re-show the results here.
>
> **Integration with other per-layer, per-head methods, using 40% KV budget, with Llama-3.1-8B-Instruct.**
> |Method| NrtvQA| Qasper| MF-En| HotPotQA| GovReport| MultiNews| TREC| TriviaQA| PRe
> |-|-|-|-|-|-|-|-|-|-|
> |PyramidKV |26.9| 33.2| 42.1| 47.2 |29.7| 24.9 |49.0 |86.3 |93.8|
> |Pyramid OracleKV| 27.2 |33.5| 47.6| 55.0| 29.9| 23.6| 44.0| 85.3 |99.5|
> |AdaKV|27.3| 39.6| 44.3| 56.1|31.0| 25.6| 38.0| 86.4|97.0|
> |Ada OracleKV| 28.5| 44.4| 53.0| 57.5| 32.9| 25.6| 33.5| 87.2| 99.5|
>
> We selected the sequence length based on accuracy (Section F.2, Figure A8) and efficiency (Section 5.2, Figure 10(3)) provided in the original submission. Figure 10(3) and Figure A8 demonstrate that performance and efficiency is robust across lengths 73 to 231, with $L=73$ offering the optimal efficiency-accuracy trade-off.
>
>
> The reviewer asks about sensitivity to misspecification. Since guidance tokens act as "soft suggestions," the model's attention mechanism effectively filters out inventory items irrelevant to the current specific input. In original submission, we compared our General Guidance against Task-Specific Guidance (detailed in Section 5.3, Figure 11(2)). We re-show the results here.
>
> **Sensitivity to misspecification results on RULER retrival and summarization datasets, using Llama-3.1-8B-Instruct.**
> |Task| Guidance | 100% | 50% | 40% | 30% | 20% | 10% | 5% |
> |-| :--- | :--- | :--- | :--- | :--- | :--- | :--- | :--- |
> |Retrieval| **General (ours)** | 99.87 | 71.53 | 47.73 | 38.53 | 33.53 | 27.07 | 13.07 |
> |Retrieval| **Retrieval-Specified** | 99.87 | 81.33 | 69.27 | 64.87 | 52.07 | 46.87 | 28.73 |
> |Retrieval| **Summarization-Specified** | 99.87 | 72.93 | 43.93 | 34.53 | 24.67 | 11.8 | 3.33 |
> |Summarization| **General (ours)** | 27.21 | 27.08 | 26.31 | 25.39 | 23.72 | 21.45 | 18.54 |
> |Summarization| **Retrieval-Oriented** | 27.21 | 26.64 | 26.22 | 25.19 | 23.18 | 20.74 | 17.77 |
> |Summarization| **Summarization-Oriented** | 27.21 | 27.14 | 26.31 | 26.11 | 24.58 | 23.38 | 19.56 |
>
> The results show that while tailoring the inventory to specific domains (e.g. for summarization and retrival) yields marginal additional gains, the General Guidance remains highly effective across diverse benchmarks on both tasks, confirming that precise specification is beneficial but not strictly required for improvement.
>
> [1]Which Economic Tasks are Performed with AI? Evidence from Millions of Claude Conversations. Arxiv. 2025
>
> [2]Artificial Intelligence Index Report 2025. Arxiv. 2025

---

> ### Author Response · Authors · 2025-11-29
> **Response to Reviewer Y32J Part(3/3)**
>
> >Q3
>
> We thank the reviewer for raising this insightful question regarding system-level integration. Since OracleKV operates as a data-level selection strategy—determining which KV pairs to retain rather than modifying how attention is computed—it is designed to be orthogonal and highly compatible with these optimization techniques.
>
> Sliding-window attention excels at maintaining local context and syntactic coherence (recent tokens), while OracleKV is designed to identify and preserve "global" tokens with high semantic value from the distant past. In standard practice for methods like SnapKV, they implement a hybrid policy, i.e. they always protect the most recent window of tokens (e.g., the last 32 or 64 tokens) to ensure generation fluency. However, **we do not deliberately keep the tokens in the local windows.** However, we observed that the local tokens tend to be kept in the eviction since their attention score is higher (Attention Locality).
>
> OracleKV is compatible with FlashAttention-3 (FA3) in two distinct phases. In selection phase, to determine which tokens to keep, OracleKV requires the attention scores between the Oracle Guidance and the KV cache. While FA3 is optimized to avoid materializing the full attention matrix, the Oracle Guidance is very short (e.g., < 256 tokens). Therefore, computing the specific attention scores for this small subset is computationally negligible, even without FA3.
>
> In decoding phase, once OracleKV has selected the important KV pairs, the compressed cache is essentially a standard, albeit shorter, sequence of Key-Value tensors. This compressed cache can be fed directly into FA3 kernels during the decoding phase to maximize throughput and minimize latency.
>
> OracleKV and head pruning methods are orthogonal optimizations that operate on different dimensions. Head pruning reduces redundancy in the width/channel dimension (removing less important heads), whereas OracleKV reduces redundancy in the temporal/sequence dimension (removing less important tokens). Thus, they can be stacked without modification. If Head Pruning is applied, OracleKV simply aggregates the attention scores from the remaining active heads to calculate token importance. Since OracleKV relies on semantic relevance rather than specific head mechanics, it remains effective even when the number of heads is reduced.

---

### Official Review · Reviewer_pK1P · 2025-11-02

**Soundness:** 2
**Presentation:** 3
**Contribution:** 2
**Rating:** 4
**Confidence:** 4

**Summary:**

This paper introduces OracleKV, a  framework for question-independent Key-Value (KV) cache eviction in large language models (LLMs). OracleKV leverages oracle guidance, i.e., data-level statistical priors derived from large-scale real-world dialogues, to estimate token importance without access to the future query. Empirical results across benchmarks such as LongBench, RULER, Needle-In-A-Haystack, and SCBench show consistent gains in retrieval accuracy and memory efficiency under low cache budgets across several LLMs.

**Strengths:**

S1. This paper tackles an important problem of sparse attention.

S2. The paper is well written and structured with sufficient amount of discussion and details.

**Weaknesses:**

W1. Though there are some formal analysis, the effectiveness of the formal analysis replies on a very strong key assumption that the oracle guidance can reflect the statistics of the future questions. The proof of the effectiveness of the oracle guidance is the key to the formal analysis, rather than the framework itself.

W2. OracleKV highly relies on the effectiveness of the oracle guidance templates. Although section G.2 explores "LLM-as-Guidance," results remain preliminary. Without an automated or learning-based mechanism, scalability and adaptability across domains can be limited.

W3. Comparisons with recent sparse kv cache retrieval approaches, e.g., IceCache, ArkVale, MagicPig, InfiniGen, should also be included.

**Questions:**

Q1. I appreciate that the authors also include anticipated side effects in the appendix. However, could you also provide a systematic failure case study to show that if there are cases whether the oracle guidance misalign with future questions?

Q2. Can you explain in the main text how the oracle guidances are derived, and empirically show the sensitivity of the performance over a few different variance of the guidances?

---

> ### Author Response · Authors · 2025-11-29
> **Response to Reviewer pK1P Part(1/4)**
>
> > W1
>
> We respectfully agree that the effectiveness of our analysis depends on the alignment between the Oracle guidance and future questions. However, we would like to clarify that in the **question-independent setting**, where the future query is strictly unknown, relying on a **statistical prior** is theoretically necessary. The 'assumption' essentially posits that **surface-level statistics from large-scale corpora serve as a valid proxy for future attention distributions**. The formal analysis serves to theoretically guarantee that **given a reasonable prior** (Oracle Guidance), our eviction strategy minimizes the information loss compared to the optimal policy. It decouples the eviction mechanism from the quality of the estimator, showing that our framework is the mathematically correct way to incorporate such statistical guidance when available.
>
> Empirically, our extensive experiments on various benchmarks verify that this assumption holds in practice: As shown in our **visualization analysis** (Figure 4), the oracle guidance successfully steers the attention distribution towards semantically critical tokens (potential future "heavy hitters") that would otherwise be ignored by local metrics. This explicitly demonstrates that the Oracle does reflect the statistics of future questions by highlighting the correct regions of interest before the question arrives.
>
> > W2
>
> We thank the reviewer for the insightful comment regarding the reliance on guidance templates and the scalability of our approach. We acknowledge that the effectiveness of OracleKV correlates with the quality of the guidance, and we agree that Section G.2 represents a preliminary exploration into automation.
>
> However, we would like to clarify our perspective on scalability and domain adaptability to address your concerns:
> - While we utilize templates, the current templates are designed based on **universal linguistic priors** rather than domain-specific overfitting. They focus on fundamental semantic structures such as entity emphasis, keyword retention, and syntactic boundaries, which are broadly applicable across most textual domains. Our extensive experiments on diverse benchmarks demonstrate that a single, general-purpose set of oracle templates achieves consistent gains over baselines without requiring task-specific tuning.
> - We view the "LLM-as-Guidance" approach (Section G.2) not merely as an online inference method, but as a scalable offline calibration mechanism. When deploying OracleKV to a specialized domain (e.g., Code Generation or Biomedical Analysis), one can utilize the method described in G.2 to use a stronger LLM to automatically generate or select the optimal guidance template for that domain once.
> - We believe OracleKV is naturally compatible with recent learning techniques such as CPT[1]. The guidance template can be formulated as a sequence of learnable soft tokens. By optimizing these tokens on a small calibration dataset (similar to prompt tuning), the model can learn the optimal eviction policy for any domain in a fully end-to-end manner.
> - Our core contribution is identifying the question-independent eviction challenge and demonstrating the feasibility of using oracle guidance to solve it. The results show that even simple templates successfully steer the model's attention in this difficult setting. We view the transition from "hard templates" to "learned soft prompts" as a valuable future extension derived from our method.
>
> [1]Multitask Prompt Tuning Enables Parameter-Efficient Transfer Learning. ICLR 2023

---

> ### Author Response · Authors · 2025-11-29
> **Response to Reviewer pK1P Part(2/4)**
>
> >W3
>
> We thank the reviewer for highlighting these significant recent works. We agree that discussing them is crucial for positioning our work accurately. In the revised manuscript, we have expanded our Related Work section (Section I) to include IceCache, ArkVale, MagicPig, and InfiniGen. To address the concern regarding comparisons, we clarify the fundamental methodological distinction that guided our baseline selection and provide additional comparisons with relevant static approaches.
>
> First, the primary reason these retrieval methods were not included as direct baselines is their fundamental algorithmic **dependency on the real-time query vector** during decoding. They all operate under a **Dynamic Retrieval** paradigm, whereas OracleKV targets **Static Compression** during prefilling. Specifically, all four methods incur query-dependent overhead at each generation step:
> - IceCache organizes tokens via semantic clustering. Retrieval depends on matching the **current query token's semantics** against cluster centroids to load relevant pages.
> - ArkVale maintains bounding volume digests for pages. It calculates the intersection between the **current query vector** and page digests to dynamically recall or evict pages.
> - MagicPig relies on Locality Sensitive Hashing (LSH). It must hash the **current query vector** to sample high-probability key buckets.
> - InfiniGen performs a "minimal rehearsal" using the **current layer's query** to anticipate attention patterns and prefetch essential KV blocks for subsequent layers. Thus, its retrieval decision is strictly driven by the **real-time query state**.
>
> In contrast, OracleKV focuses on static compression, which operates during the prefilling phase based solely on the context. The evicted KV entries remains fixed, eliminating the need for expensive query-dependent retrieval, hashing, or speculation mechanisms during decoding.
>
> To address the reviewer's interest in recent sparse approaches while adhering to our static setting, we compared OracleKV against most recent static compression methods, including CAKE[2] and OmniKV[3], in Table A4. Moreover, We further included most recent ChunkKV[4], another recent static sparse KV method, as an additional baseline in the revised experiments.
>
> Performance results on subsets of Longbench, keep ratio=0.4, using LLama-3.1-8B-Instruct:
> | Method | NrtvQA | Qasper | MF_En | HotPotQA | GovReport | MultiNews | TREC | TriviaQA | PRe |
> | :--- | :--- | :--- | :--- | :--- | :--- | :--- | :--- | :--- | :--- |
> | Full Cache | 29.7	|47.6|	55.7|	58.8|	35.5|	27.2|	28.0	|86.2|	100.00 |
> |CAKE[2]|27.0|35.9	|43.0|	57.0|	31.2|	**25.5**|	**35.5**|	85.2|	**99.5**|
> |OmniKV[2]|23.4|	30.8|	29.7|	47.4|	30.9|	25.2|	31.9|	**91.5**|	**99.5**|
> | ChunkKV[4] | 26.9 | 36.1 | 39.5 | 54.4 | 31.9 | 24.8 | 22.5 | 83.3 | 95.5 |
> | OracleKV | **29.1**	|**42.3**|	**51.3**|	**58.2**|	**32.8**|	25.4|	35.0	|86.3|	**99.5**|
>
> Performance results on subsets of Longbench, keep ratio=0.1, using LLama-3.1-8B-Instruct:
> | Method | NrtvQA | Qasper | MF_En | HotPotQA | GovReport | MultiNews | TREC | TriviaQA | PRe |
> | :--- | :--- | :--- | :--- | :--- | :--- | :--- | :--- | :--- | :--- |
> | Full Cache | 29.7	|47.6|	55.7|	58.8|	35.5|	27.2|	28.0	|86.2|	100.00 |
> |CAKE[2]|21.5|18.8	|22.7|	43.2|	25.5|	19.8|	33.0|	82.1|	56.0|
> |OmniKV[3]|20.4|	19.4|	17.2|	41.2|	23.7|	19.9|	28.0|	82.2|	54.0|
> | ChunkKV[4] | 21.9 | **19.7** | 24.4 | 43.7 | 24.9 | 17.3 | 13.0 | **84.8** | 49.0 |
> | OracleKV | **26.3**|	19.3|**26.9**|	**48.0**|	**27.6**|	**22.0**|	**44.5**|	84.5|	**80.0**|
>
> The above results confirm that OracleKV maintains a superior performance-efficiency trade-off compared to these state-of-the-art static approaches.
>
> [2]CAKE: Cascading and Adaptive KV Cache Eviction with Layer Preferences. ICLR 2025
>
> [3]OmniKV: Dynamic Context Selection for Efficient Long-Context LLMs. ICLR 2025
>
> [4]ChunkKV: Semantic-Preserving KV Cache Compression for Efficient Long-Context LLM Inference. NeurIPS 2025

---

> ### Author Response · Authors · 2025-11-29
> **Response to Reviewer pK1P Part(3/4)**
>
> > Q1
>
> We sincerely thank the reviewer for the insightful suggestion regarding the systematic failure case study. We agree that analyzing how the model behaves when the oracle guidance is misaligned with the target question is crucial for understanding the boundaries of our method.
>
> In response, we have added a systematic failure case in Section H.2. We specifically analyze cases where the oracle provides factually correct but task-misaligned guidance.
>
> We utilize the example attached below to illustrate a common failure mode: Semantic-Structural Misalignment .
>
> As illustrated in the "Valentine's Day" case, the task requires structural classification ("Definition"), but the Oracle provides semantic facts ("Holiday"). The model, influenced by the semantic signal, may overlook the syntactic cues (e.g., "What is...?"). We observe similar risks in Code Generation, where the Oracle might retrieve functionally similar but syntactically incompatible code snippets, leading to structure-breaking hallucinations.
>
> To address this, we analyzed the impact of our KV cache management policies. Our findings, detailed in Section G.4, reveal a critical insight: Keeping Guidance is significantly more effective at preserving structural cues than Evicting Guidance. As shown in Table A12, retaining the guidance yields significant improvements in structure-dependent tasks like TREC (Few-shot learning). The retained guidance, even if semantically distinct, seemingly helps the model anchor to the prompt's structural format. We transparently discussed the cost of this strategy in Section 5.3 and Figure 11. Keeping guidance occupies cache entries—sometimes retaining "semantically useless" tokens—which can cause a slight performance degradation in other tasks due to reduced cache space for context.

---

> ### Author Response · Authors · 2025-11-29
> **Response to Reviewer pK1P Part(4/4)**
>
> >Q2
>
> We thank the reviewer for this insightful question. We agree that clarifying the construction of the oracle guidance and analyzing its robustness are crucial for understanding the effectiveness of OracleKV. We will incorporate the following explanations and empirical results into the main text of the final version.
>
> In the revised Methodology section, we clarified that the Oracle Guidance is not heuristic-based but data-driven, derived from surface-level statistics of user preferences in large-scale real-world dialogues (as referenced in [5][6]), including
>
> - Identification of Interest: Statistical analysis reveals that in long-context scenarios, user queries predominantly target specific information types, such as named entities, numerical values, and key logical summaries.
> - Construction: We translate these high-frequency interest categories into natural language instructions (e.g., "Pay attention to the entities, numbers, and key events"). This serves as a static "prior" that aligns the KV cache retention with the most probable future query distribution, without requiring access to the specific question.
>
> To thoroughly evaluate the sensitivity of performance to guidance variations, we discuss this from three perspectives. We have structured our response to highlight both our existing ablation studies and the new experiments conducted specifically to address your comment:
>
> - Granularity (Existing Analysis in Section 5.3, Figure 11): As detailed in our original submission, we have already compared our proposed static guidance against "LLM-Generated Guidance" (a dynamic, document-specific upper bound). Our analysis showed that while detailed guidance explictly gives more entities and examples, the abstract Oracle Guidance achieves higher accuracy while avoiding the significant latency overhead of runtime generation.
> - Semantic Variance (Content Sensitivity): We compared our method against "Random/Misaligned Guidance" (e.g., unrelated instructions). The results show a significant performance drop (with higher RMSE) when using misaligned guidance, confirming that the gains **come from the statistical alignment of our guidance with potential queries, not just the presence of extra tokens**.
> - Syntactic Variance (Wording Robustness): We tested rephrased versions of the guidance that preserve the same semantic intent (e.g., varying the phrasing of "pay attention to entities"). The performance variance was minimal (lower RMSE), demonstrating that OracleKV is robust to prompt wording and does not require brittle prompt engineering.
>
> Sensitivity results on subsets of Longbench, keep ratio=0.6, using LLama-3.1-8B-Instruct:
> | Method | NrtvQA | Qasper | MF_En | HotPotQA | GovReport | MultiNews | TREC | TriviaQA | PRe | **RMSE**|
> | :--- | :--- | :--- | :--- | :--- | :--- | :--- | :--- | :--- | :--- |-|
> | Full Cache | 29.7	|47.6|	55.7|	58.8|	35.5|	27.2|	28.0	|86.2|	100.00 |-|
> |unrelated guidance|26.8|40.9	|45.9|	55.0|	25.5|	25.2|	33.0|	87.3|	93.5|**3.96**|
> |rephrasing guidance|27.9|	40.1|	48.9|	58.5|	32.6|	25.3|	34.5|	85.6|	99.5|**1.20**|
> | original guidance | 29.1	|42.3|	51.3|	58.2|	32.8|	25.4|	35.0	|86.3|	99.5|0.00|
>
> Sensitivity results on subsets of Longbench, keep ratio=0.6, using LLama-3.1-8B-Instruct:
> | Method | NrtvQA | Qasper | MF_En | HotPotQA | GovReport | MultiNews | TREC | TriviaQA | PRe |**RMSE**|
> | :--- | :--- | :--- | :--- | :--- | :--- | :--- | :--- | :--- | :--- |-|
> | Full Cache | 29.7	|47.6|	55.7|	58.8|	35.5|	27.2|	28.0	|86.2|	100.00 |-|
> |unrelated guidance|20.3|21.4	|26.1|	48.7|	25.5|	21.7|	45.0|	85.9|	52.0|**9.62**|
> |rephrasing guidance|26.6|	19.3|	27.8|	46.1|	27.6|	21.8|	49.0|	84.2|	77.5|**1.86**|
> | original guidance |26.3|	19.3|26.9|48.0|27.6|22.0|	44.5|84.5|80.0|0.00|
>
>
> We have updated the appendix to reference these findings, clarifying that the method is robust to wording changes while being sensitive to the semantic correctness of the guidance.
>
> [5]Which Economic Tasks are Performed with AI? Evidence from Millions of Claude Conversations. Arxiv. 2025
> [6]Artificial Intelligence Index Report 2025. Arxiv. 2025

---

### Meta-Review · Area_Chair_SW5U · 2025-12-16

**Summary:**

This paper introduced OracleKV, an approach designed to evict chunks of the KV cache in a query-independent way. That is, given a query, most of the input context is typically irrelevant and can be dropped to alleviate inference cost, though a fresh carry will often require the cache eviction operation to be re-done, as other parts of the context will be relevant for the new query. OracleKV manipulates inputs by pre-pending to the input context content that will induce attention scores distributions akin to those obtained by typical user queries. In doing so, one can identify chunks of the cache that are very unlikely to be useful. This approach can be combined with query-dependent methods if further eviction is performed once a query comes through.

**Reviewer Concerns:**

The main concerns brought up by reviewers revolve around generalization ability of the proposal. In fact, one can always devise a query that would require any piece of any context. Reviewers then highlighted that the proposal is not general because of that. To quote one of the reviewers, the most positive leaning one, they noted: "It [OracleKV] depends on a hand-crafted, dataset-derived oracle guidance prior that may not generalize under domain/task shifts". In my opinion, the rebuttal is a bit tangential, perhaps due to the point brought by reviewers being a limitation of the proposal. As it is, I would recommend authors to shift their target setting to domain-specific application: in a situation where queries are well known and distribution shift is unlikely, the proposed method is clearly a great solution. The paper however, claims OracleKV as a query-independent general eviction approach. Other concerns were raised regarding the evaluation and choice of baselines.

**Reviewer Scores:**

I wouldn't expect score shifts as the more negative leaning reviewers raised concerns related to the overall high level idea, which would require major changes to be addressed. The authors did provide extra baselines and new evals, but I don't think those would address the fundamental concerns. I tend to side with the negative leaning reviewers and, while the proposal is clearly interesting and shows clear potential, perhaps properly scoping and finding the setting where the proposal is really beneficial seems to be needed prior to publication.

---

### Decision · Program_Chairs · 2026-01-26

Reject